# Banded Square Root Matrix Factorization for Differentially Private Model Training

**Nikita Kalinin**
Institute of Science and Technology (ISTA)
Klosterneuburg, Austria
nikita.kalinin@ist.ac.at

**Christoph Lampert**
Institute of Science and Technology (ISTA)
Klosterneuburg, Austria
chl@ist.ac.at

## Abstract

Current state-of-the-art methods for differentially private model training are based on matrix factorization techniques. However, these methods suffer from high computational overhead because they require numerically solving a demanding optimization problem to determine an approximately optimal factorization prior to the actual model training. In this work, we present a new matrix factorization approach, BSR, which overcomes this computational bottleneck. By exploiting properties of the standard matrix square root, BSR allows to efficiently handle also large-scale problems. For the key scenario of stochastic gradient descent with momentum and weight decay, we even derive analytical expressions for BSR that render the computational overhead negligible. We prove bounds on the approximation quality that hold both in the centralized and in the federated learning setting. Our numerical experiments demonstrate that models trained using BSR perform on par with the best existing methods, while completely avoiding their computational overhead.

## 1 Introduction

We study the problem of *differentially private (DP) model training with stochastic gradient descent (SGD)* in the setting of either federated or centralized learning. This task has recently emerged as one of the most promising ways to train powerful machine learning models but nevertheless guarantee the privacy of the used data, which led to a number of studies, both theoretical as well as application-driven [Abadi et al., 2016, Yu et al., 2020, Zhang et al., 2021, Kairouz et al., 2021, Denisov et al., 2022]. The state of the art in the field are approaches based on the *matrix factorization (MF) mechanism* [Li et al., 2015, Henzinger et al., 2024], which combines theoretical guarantees with practical applicability [Choquette-Choo et al., 2023a,c,b, 2024][1] It is based on the observation that all iterates of SGD are simply linear combinations of model gradients, which are computed at intermediate time steps. Consequently, the iterates can be written formally as the result of multiplying the matrix of coefficients, called *workload matrix*, with the row-stacked gradient vectors. To preserve the privacy of the training data in this process one adds suitably scaled Gaussian noise at intermediate steps of the computation. The MF mechanism provides a way to select the noise covariance structure based on a factorization of the workload matrix into two matrices.

Identifying the minimal amount of noise necessary to achieve a desired privacy level requires solving an optimization problem over all possible factorizations, subject to *data participation constraints*. For some specific settings, the optimal solutions have been characterized: for *streaming learning*, when each data batch contributes at most once to the gradients, Li et al. [2015] presented a formulation of this problem as a semi-definite program. Henzinger et al. [2024] proved that a square root factorization

---

[1]Note that this specific type of MF should not be confused with other occurrences of matrix factorization in, potentially private, machine learning, such as in recommender systems [Shin et al., 2018, Li et al., 2021].

of the workload matrix is asymptotically optimal for different linear workloads, including continual summation and decaying sums.

In this work, our focus lies on the settings that are most relevant to machine learning tasks: workload matrices that reflect SGD-like optimization, and participation schemes in which each data batch can potentially contribute to more than one gradient vector, as it is the case for standard multi-epoch training. Assuming that this happens at most once every $b$ steps, for some value $b \geq 1$, leads to the problem of optimal matrix factorization in the context of $b$-*min-separated participation* sensitivity. Unfortunately, as shown in Choquette-Choo et al. [2023a], finding the optimal matrix factorization in this setting is computationally intractable. Instead, the authors proposed an approximate solution by posing additional constraints on the solution set. The result is a semi-definite program that is tractable, but still has high computational cost, making it practical only for small to medium-sized problem settings.

Subsequent work concentrated on improving or better understanding the factorizations for specific algorithms, such as plain SGD without gradient clipping, momentum, or weight decay [Koloskova et al., 2023] or specific, e.g. convex, objective functions [Choquette-Choo et al., 2024]. Often, streaming data was asssumed, i.e. each data item can contribute at most to one model update, which is easier to analyze theoretical, but further removed from real-world applications [Dvijotham et al., 2024]. A concurrent line of works has also focused on scaling matrix factorizations for large-scale training. McMahan et al. [2024] extended the Buffered Linear Toeplitz (BLT) mechanism to support multi-participation in federated learning, improving privacy-utility tradeoffs, while ensuring memory efficiency. McKenna [2024] improved DP Banded Matrix Factorization's scalability, enabling it to handle millions of iterations and large models efficiently. These advancements enhance the practicality of differentially private training in real-world applications.

Our work aims at a general-purpose solution that covers as many realistic scenarios as possible. Our ultimate goal is to make general-purpose differentially private model training as simple and efficient to use as currently dominating non-private technique. Our main contributions are:

1. We introduce a new factorization, **the *banded squared root (BSR)*, which is efficiently computable even for large workload matrices** and agnostic to the underlying training objective. For matrices stemming from SGD optimization potentially with momentum and/or weight decay, we even provide **closed form expressions.**

2. We provide **general lower bounds** on the approximate error for any factorization, along with specific **upper and lower bounds for the BSR**, Square Root, and baseline factorizations, in the contexts of both single participation (streaming) and **repeated participation** (e.g., multi-epoch) training.

3. We demonstrate experimentally that **BSR's approximation error is comparable to the state-of-the-art** method, and that **both methods also perform comparably in real-world training tasks**.

Overall, the proposed **BSR factorization achieves training high-accuracy models with provable privacy guarantees while staying computationally efficient even for large-scale training tasks.**

## 2 Background

Our work falls into the areas of *differentially private (stochastic) optimization*, of which we remind the reader here, following mostly the description of Denisov et al. [2022] and Choquette-Choo et al. [2023a]. The goal is to estimate a sequence of parameter vectors, $\Theta = (\theta_1, \ldots, \theta_n) \in \mathbb{R}^d$, where each $\theta_i$ is a linear combination of *update vectors*, $x_1, \ldots, x_i \in \mathbb{R}^d$, that were computed in previous steps, typically as gradients of a model with respect to training data that is meant to stay private. We assume that all $x_i$ have a bounded norm, $\|x_i\| \leq \zeta$. Compactly, we write $\Theta = AX$, where the lower triangular *workload matrix* $A$ contains the coefficients and $X \in \mathbb{R}^{n \times d}$ is formed by stacking the update vectors as rows. With different choices of $A$, the setting then reflects many popular first-order optimization algorithms, in particular stochastic gradient descent (SGD), potentially with momentum and/or weight decay. Depending on how exactly $x_1, \ldots, x_n$ are obtained, the setting can express different centralized as well as federated training paradigms. To formalize this aspect, we adopt the concept of $b$-*min-separated participation* [Choquette-Choo et al., 2023a]. For some integer $b \geq 1$, it states that if a data item (e.g. a single training example in central training, or a client batch in

---

**Algorithm 1** Differentially Private SGD with Matrix Factorization

---

**Input:** Initial model $\theta_0 \in \mathbb{R}^d$, dataset $D$, batchsize $b$, matrix $C \in \mathbb{R}^{n \times n}$, model loss $\ell(\theta, d)$,
   clipnorm $\zeta$, noise matrix $Z \in \mathbb{R}^{n \times d}$ with i.i.d. entries $\sim \mathcal{N}(0, s^2)$, where $s = \sigma \, \mathrm{sens}_{k,b}(C)$.
   **for** $i = 1, 2, \ldots, n$ **do**
       $S_i \leftarrow \{d_1, \ldots, d_m\} \subseteq D$    select a data batch, respecting the data participation constraints
       $g_i \leftarrow \nabla_\theta \ell(\theta_{i-1}, d_j))$   for $j = 1, \ldots, m$
       $x_i \leftarrow \sum_{j=1}^m \mathrm{clip}_\zeta(g_j)$    where $\mathrm{clip}_\zeta(d) = \min(1, \zeta/\|d\|)d$
       $\hat{x}_i \leftarrow x_i + \zeta[C^{-1}Z]_{[i,\cdot]}$
       $\theta_i \leftarrow \mathrm{update}(\theta_{i-1}, \hat{x}_i),$        // SGD model updates
**Output:** $\Theta = (\theta_1, \ldots, \theta_n)$

---

federated learning) contributed to an update $x_i$, the earliest it can contribute again is the update $x_{i+b}$. Additionally, let $1 \le k \le \frac{n}{b}$ be the maximal number any data point can contribute. In particular, this notion also allows us to treat in a unified way *streaming data* ($b = n$ or $k = 1$), as well as *unrestricted access* patterns ($k = n$ with $b = 1$), but also intermediate settings, such as *multi-epoch training* on a fixed-size dataset.

The *matrix factorization* approach [Li et al., 2015] adopts a *factorization* $A = BC$ of the workload matrix and computes $\Theta^{\mathrm{MF}} = B(CX + Z)$, where $Z$ is Gaussian noise that is chosen appropriately to make the intermediate result $CX + Z$ private to the desired level. Algorithm 1 shows the resulting algorithm in pseudocode. It exploits the fact that instead of explicit multiplication by $C$ and $B$, standard optimization toolboxes can be employed with suitably modified update vectors, because also $\Theta^{\mathrm{MF}} = A(X + C^{-1}Z)$, and multiplication by $A$ corresponds to performing the optimization.

Different factorizations recover different algorithms from the literature. For example, $B = A$, $C = \mathrm{Id}$ recovers DP-SGD [Abadi et al., 2016], where noise is added directly to the gradients. Conversely, $B = \mathrm{Id}$, $C = A$ simply adds noise to each iterate of the optimization [Dwork et al., 2006]. However, better choices than these baselines are possible, in the sense that they can guarantee the same levels of privacy with less added noise, and therefore potentially with higher retained accuracy. The reason lies in the fact that $B$ and $C$ play different roles: $B$ acts as a *post-processing* operation of already private data. Hence, it has no further effect on privacy, but it influences to what amount the added noise affects the expected error in the approximation of $\Theta$. Specifically, for $Z \sim \mathcal{N}(0; s \, \mathrm{Id})$,

$$\mathbb{E}_Z \|\Theta - \Theta^{\mathrm{MF}}\|_F^2 = \mathbb{E}_Z \|BZ\|_F^2 = s^2 \|B\|_F^2. \tag{1}$$

In contrast, $CX$ is the quantity that is meant to be made private. Doing so requires noise of a strength proportional to $C$'s *sensitivity*, $\mathrm{sens}(C) := \sup_{X \sim X'} \|CX - CX'\|_F$, where the *neighborhood relation*, $X \sim X'$, indicates that the two sequences of update vectors differ only in those entries that correspond to a single data item[2] As shown in Choquette-Choo et al. [2023a], in the setting of $b$-min-separated repeated participation, it holds that

$$\mathrm{sens}_{k,b}(C) \le \max_{\pi \in \Pi_{k,b}} \sqrt{\sum_{i,j \in \pi} \left| (C^\top C)_{[i,j]} \right|}, \tag{2}$$

where $\Pi_{k,b} = \{ \pi \subset \{1, \ldots, n\} : |\pi| \le k \wedge (\{i,j\} \subset \pi \Rightarrow i = j \vee |i - j| \ge b) \}$, is the set of possible $b$-min-separated index sets with at most $k$ participation. Furthermore, (2) holds even with equality if all entries of $C^\top C$ are non-negative.

Combining (1) with $s = \mathrm{sens}_{k,b}(C)$ yields a quantitative measure for the quality of a factorization.

**Definition 1.** *For any factorization $A = BC$, its **expected approximation error** is*

$$\mathcal{E}(B, C) := \sqrt{\mathbb{E}_Z \|\Theta - \Theta^{MF}\|_F^2 / n} = \frac{1}{\sqrt{n}} \mathrm{sens}_{k,b}(C) \|B\|_F, \tag{3}$$

*where the $1/\sqrt{n}$ factor is meant to make the quantity comparable across different problem sizes.*

---

[2] As proved in Denisov et al. [2022, Theorem 2.1], establishing privacy in this *non-adaptive* setting suffices to guarantee also privacy in the *adaptive* setting, where the update vectors depend not only on the data items but also the intermediate estimates of the model parameters, as it is the case for private model training.

The *optimal factorization* by this reasoning would be the one of smallest expected approximation error. Unfortunately, minimizing (3) across all factorizations it is generally computationally intractable. Instead, Choquette-Choo et al. [2023a] propose an *approximately optimal factorization*.

**Definition 2.** *For a workload matrix $A$, let $S$ be the solution to the optimization problem*

$$\arg \min_{S \in \mathcal{S}_+^n} \text{trace}[A^\top A S^{-1}] \quad \textit{subject to} \quad \text{diag}(S) = 1 \ \textit{and} \ S_{[i,j]} = 0 \ \textit{for} \ |i-j| \geq b, \quad (4)$$

*where $\mathcal{S}_+^n$ is the cone of positive definite $n \times n$ matrices. Then, $A = BC$ is called the **approximately optimal factorization (AOF)**, if $C$ is lower triangular and fulfills $C^\top C = S$.*

The optimization problem (4) is a semi-definite program (SDP), and can therefore be solved numerically using standard packages. However, this is computationally costly, and for large problems (e.g. $n > 5000$) computing the AOF solution is impractical. This poses a problem for real-world training tasks, where the number of update steps are commonly thousands or tens of thousands.

Solving (4) itself only approximately can mitigate this problem to some extent, but as we will discuss in Section 4, this can lead to robustness problems, especially because the recovery of $C$ from $S$ in Definition 2, e.g. by a Cholesky decomposition, tends to be sensitive to numerical errors.

## 3 Banded Square Root Factorization

In the following section, we introduce our main contribution: a general-purpose factorization for the task of differentially private stochastic optimization that can be computed efficiently even for large problem sizes.

**Definition 3** (Banded Square Root Factorization). *Let $A \in \mathbb{R}^{n \times n}$ be a lower triangular workload matrix with strictly positive diagonal entries. Then, we call $A = C^2$ the **square root factorization (SR)**, when $C$ denotes the unique matrix square root that also has strictly positive diagonal entries. Furthermore, for any* bandwidth $p \in \{1, \ldots, n\}$, *we define the **banded square root factorization of bandwidth** $p$ **($p$-BSR)** of $A$, as*

$$A = B^{|p|}C^{|p|} \quad (5)$$

*where $C^{|p|}$ is created from $C$ by setting all entries below the $p$-th diagonal to $0$,*

$$C_{[i,j]}^{|p|} = \begin{cases} C_{[i,j]} & \textit{if } i - j < p, \\ 0 & \textit{otherwise.} \end{cases} \quad \textit{and} \quad B^{|p|} = A(C^{|p|})^{-1}. \quad (6)$$

Note that determining the SR, and therefore any $p$-BSR, is generally efficient even for large workload matrices, because explicit recursive expressions exist for computing the square root of a lower triangular matrix [Björck and Hammarling, 1983, Deadman et al., 2012].

In the rest of this work, we focus on the case where the workload matrix stems from *SGD with momentum and/or weight decay*, and we show that then even closed form expressions for the entries of $C^{|p|}$ exist that renders the computational cost negligible.

### 3.1 Banded Square Root Factorization for SGD with Momentum and Weight Decay

We recall the update steps of SGD with momentum and weight decay:

$$\theta_i = \alpha\theta_{i-1} - \eta m_i \quad \text{for} \quad m_i = \beta m_{i-1} + x_i \quad (7)$$

where $x_1, \ldots, x_n$ are the update vectors, $\eta > 0$ is the *learning rate* and $0 \leq \beta < 1$ is the *momentum strength* and $0 < \alpha \leq 1$ is the weight decay parameter. Note that our results also hold for $\beta = 0$, i.e. without momentum, and for $\alpha = 1$, i.e., without weight decay. In line with real algorithms and to avoid degenerate cases, we assume $\beta < \alpha$ throughout this work. The update vectors are typically gradients of the model with respect to its parameters, but additional operations such as normalization or clipping might have been applied.

Unrolling the recursion, we obtain an expression for $\theta_i$ as a linear combination of update vectors as

$$\theta_i = \eta \sum_{j=1}^{i} x_j \left( \sum_{k=j}^{i} \alpha^{i-k}\beta^{k-j} \right). \quad (8)$$

Consequently, the workload matrix has the explicit form $A = \eta A_{\alpha,\beta}$ for

$$A_{\alpha,\beta} = \begin{pmatrix} a_0 & 0 & 0 & \dots & 0 \\ a_1 & a_0 & 0 & \dots & 0 \\ a_2 & a_1 & a_0 & \dots & 0 \\ \vdots & \vdots & \ddots & \ddots & \vdots \\ a_{n-1} & a_{n-2} & \dots & a_1 & a_0 \end{pmatrix} \quad \text{with } a_j = \sum_{i=0}^{j} \alpha^i \beta^{j-i} = \frac{\alpha^{j+1} - \beta^{j+1}}{\alpha - \beta}. \quad (9)$$

As one can see, $A_{\alpha,\beta}$ is a lower triangular Toeplitz-matrix, so it is completely determined by the entries of its first column. In the following, we use the notation $\mathrm{LDToep}(m_1, \dots, m_n)$ to denote a lower triangular Toeplitz matrix with first column $m_1, \dots, m_n$, i.e. $A_{\alpha,\beta} = \mathrm{LDToep}(a_0, \dots, a_{n-1})$.

Our first result is an explicit expression for the positive square root of $A_{\alpha,\beta}$ (and thereby its $p$-BSR).

**Theorem 1** (Square-Root of SGD Workload Matrix). *Let $A_{\alpha,\beta}$ be the workload matrix (9). Then $A_{\alpha,\beta} = C_{\alpha,\beta}^2$ for $C_{\alpha,\beta} = \mathrm{LDToep}(c_0, \dots, c_{n-1})$, with $c_0 = 1$ and $c_j = \sum_{i=0}^{j} \alpha^{j-i} r_{j-i} r_i \beta^i$ for $j = 1, \dots, n-1$ with coefficients $r_i = \left| \binom{-1/2}{i} \right|$. For any $p \in \{1, \dots, n\}$, the $p$-banded BSR matrix $C_{\alpha,\beta}^{|p|}$ is obtained from this by setting all coefficients $c_j = 0$ for $j \geq p$.*

*Proof sketch.* The proof be found in Appendix F.1. Its main idea is to factorize $A_{\alpha,\beta}$ into a product of two simpler lower triangular matrices, each of which has a closed-form square root. We show that the two roots commute and that the matrix $C_{\alpha,\beta}$ is their product, which implies the theorem. $\square$

### 3.2 Efficiency

We first establish that the $p$-BSR for SGD can be computed efficiently even for large problem sizes.

**Lemma 1** (Efficiency of BSR). *The entries of $C_{\alpha,\beta}^{|p|}$ can be determined in runtime $O(p \log p)$, i.e., in particular independent of $n$.*

*Proof sketch.* As a lower triangular Toeplitz matrix, $C_{\alpha,\beta}^{|p|}$ is fully determined by the values of its first column. By construction $c_{p+1}, \dots, c_n = 0$, so only the complexity of computing $c_1, \dots, c_{p-1}$ matters. These can be computed efficiently by writing them as the convolution of vectors $(\alpha^i r_i)_{i=0,\dots,p-1}$ and $(\beta^i r_i)_{i=0,\dots,p-1}$ and, e.g., employing the fast Fourier transform. $\square$

Note that for running Algorithm 1, the matrix $B$ of the factorization is not actually required. However, one needs to know the *sensitivity* of $C_{\alpha,\beta}^{|p|}$, as this determines the necessary amount of noise. The following theorem establishes that for a large class of matrices, including the BSR in the SGD setting, this is possible exactly and in closed form.

**Theorem 2** (Sensitivity for decreasing non-negative Toeplitz matrices). *Let $M = \mathrm{LDToep}(m_0, \dots, m_{n-1})$ be a lower triangular Toeplitz matrix with decreasing non-negative entries, i.e.*

$m_0 \geq m_1 \geq m_2 \geq \dots m_{n-1} \geq 0$. *Then its* sensitivity (2) *in the setting of $b$-min-separation is*

$$\mathrm{sens}_{k,b}(M) = \left\| \sum_{j=0}^{k-1} M_{[\cdot,1+jb]} \right\| = \left( \sum_{i=0}^{n-1} \left( \sum_{j=0}^{\min\{k-1,i/b\}} m_{i-jb} \right)^2 \right)^{1/2}, \quad (10)$$

*where $M_{[\cdot,1+jb]}$ denotes the $(1+jb)$-th column of $M$.*

*Proof sketch.* The proof can be found in Appendix F.2. It builds on the identity (2), which holds with equality because of the non-negative entries of $M$. Using the fact that the entries of $M$ are non-increasing one establishes that an optimal $b$-separated index set is $\{1, 1+b, \cdots, 1+(k-1)b\}$. From this, the identity (10) follows. $\square$

**Corollary 1.** *The sensitivity of the $p$-BSR for SGD can be computed using formula (10).*

*Proof sketch.* It suffices to show that the coefficients $c_0, \dots, c_{n-1}$ of Theorem 1 are monotonically decreasing. We do so by an explicit computation, see Appendix F.3. $\square$

## 3.3 Approximation Quality – Single Participation

Having established the efficiency of BSR, we now demonstrate its suitability for high-quality model training. To avoid corner cases, we assume that $\frac{n}{b}$ is an integer, which does not affect the asymptotic behavior. We also discuss only the case in which the update vectors have bounded norm $\zeta = 1$. Results for general $\zeta$ can readily be derived using the linearity of the sensitivity with respect to $\zeta$.

We first discuss the case of model training with single participation ($k = 1$), where more precise results are possible than the general case. Our main result are bounds on the expected approximation error of the square root factorization that, in particular, prove its asymptotic optimality.

**Theorem 3** (Expected approximation error with single participation). *Let $A_{\alpha,\beta} \in \mathbb{R}^{n \times n}$ be the workload matrix* (9) *of SGD with momentum $0 \leq \beta < 1$ and weight decay parameter $0 < \alpha \leq 1$, where $\alpha > \beta$. Assume that each data item can contribute at most once to an update vector (e.g. single participation, $k = 1$). Then, the expected approximation error of the* square root factorization, $A_{\alpha,\beta} = C_{\alpha,\beta}^2$, *fulfills*

$$1 \leq \mathcal{E}(C_{\alpha,\beta}, C_{\alpha,\beta}) \leq \frac{1}{(\alpha - \beta)^2} \log \frac{1}{1 - \alpha^2} \tag{11}$$

*for $\alpha < 1$, and*

$$\max\left\{1, \frac{\log(n+1) - 1}{4}\right\} \leq \mathcal{E}(C_{1,\beta}, C_{1,\beta}) \leq \frac{1 + \log(n)}{(1 - \beta)^2}. \tag{12}$$

*Proof sketch.* For the proof, we establish a relations between $\mathrm{sens}_{1,n}(C)$ and $\|C_{\alpha,\beta}\|_F$, and then we bound the resulting expressions by an explicit analysis of the norm. For details, see Appendix F.5. □

The following two results provide context for the interpretation of Theorem 3.

**Theorem 4.** *Assume the setting of Theorem 3. Then, for any factorization $A_{\alpha,\beta} = BC$ with $C^\top C \geq 0$, the expected approximation error fulfills*

$$\mathcal{E}(B, C) = \begin{cases} \Omega(1) & \text{for } \alpha < 1, \\ \Omega(\log n) & \text{for } \alpha = 1. \end{cases} \tag{13}$$

*Proof sketch.* The theorem is the special case $k = 1$ of Theorem 8, which we state in the next section and prove in Section F.9. □

**Theorem 5.** *Assume the setting of Theorem 3. Then, the baseline factorizations $A_{\alpha,\beta} = A_{\alpha,\beta} \cdot \mathrm{Id}$ and $A_{\alpha,\beta} = \mathrm{Id} \cdot A_{\alpha,\beta}$ fulfill, for $\alpha < 1$,*

$$\mathcal{E}(A_{\alpha,\beta}, \mathrm{Id}) = \frac{\sqrt{1 + \alpha\beta}}{\sqrt{(1 - \alpha\beta)(1 - \alpha^2)(1 - \beta^2)}} + o(1) \quad and \quad \mathcal{E}(A_{1,\beta}, \mathrm{Id}) \leq \frac{\sqrt{n}}{\sqrt{2}(1 - \beta)} + o(\sqrt{n}) \tag{14}$$

$$\mathcal{E}(\mathrm{Id}, A_{\alpha,\beta}) = \frac{\sqrt{1 + \alpha\beta}}{\sqrt{(1 - \alpha\beta)(1 - \alpha^2)(1 - \beta^2)}} + o(1) \quad and \quad \mathcal{E}(\mathrm{Id}, A_{1,\beta}) \leq \frac{\sqrt{n}}{1 - \beta} + o(\sqrt{n}). \tag{15}$$

*Proof sketch.* The result follows from an explicit analysis of the coefficients, see Appendix F.6. □

**Discussion.** Theorems 3 to 5 provide a full characterization of the approximation quality of the square root factorization as well as its alternatives: 1) the square root factorization has asymptotically optimal approximation quality, because the upper bounds in Equation (12) match the lower bounds in Equation (13); 2) the AOF from Definition 2 also fulfills the conditions of Theorem 4. Therefore, it must also adhere to the lower bound (13) and cannot be asymptotically better than the square root factorization; 3) the approximation qualities of the baseline factorizations in Equation (14) and (15) are asymptotically worse than optimal in the $\alpha = 1$ setting, and worse by a constant factor for $\alpha < 1$. The BSR factorization can be applied even in more general scenarios, such as with varying learning rates. However, in this case, the workload matrix will no longer be Toeplitz. This makes it difficult to provide analytical guarantees for the matrix, but it can still be applied numerically.

## 3.4 Approximation Quality – Repeated Participation.

We now provide mostly asymptotic statements about the approximation quality of BSR and baselines in the setting where data items can contribute more than once to the update vectors.

**Theorem 6** (Approximation error of BSR). *Let $A_{\alpha,\beta} \in \mathbb{R}^{n \times n}$ be the workload matrix (9) of SGD with momentum $0 \leq \beta < 1$ and weight decay $0 < \alpha \leq 1$, with $\alpha > \beta$. Let $A_{\alpha,\beta} = B_{\alpha,\beta}^{|p|} C_{\alpha,\beta}^{|p|}$, be its banded square root factorization as in Definition 3. Then, for any $b \in \{1, \ldots, n\}$, $p \leq b$, and $k \in \{1, \ldots, \frac{n}{b}\}$ it holds:*

$$
\mathcal{E}(B_{\alpha,\beta}^{|p|}, C_{\alpha,\beta}^{|p|}) = \begin{cases} O_\beta\left(\sqrt{\dfrac{nk \log p}{p}}\right) + O_{\beta,p}(\sqrt{k}) & \text{for } \alpha = 1, \\ O_{\beta,p,\alpha}\left(\sqrt{k}\right) & \text{for } \alpha < 1. \end{cases}
\tag{16}
$$

*Proof sketch.* For the proof, we separately bound the sensitivity of $C_{\alpha,\beta}^{|p|}$ and the Frobenius norm of $B_{\alpha,\beta}^{|p|}$. The former is straightforward because of the matrix's band structure. The latter requires an in-depth analysis of the inverse matrix' coefficient. Both steps are detailed in Appendix F.7. □

The following results provide context for the interpretation of Theorem 6.

**Theorem 7** (Approximation error of Square Root Factorization). *Let $A_{\alpha,\beta} \in \mathbb{R}^{n \times n}$ be the workload matrix (9) of SGD with momentum $0 \leq \beta < 1$ and weight decay $0 < \alpha \leq 1$, with $\alpha > \beta$. Let $A_{\alpha,\beta} = C_{\alpha,\beta}^2$ be its square root factorization. Then, for any $b \in \{1, \ldots, n\}$ and $k = \frac{n}{b}$ it holds:*

$$
\mathcal{E}(C_{\alpha,\beta}, C_{\alpha,\beta}) = \begin{cases} \Theta_\beta\left(k\sqrt{\log n} + \sqrt{k}\log n\right) & \text{for } \alpha = 1, \\ \Theta_{\alpha,\beta}(\sqrt{k}) & \text{for } \alpha < 1. \end{cases}
\tag{17}
$$

*Proof sketch.* We bound $\mathrm{sens}_{k,b}(C_{\alpha,\beta})$ and $\|C_{\alpha,\beta}\|_F$ using the explicit entries for $C_{\alpha,\beta}$ from Theorem 1. Details are provided in Appendix F.8. □

**Theorem 8.** *Assume the setting of Theorem 6. Then, for any factorization $A_{\alpha,\beta} = BC$ with $C^\top C \geq 0$, the approximation error fulfills*

$$
\mathcal{E}(B, C) \geq \begin{cases} \sqrt{k}\log n & \text{for } \alpha = 1, \\ \sqrt{k} & \text{for } \alpha < 1, \end{cases}
\tag{18}
$$

*Proof sketch.* The proof is based on the observation that $\|X\|_F\|Y\|_F \geq \|XY\|_*$ for any matrices $X, Y$, where $\|\cdot\|_*$ denotes the nuclear norm. To derive (18), we show that $\mathrm{sens}_{k,b}(C)$ is lower bounded by $\frac{\sqrt{k}}{n}\|C\|_F$, and derive explicit bounds on the singular values of $A_{\alpha,\beta}$. □

**Theorem 9.** *Assume the setting of Theorem 6. Then, the baseline factorizations $A_{\alpha,\beta} = A_{\alpha,\beta} \cdot \mathrm{Id}$ and $A_{\alpha,\beta} = \mathrm{Id} \cdot A_{\alpha,\beta}$ fulfill*

$$
\mathcal{E}(A_{\alpha,\beta}, \mathrm{Id}) \geq \begin{cases} \sqrt{\dfrac{nk}{2}} & \text{for } \alpha = 1, \\ \sqrt{k} & \text{for } \alpha < 1. \end{cases} \qquad \mathcal{E}(\mathrm{Id}, A_{\alpha,\beta}) \geq \begin{cases} \dfrac{k\sqrt{n}}{\sqrt{3}} & \text{for } \alpha = 1, \\ \sqrt{k} & \text{for } \alpha < 1. \end{cases}
\tag{19}
$$

*Proof sketch.* The proof relies on the fact that the workload matrices can be lower bounded componentwise by simpler matrices: $A_{\alpha,\beta} \geq A_{\alpha,0}$ and $A_{\alpha,0} \geq \mathrm{Id}$. For the simpler matrices, the bounds (19) can then be derived analytically, and the general case follows by monotonicity. □

**Discussion.**     Analogously to the case of single participation, Theorems 6 to 9 again establish that the proposed BSR is asymptotically superior to the baseline factorizations if $\alpha = 1$. A comparison of Theorems 6 and 7 suggests that, at least for maximal participation, $k = \frac{n}{b}$ and $p = b$, the bandedness of the $p$-BSR improves the approximation quality, specifically in the practically relevant regime where $b \ll n$. While none of the methods match the lower bound of Theorem 6, we conjecture that this is not because any asymptotically better methods would exist, but rather a sign of Equation (18) is not tight. Both theoretical consideration and experiments suggest that a term linear in $k$ should appear there. For $\alpha < 1$, all studied methods are asymptotically identical and, in fact, optimal.

## 4   Experiments

To demonstrate that BSR can achieve high accuracy not only in theory but also in practice, we compare it to AOF and baselines in numerical experiments. **Our results show that BSR achieves quality comparable to the AOF, but without the computational overhead, and it clearly outperforms the baseline factorizations.** The privacy guarantees are identical for all methods, so we do not discuss them explicitly.

**Implementation and computational cost.**     We implement BSR by the closed-form expressions of Theorem (1). For single data participation, we use the square root decomposition directly. For repeated data participation we use $p$-BSR with $p = b$. Using standard *python/numpy* code, computing the BSR as dense matrices are memory-bound rather than compute-bound. Even sizes of $n = 10,000$ or more take at most a few seconds. Computing only the Toeplitz coefficients is even faster, of course.

To compute AOF, we solve the optimization problem (4) using the `cvxpy` package with `SCS` backend, see Algorithm B for the source code[3]. With the default numerical tolerance, $10^{-4}$, each factorization took a few minutes ($n \leq 100$) to hours ($n \leq 500$) to several days ($n \geq 700$) of CPU time. Note that this overhead reappears for any change in the number of update steps, $n$, weight decay, $\alpha$, or momentum, $\beta$, as these induce different workload matrices. In our experiments, when the optimization for AOF did not terminate within 10 days, we reran the optimization problem with the tolerance increased by a factor of 10. The runtime depends not only on the matrix size but also on the entries. In particular, we observe matrices with momentum to be harder to factorize than without. For large matrix sizes we frequently encountered numerical problems: the intermediate matrices, $S$, in (4), often did not fulfill the positive definiteness condition required to solve the subsequent Cholesky decomposition for $C$. Unfortunately, simply projecting the intermediates back to the cone of positive semi-definite matrices is not enough, because the resulting $C$ matrices also have to be invertible and not too badly conditioned. Ultimately, we adopted a postprocessing step for $S$ that ensures that all its eigenvalues were at least of value $\sqrt{1/n}$, which we find to be a reasonable modification to ensure the stability of the convergence. Enforcing this empirically found value leads to generally good results, as our experiments below show, but it does add an undesirable extra level of complexity to the process. In contrast, due to its analytic expressions, BSR does not suffer from numerical problems. It also does not possess additional hyperparameters, such as a numeric tolerance or the number of optimization steps.

Apart from the factorization itself, the computational cost of BSR and AOF are nearly identical. Both methods produce (banded) lower triangular matrices, so computing the inverse matrices or solving linear systems can be done within milliseconds to seconds using forward substitution. Note that, in principle, one could even exploit the Toeplitz structure of $p$-BSR, but we found this not to yield any practical benefit in our experiments. Computing the sensitivity is trivial for $p$-BSR using Corollary 1, and it is still efficient for AOF by the dynamic program proposed in Choquette-Choo et al. [2023a].

**Expected Approximation Error.**     As a first numeric experiment, we evaluate the expected approximation error for workload matrices that reflect different SGD settings. Specifically, we use workload matrices (9) for $n \in \{100, 200, \ldots, 1000, 1500, 2000\}$, with $\alpha = \{0.99, 0.999, 0.9999, 1\}$, and $\beta \in \{0, 0.9\}$, either with single participation, $k = 1$, or repeated participation, $b = 100$, $k = n/100$. Figure 1 shows the expected approximate error, $\mathcal{E}(B, C)$, of the proposed BSR, AOF, as well as the baseline factorizations, $A = A \cdot \text{Id}$ and $A = \text{Id} \cdot A$ in two exemplary cases. Additional results for other privacy levels can be found in Appendix C.

---

[3]Additional experiments with gradient-based optimizers can be found in Appendix E.

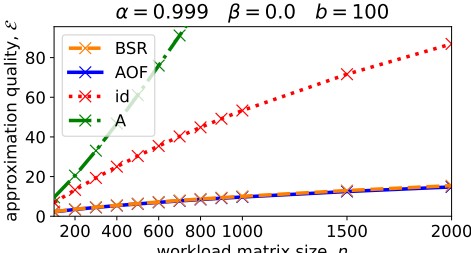 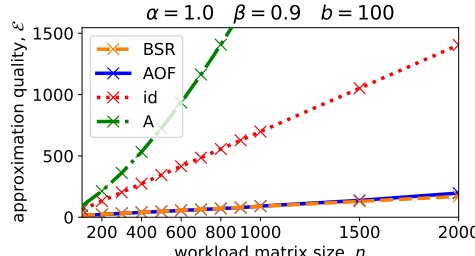

Figure 1: Expected approximation error of BSR, AOF and baseline factorizations for two different hyperparameter settings (left: $\alpha = 0.999, \beta = 0$, right: $\alpha = 1, \beta = 0.9$) with repeated participation ($b = 100, k = n/100$). See Section 4 for details.

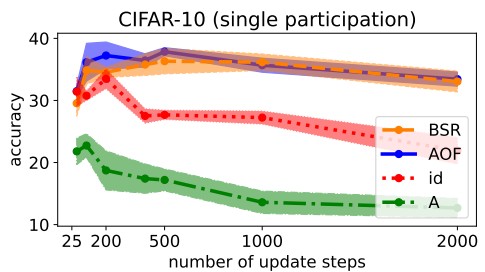 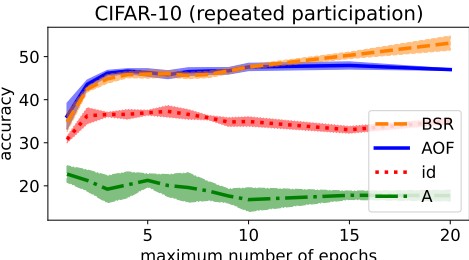

Figure 2: Classification accuracy (mean and standard deviation over 5 runs with different random seeds) on CIFAR-10 for BSR, AOF, and baselines for $(\epsilon, \delta) = (4, 10^{-5})$ for independent training runs. Left: one epoch, different batch sizes. Right: different number of epochs, constant batch size.

The results confirm our expectations from the theoretical analysis: in particular, BSR's expected approximation error is quite close to AOF's, typically within a few percent (left plot). Both methods are clearly superior to the naive factorizations. For large matrix sizes, BSR sometimes even yields slightly better values than AOF (right plot). However, we believe this to be a numeric artifact of us having to solve AOF with less-than-perfect precision.

**Private Model Training on CIFAR-10.** To demonstrate the usefulness of BSR in practical settings, we follow the setup of Kairouz et al. [2021] and report results for training a simple ConvNet on the CIFAR-10 dataset (see Table 1 in Appendix C for the architecture). Specifically, we adapt Google's reference implementation of DP-SGD in `jax` Bradbury et al. [2018] to work with the different matrix factorizations: BSR, AOF, and the two baselines. To reflect the setting of single-participation training, we split the 50,000 training examples into batches of size $m \in \{1000, 500, 250, 200, 100, 50, 25\}$, resulting in $n \in \{100, 200, 400, 500, 1000, 2000\}$ update steps. For repeated participation, we fix the batch size to 500 and run $k \in \{1, 2, \ldots, 10, 15, 20\}$ epoch of training, i.e. $n = 100k$ and $b = 100$. In both cases, 20% of the training examples are used as validation sets to determine the learning rate $\eta \in \{0.01, 0.05, 0.1, 0.5, 1\}$, weight decay parameters $\alpha \in \{0.99, 0.999, 0.9999, 1\}$, and momentum $\beta \in \{0, 0.9\}$. Figure 2 shows the test set accuracy of the model trained with hyperparameters that achieved the highest validation accuracy.[4] One can see the expected effect that in DP model training, more update steps/epochs do not necessary lead to higher accuracy due to the need to add more noise. The quality of models trained with BSR is mostly identical to AOF. When training for a large number of epochs it achieves even better slightly results, but this could also be an artifact of us having to solve AOF with reduced precision in this regime. Both methods are clearly superior to the baselines.

---

[4]Such a setting would not optimal for real-world private training, because the many repeated experiments reduce the privacy guarantees [Papernot and Steinke, 2021, Kurakin et al., 2022, Ponomareva et al., 2023]. We nevertheless adopt it here to allow for a simpler and fair comparison between methods.

# 5 Conclusion and Discussion

We introduce an efficient and effective approach to the matrix factorization mechanism for SGD-based model training with differential privacy. The proposed banded square root factorization (BSR) factorization achieves results on par with the previous state-of-the-art, and clearly superior to baseline methods. At the same time, it does not suffer from the previous method's computational overhead, thereby making differentially private model training practical even for large scale problems.

Despite the promising results, some open questions remain. On the theoretical side, the asymptotic optimality of BSR without weight decay is still unresolved because the current upper bounds on the expected approximation error do not match the provided lower bounds. Based on the experimental results, we believe this discrepancy lies with the lower bounds, which we suspect should be linear in the number of participations. We observe that BSR achieves results comparable to AOF, although we cannot currently prove this due to the insufficient understanding of AOF's theoretical properties; nonetheless, we consider it a promising research direction. On the practical side, it would be interesting to extend the guarantees to even more learning scenarios, such as variable learning rates.

## Acknowledgments and Disclosure of Funding

This research was supported by the Scientific Service Units (SSU) of ISTA through resources provided by Scientific Computing (SciComp). We thank Monika Henzinger and Jalaj Upadhyay for their valuable comments on the earlier versions of this manuscript.

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

# Appendix

## A    General introduction to differential privacy.

Differential privacy [Dwork, 2006] is a robust framework designed to provide strong privacy guarantees for statistical analyses and data sharing. It aims to protect individual data points in a dataset while still allowing meaningful aggregate information to be extracted. Unlike traditional data anonymization techniques, which might involve removing identifiers or aggregating data, differential privacy offers a mathematical definition of privacy that quantifies the amount of privacy loss and ensures that the risk of identifying any individual's data remains low, even when combined with other data sources.To formalize this concept, a randomized mechanism $M$ is said to provide $(\varepsilon, \delta)$-differential privacy if, for all data sets $D$ and $D'$ that differ in one element, and for all subsets of the mechanism's output space $S$:

$$\Pr[M(D) \in S] \leq e^{\varepsilon} \cdot \Pr[M(D') \in S] + \delta$$

For a detailed introduction to differential privacy, we recommend the books "The Algorithmic Foundations of Differential Privacy" by Dwork and Roth [2014] and "The Complexity of Differential Privacy" by Vadhan [2017].

## B    Source code for computing AOF

---

**Algorithm 2** Source code for computing AOF using `cvxpy`.

---

```python
import cvxpy as cp
import numpy as np

def banded_factorization(A, b):
    n = len(A)
    X = cp.Variable((n, n), PSD=True)

    # cp.matrix_frac(A, X) = tr(A.T @ X^-1 @ A)
    objective = cp.Minimize(cp.matrix_frac(A.T, X) * np.ceil(n / b))
    constraints = [cp.diag(X) == 1]

    for i in range(b, n):
        constraints += [cp.diag(X, i) == 0, cp.diag(X, -i) == 0]

    prob = cp.Problem(objective, constraints)
    return prob.solve(solver='SCS'), X.value
```

---

## C    Network architecture for CIFAR-10 experiments

Table 1: ConvNet architecture for CIFAR-10 experiments

| |
|---|
| Conv2D(channels=32, kernel=(3, 3), strides=(1, 1), padding='SAME', activation='relu') |
| Conv2D(channels=32, kernel=(3, 3), strides=(1, 1), padding='SAME', activation='relu') |
| MaxPool(kernel=(2, 2), strides=(2, 2)) |
| Conv2D(channels=64, kernel=(3, 3), strides=(1, 1), padding='SAME'), activation='relu') |
| Conv2D(channels=64, kernel=(3, 3), strides=(1, 1), padding='SAME'), activation='relu') |
| MaxPool(kernel=(2, 2), strides=(2, 2)) |
| Conv2D(channels=128, kernel=(3, 3), strides=(1, 1), padding='SAME'), activation='relu') |
| Conv2D(channels=128, kernel=(3, 3), strides=(1, 1), padding='SAME'), activation='relu') |
| MaxPool(kernel=(2, 2), strides=(2, 2)) |
| Flatten() |
| Dense(outputs=10) |

# D  Additional Experimental Results

In this section we provide additional experiments comparing BSR, AOF and baselines: Figures 3 and 4 and following tables show their *expected approximation error* (lower is better) for workload matrices stemming from SGD with different hyperparameter settings. Figure 5) and following tables show the accuracy of resulting classifiers on CIFAR-10 (higher is better) for different privacy levels.

The results show the same trends as the one in Section 4. BSR achieves almost identical expected approximation error as AOF, and results in equally good classifiers. In some cases, results for BSR even improve over AOF's. Presumably this is because of numerical issues in solving the optimization problem for AOF.

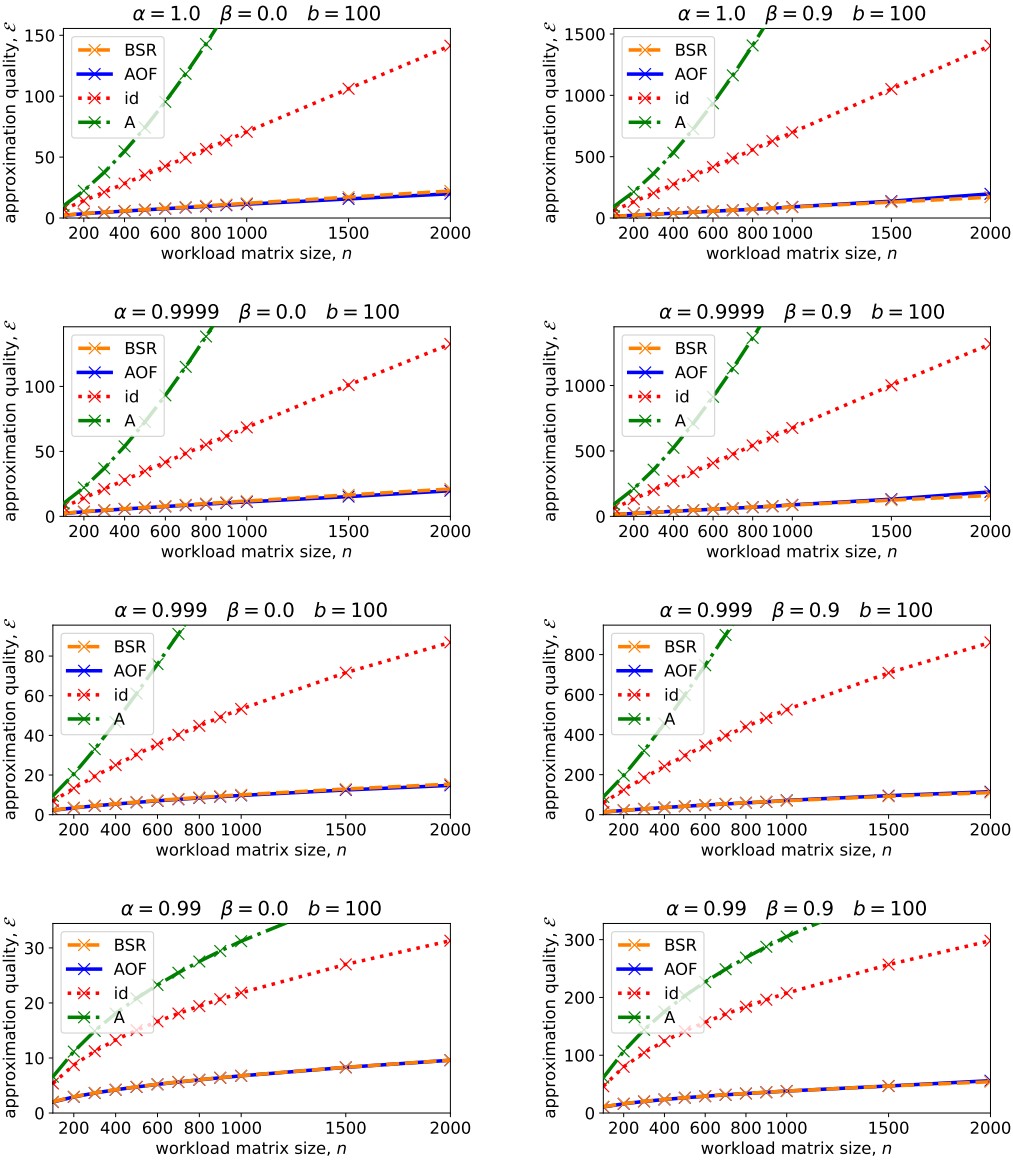

Figure 3: Expected approximation error of $p$-BSR, AOF and baseline factorizations with lmultiple participations and $p = b = 100$.

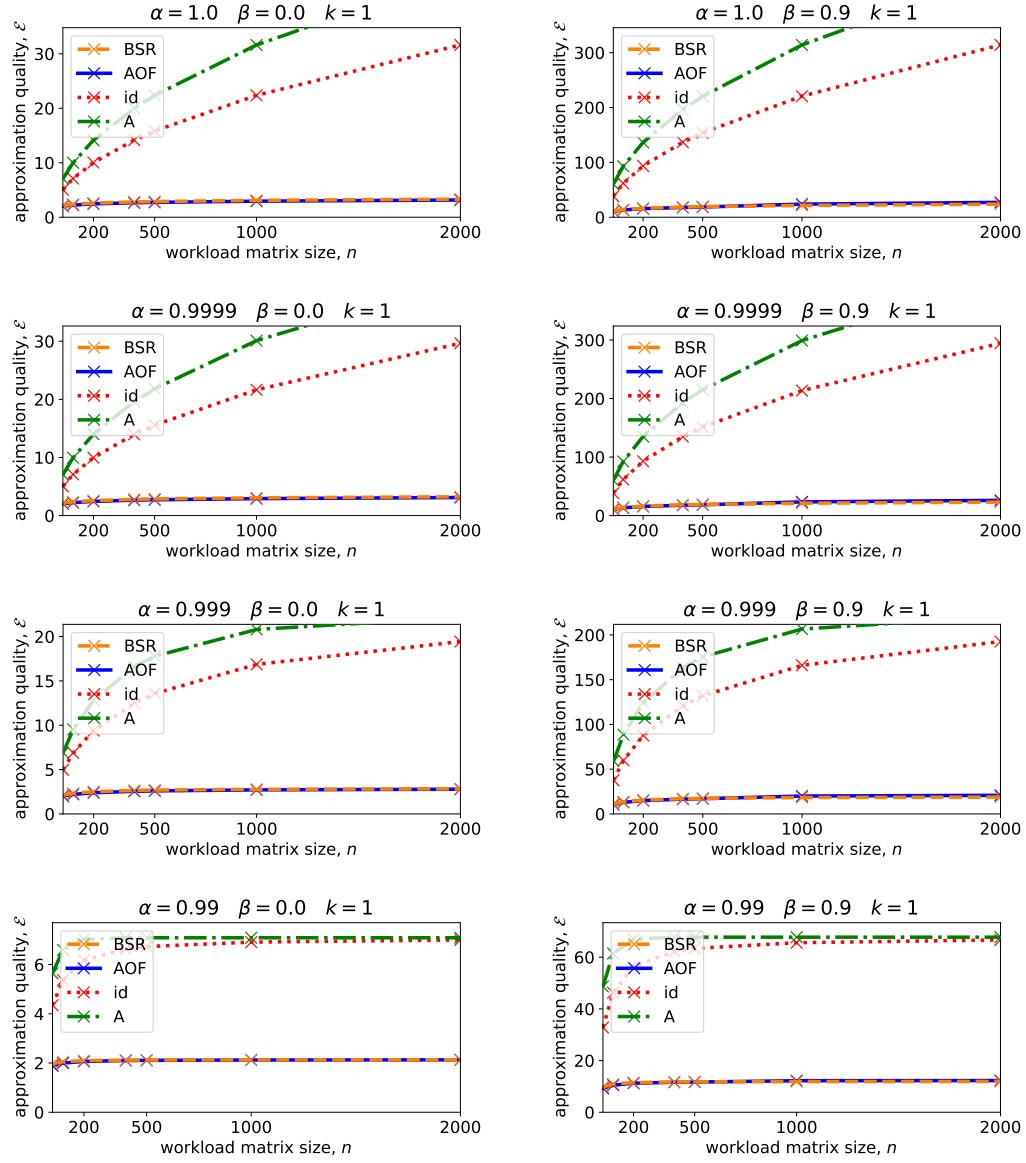

Figure 4: Expected approximation error of BSR, AOF and baseline factorizations with single participation ($k = 1$, $p = b = n$).

| | expected factorization error | | | | |
|---|---|---|---|---|---|
| $n$ | BSR | sqrt | AOF | Id | $A$ |
| 100 | 2.4 | 2.4 | 2.2 | 7.1 | 10.0 |
| 200 | 3.6 | 4.0 | 3.5 | 14.2 | 22.4 |
| 300 | 4.8 | 5.5 | 4.6 | 21.2 | 37.4 |
| 400 | 5.9 | 6.9 | 5.7 | 28.3 | 54.8 |
| 500 | 6.9 | 8.4 | 6.7 | 35.4 | 74.2 |
| 600 | 8.0 | 9.9 | 7.7 | 42.5 | 95.4 |
| 700 | 9.0 | 11.3 | 8.6 | 49.5 | 118.3 |
| 800 | 10.1 | 12.8 | 9.5 | 56.6 | 142.8 |
| 900 | 11.1 | 14.2 | 10.4 | 63.7 | 168.8 |
| 1000 | 12.1 | 15.7 | 11.3 | 70.7 | 196.2 |
| 1500 | 17.2 | 23.1 | 15.7 | 106.1 | 352.1 |
| 2000 | 22.2 | 30.6 | 19.9 | 141.5 | 535.7 |

Table 2: Numeric results for Figure 3 as well as a plain square root decomposition: $\alpha = 1$, $\beta = 0$, $k = 1$, $b = k/n$

| | expected factorization error | | | | |
|---|---|---|---|---|---|
| $n$ | BSR | sqrt | AOF | Id | $A$ |
| 100 | 13.9 | 13.9 | 13.3 | 61.8 | 92.9 |
| 200 | 22.9 | 25.7 | 22.4 | 132.3 | 213.2 |
| 300 | 31.4 | 37.4 | 31.0 | 202.9 | 361.2 |
| 400 | 39.7 | 49.1 | 39.3 | 273.6 | 532.6 |
| 500 | 48.0 | 61.0 | 47.4 | 344.3 | 724.7 |
| 600 | 56.2 | 73.0 | 55.4 | 415.0 | 935.3 |
| 700 | 64.3 | 85.1 | 63.2 | 485.7 | 1163.0 |
| 800 | 72.5 | 97.2 | 71.1 | 556.4 | 1406.6 |
| 900 | 80.6 | 109.5 | 78.8 | 627.1 | 1665.2 |
| 1000 | 88.7 | 121.8 | 90.6 | 697.8 | 1937.9 |
| 1500 | 129.2 | 184.2 | 137.2 | 1051.3 | 3491.5 |
| 2000 | 169.7 | 247.8 | 196.9 | 1404.9 | 5322.6 |

Table 3: Numeric results for Figure 3 as well as a plain square root decomposition: $\alpha = 1$, $\beta = 0.9$, $k = 1$, $b = k/n$

| | expected factorization error | | | | |
|---|---|---|---|---|---|
| $n$ | BSR | sqrt | AOF | Id | $A$ |
| 100 | 2.4 | 2.4 | 2.2 | 7.1 | 10.0 |
| 200 | 3.6 | 3.9 | 3.5 | 14.1 | 22.2 |
| 300 | 4.8 | 5.4 | 4.6 | 21.0 | 36.9 |
| 400 | 5.8 | 6.9 | 5.7 | 27.9 | 53.9 |
| 500 | 6.9 | 8.3 | 6.6 | 34.8 | 72.7 |
| 600 | 7.9 | 9.7 | 7.6 | 41.6 | 93.1 |
| 700 | 8.9 | 11.1 | 8.5 | 48.4 | 115.1 |
| 800 | 9.9 | 12.5 | 9.4 | 55.1 | 138.4 |
| 900 | 10.9 | 13.9 | 10.3 | 61.8 | 163.0 |
| 1000 | 11.8 | 15.3 | 11.1 | 68.5 | 188.7 |
| 1500 | 16.5 | 22.3 | 15.2 | 101.1 | 332.6 |
| 2000 | 21.0 | 29.1 | 19.7 | 132.6 | 496.8 |

Table 4: Numeric results for Figure 3 as well as a plain square root decomposition: $\alpha = 0.9999$, $\beta = 0$, $k = 1$, $b = k/n$

| | | expected factorization error | | | |
|---|---|---|---|---|---|
| $n$ | BSR | sqrt | AOF | Id | $A$ |
| 100 | 13.9 | 13.9 | 13.2 | 61.6 | 92.4 |
| 200 | 22.8 | 25.5 | 22.3 | 131.4 | 211.4 |
| 300 | 31.2 | 37.0 | 30.8 | 200.9 | 356.7 |
| 400 | 39.3 | 48.6 | 38.9 | 270.0 | 524.0 |
| 500 | 47.3 | 60.1 | 46.8 | 338.6 | 710.3 |
| 600 | 55.3 | 71.7 | 54.5 | 406.8 | 913.3 |
| 700 | 63.1 | 83.3 | 62.1 | 474.6 | 1131.5 |
| 800 | 70.9 | 95.0 | 69.6 | 541.9 | 1363.5 |
| 900 | 78.6 | 106.6 | 77.0 | 608.8 | 1608.1 |
| 1000 | 86.2 | 118.3 | 88.1 | 675.3 | 1864.6 |
| 1500 | 123.7 | 176.5 | 131.0 | 1001.3 | 3298.4 |
| 2000 | 159.9 | 234.4 | 187.0 | 1317.1 | 4937.9 |

Table 5: Numeric results for Figure 3 as well as a plain square root decomposition: $\alpha = 0.9999$, $\beta = 0.9$, $k = 1$, $b = k/n$

| | | expected factorization error | | | |
|---|---|---|---|---|---|
| $n$ | BSR | sqrt | AOF | Id | $A$ |
| 100 | 2.3 | 2.3 | 2.2 | 6.9 | 9.5 |
| 200 | 3.5 | 3.8 | 3.4 | 13.3 | 20.5 |
| 300 | 4.6 | 5.1 | 4.4 | 19.3 | 33.1 |
| 400 | 5.5 | 6.4 | 5.4 | 25.0 | 46.7 |
| 500 | 6.4 | 7.6 | 6.2 | 30.4 | 61.1 |
| 600 | 7.2 | 8.7 | 7.0 | 35.4 | 76.0 |
| 700 | 8.0 | 9.8 | 7.7 | 40.2 | 91.2 |
| 800 | 8.7 | 10.8 | 8.4 | 44.8 | 106.5 |
| 900 | 9.4 | 11.8 | 9.1 | 49.2 | 122.0 |
| 1000 | 10.0 | 12.8 | 9.7 | 53.3 | 137.4 |
| 1500 | 13.0 | 17.2 | 12.5 | 71.6 | 212.7 |
| 2000 | 15.4 | 21.1 | 14.8 | 86.9 | 282.9 |

Table 6: Numeric results for Figure 3 as well as a plain square root decomposition: $\alpha = 0.999$, $\beta = 0$, $k = 1$, $b = k/n$

| | | expected factorization error | | | |
|---|---|---|---|---|---|
| $n$ | BSR | sqrt | AOF | Id | $A$ |
| 100 | 13.5 | 13.5 | 12.9 | 59.8 | 88.6 |
| 200 | 21.8 | 24.2 | 21.4 | 124.0 | 195.8 |
| 300 | 29.2 | 34.3 | 28.9 | 184.5 | 319.9 |
| 400 | 36.1 | 44.0 | 35.8 | 241.4 | 455.3 |
| 500 | 42.5 | 53.3 | 42.2 | 295.2 | 598.5 |
| 600 | 48.6 | 62.3 | 48.1 | 346.0 | 746.7 |
| 700 | 54.3 | 70.9 | 53.7 | 394.2 | 898.2 |
| 800 | 59.8 | 79.2 | 59.0 | 440.0 | 1051.7 |
| 900 | 65.0 | 87.2 | 64.1 | 483.6 | 1205.9 |
| 1000 | 70.0 | 95.0 | 71.8 | 525.1 | 1360.2 |
| 1500 | 91.9 | 130.3 | 95.1 | 708.4 | 2113.9 |
| 2000 | 110.3 | 161.0 | 115.0 | 861.2 | 2816.2 |

Table 7: Numeric results for Figure 3 as well as a plain square root decomposition: $\alpha = 0.999$, $\beta = 0.9$, $k = 1$, $b = k/n$

|       | expected factorization error | | | | |
| $n$ | BSR | sqrt | AOF | Id | $A$ |
|-------|-------|-------|-------|-------|-------|
| 100   | 2.1   | 2.1   | 2.0   | 5.4   | 6.6   |
| 200   | 3.0   | 3.1   | 2.9   | 8.7   | 11.2  |
| 300   | 3.7   | 3.8   | 3.6   | 11.2  | 14.9  |
| 400   | 4.3   | 4.5   | 4.2   | 13.3  | 18.1  |
| 500   | 4.8   | 5.0   | 4.7   | 15.1  | 20.8  |
| 600   | 5.2   | 5.5   | 5.2   | 16.6  | 23.3  |
| 700   | 5.7   | 6.0   | 5.6   | 18.1  | 25.5  |
| 800   | 6.1   | 6.4   | 6.0   | 19.4  | 27.5  |
| 900   | 6.4   | 6.8   | 6.4   | 20.7  | 29.4  |
| 1000  | 6.8   | 7.2   | 6.8   | 21.9  | 31.2  |
| 1500  | 8.3   | 8.9   | 8.3   | 27.0  | 38.9  |
| 2000  | 9.6   | 10.3  | 9.6   | 31.3  | 45.4  |

Table 8: Numeric results for Figure 3 as well as a plain square root decomposition: $\alpha = 0.99$, $\beta = 0$, $k = 1$, $b = k/n$

|       | expected factorization error | | | | |
| $n$ | BSR | sqrt | AOF | Id | $A$ |
|-------|-------|-------|-------|-------|-------|
| 100   | 10.9  | 10.9  | 10.5  | 46.2  | 61.4  |
| 200   | 16.2  | 17.0  | 15.9  | 79.9  | 106.7 |
| 300   | 20.3  | 21.7  | 20.0  | 104.5 | 144.0 |
| 400   | 23.6  | 25.7  | 23.4  | 124.5 | 175.4 |
| 500   | 26.6  | 29.1  | 26.4  | 141.7 | 202.7 |
| 600   | 29.2  | 32.1  | 29.1  | 157.1 | 226.9 |
| 700   | 31.7  | 34.9  | 31.5  | 171.1 | 248.8 |
| 800   | 33.9  | 37.5  | 33.8  | 184.0 | 268.9 |
| 900   | 36.0  | 40.0  | 35.9  | 196.1 | 287.7 |
| 1000  | 38.0  | 42.3  | 38.1  | 207.4 | 305.3 |
| 1500  | 46.7  | 52.2  | 46.8  | 256.9 | 381.4 |
| 2000  | 54.1  | 60.6  | 56.0  | 298.2 | 444.6 |

Table 9: Numeric results for Figure 3 as well as a plain square root decomposition: $\alpha = 0.99$, $\beta = 0.9$, $k = 1$, $b = k/n$

|       | expected factorization error | | | | |
| $n$ | BSR | sqrt | AOF | Id | $A$ |
|-------|-------|-------|-------|-------|-------|
| 50    | 2.2   | 2.2   | 2.0   | 5.0   | 7.1   |
| 100   | 2.4   | 2.4   | 2.2   | 7.1   | 10.0  |
| 200   | 2.6   | 2.6   | 2.4   | 10.0  | 14.1  |
| 400   | 2.8   | 2.8   | 2.7   | 14.2  | 20.0  |
| 500   | 2.9   | 2.9   | 2.7   | 15.8  | 22.4  |
| 1000  | 3.1   | 3.1   | 2.9   | 22.4  | 31.6  |
| 2000  | 3.3   | 3.3   | 3.2   | 31.6  | 44.7  |

Table 10: Numeric results for Figure 4 as well as a plain square root decomposition: $\alpha = 1$, $\beta = 0$, $b = 0$

| | expected factorization error | | | | |
|---|---|---|---|---|---|
| $n$ | BSR | sqrt | AOF | Id | $A$ |
| 50 | 11.4 | 11.4 | 10.6 | 38.2 | 60.3 |
| 100 | 13.9 | 13.9 | 13.3 | 61.8 | 92.9 |
| 200 | 16.3 | 16.3 | 15.7 | 93.5 | 136.5 |
| 400 | 18.6 | 18.6 | 17.8 | 136.8 | 196.5 |
| 500 | 19.3 | 19.3 | 18.6 | 154.0 | 220.5 |
| 1000 | 21.6 | 21.6 | 23.9 | 220.7 | 314.0 |
| 2000 | 23.8 | 23.8 | 27.1 | 314.1 | 445.7 |

Table 11: Numeric results for Figure 4 as well as a plain square root decomposition: $\alpha = 1$, $\beta = 0.9$, $b = 0$

| | expected factorization error | | | | |
|---|---|---|---|---|---|
| $n$ | BSR | sqrt | AOF | Id | $A$ |
| 50 | 2.1 | 2.1 | 2.0 | 5.0 | 7.1 |
| 100 | 2.4 | 2.4 | 2.2 | 7.1 | 10.0 |
| 200 | 2.6 | 2.6 | 2.4 | 10.0 | 14.0 |
| 400 | 2.8 | 2.8 | 2.7 | 14.0 | 19.6 |
| 500 | 2.9 | 2.9 | 2.7 | 15.6 | 21.8 |
| 1000 | 3.1 | 3.1 | 2.9 | 21.7 | 30.1 |
| 2000 | 3.2 | 3.2 | 3.1 | 29.7 | 40.6 |

Table 12: Numeric results for Figure 4 as well as a plain square root decomposition: $\alpha = 0.9999$, $\beta = 0$, $b = 0$

| | expected factorization error | | | | |
|---|---|---|---|---|---|
| $n$ | BSR | sqrt | AOF | Id | $A$ |
| 50 | 11.4 | 11.4 | 10.6 | 38.2 | 60.2 |
| 100 | 13.9 | 13.9 | 13.2 | 61.6 | 92.4 |
| 200 | 16.2 | 16.2 | 15.6 | 92.9 | 135.2 |
| 400 | 18.4 | 18.4 | 17.7 | 135.0 | 192.7 |
| 500 | 19.1 | 19.1 | 18.4 | 151.4 | 215.2 |
| 1000 | 21.1 | 21.1 | 23.4 | 213.5 | 299.1 |
| 2000 | 23.0 | 23.0 | 26.0 | 294.5 | 404.7 |

Table 13: Numeric results for Figure 4 as well as a plain square root decomposition: $\alpha = 0.9999$, $\beta = 0.9$, $b = 0$

| | expected factorization error | | | | |
|---|---|---|---|---|---|
| $n$ | BSR | sqrt | AOF | Id | $A$ |
| 50 | 2.1 | 2.1 | 2.0 | 5.0 | 6.9 |
| 100 | 2.3 | 2.3 | 2.2 | 6.9 | 9.5 |
| 200 | 2.5 | 2.5 | 2.4 | 9.4 | 12.8 |
| 400 | 2.7 | 2.7 | 2.6 | 12.5 | 16.6 |
| 500 | 2.7 | 2.7 | 2.6 | 13.6 | 17.8 |
| 1000 | 2.8 | 2.8 | 2.7 | 16.9 | 20.8 |
| 2000 | 2.8 | 2.8 | 2.8 | 19.4 | 22.2 |

Table 14: Numeric results for Figure 4 as well as a plain square root decomposition: $\alpha = 0.999$, $\beta = 0$, $b = 0$

| $n$ | expected factorization error | | | | |
| | BSR | sqrt | AOF | Id | $A$ |
|---|---|---|---|---|---|
| 50 | 11.2 | 11.2 | 10.4 | 37.6 | 59.0 |
| 100 | 13.5 | 13.5 | 12.9 | 59.8 | 88.6 |
| 200 | 15.5 | 15.5 | 14.9 | 87.7 | 124.2 |
| 400 | 17.0 | 17.0 | 16.5 | 120.7 | 163.3 |
| 500 | 17.4 | 17.4 | 17.0 | 132.0 | 175.6 |
| 1000 | 18.4 | 18.4 | 20.0 | 166.1 | 206.6 |
| 2000 | 18.8 | 18.8 | 20.8 | 192.6 | 220.5 |

Table 15: Numeric results for Figure 4 as well as a plain square root decomposition: $\alpha = 0.999$, $\beta = 0.9$, $b = 0$

| $n$ | expected factorization error | | | | |
| | BSR | sqrt | AOF | Id | $A$ |
|---|---|---|---|---|---|
| 50 | 2.0 | 2.0 | 1.9 | 4.3 | 5.6 |
| 100 | 2.1 | 2.1 | 2.0 | 5.4 | 6.6 |
| 200 | 2.1 | 2.1 | 2.1 | 6.2 | 7.0 |
| 400 | 2.1 | 2.1 | 2.1 | 6.6 | 7.1 |
| 500 | 2.1 | 2.1 | 2.1 | 6.7 | 7.1 |
| 1000 | 2.1 | 2.1 | 2.1 | 6.9 | 7.1 |
| 2000 | 2.1 | 2.1 | 2.1 | 7.0 | 7.1 |

Table 16: Numeric results for Figure 4 as well as a plain square root decomposition: $\alpha = 0.99$, $\beta = 0$, $b = 0$

| $n$ | expected factorization error | | | | |
| | BSR | sqrt | AOF | Id | $A$ |
|---|---|---|---|---|---|
| 50 | 9.8 | 9.8 | 9.2 | 32.8 | 48.7 |
| 100 | 10.9 | 10.9 | 10.5 | 46.2 | 61.4 |
| 200 | 11.5 | 11.5 | 11.2 | 56.5 | 66.9 |
| 400 | 11.7 | 11.7 | 11.6 | 62.3 | 67.7 |
| 500 | 11.8 | 11.8 | 11.7 | 63.4 | 67.7 |
| 1000 | 11.9 | 11.9 | 12.2 | 65.6 | 67.7 |
| 2000 | 11.9 | 11.9 | 12.3 | 66.7 | 67.7 |

Table 17: Numeric results for Figure 4 as well as a plain square root decomposition: $\alpha = 0.99$, $\beta = 0.9$, $b = 0$

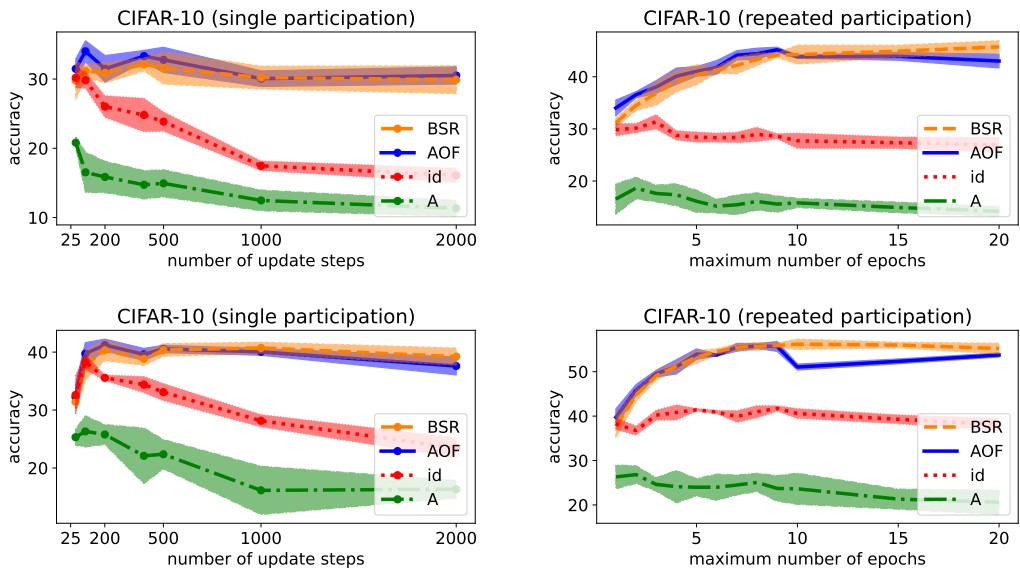

Figure 5: Classification accuracy (mean and standard deviation over 5 runs with different random seeds) on CIFAR-10 for BSR, AOF, and baselines for independent training runs. Top row: classification accuracy on CIFAR-10 with $(\epsilon, \delta) = (2, 10^{-5})$. Bottom row: classification accuracy on CIFAR-10 with $(\epsilon, \delta) = (8, 10^{-5})$. Left plots: one epoch, different batch sizes. Right plots: different number of epochs, constant batch size.

| | accuracy | | | |
|---|---|---|---|---|
| number of updates | BSR | AOF | Id | $A$ |
| 50 | $29.5 \pm 2.1$ | $31.5 \pm 1.0$ | $31.4 \pm 2.3$ | $21.8 \pm 2.1$ |
| 100 | $34.8 \pm 2.2$ | $36.2 \pm 3.0$ | $30.8 \pm 0.8$ | $22.7 \pm 1.9$ |
| 200 | $34.5 \pm 0.8$ | $37.2 \pm 2.2$ | $33.5 \pm 1.6$ | $18.7 \pm 3.1$ |
| 400 | $35.8 \pm 1.7$ | $36.4 \pm 1.1$ | $27.5 \pm 1.1$ | $17.4 \pm 2.4$ |
| 500 | $36.3 \pm 2.0$ | $37.9 \pm 0.7$ | $27.7 \pm 0.7$ | $17.2 \pm 1.8$ |
| 1000 | $36.2 \pm 1.3$ | $35.7 \pm 1.1$ | $27.2 \pm 1.0$ | $13.6 \pm 1.8$ |
| 2000 | $33.0 \pm 1.7$ | $33.4 \pm 1.1$ | $21.9 \pm 2.1$ | $12.7 \pm 1.5$ |

Table 18: Numeric values for results in Table 2 left plot (CIFAR-10, single participation, $(\epsilon, \delta) = (4, 10^{-5})$).

| number of epochs | accuracy | | | |
|---|---|---|---|---|
| | BSR | AOF | Id | $A$ |
| 1 | $34.8 \pm 2.2$ | $36.2 \pm 3.0$ | $30.8 \pm 0.8$ | $22.7 \pm 1.9$ |
| 2 | $42.4 \pm 0.6$ | $43.6 \pm 1.0$ | $36.3 \pm 1.9$ | $21.3 \pm 2.4$ |
| 3 | $44.8 \pm 1.0$ | $46.1 \pm 0.7$ | $36.6 \pm 0.5$ | $19.2 \pm 3.1$ |
| 4 | $45.8 \pm 0.7$ | $46.6 \pm 0.6$ | $36.6 \pm 1.1$ | $20.2 \pm 3.0$ |
| 5 | $46.0 \pm 0.9$ | $46.3 \pm 0.9$ | $37.0 \pm 0.6$ | $21.3 \pm 1.5$ |
| 6 | $46.0 \pm 0.8$ | $45.8 \pm 0.9$ | $37.3 \pm 1.4$ | $20.1 \pm 2.5$ |
| 7 | $45.6 \pm 0.7$ | $46.5 \pm 0.9$ | $36.6 \pm 1.1$ | $19.6 \pm 3.4$ |
| 8 | $45.9 \pm 0.7$ | $46.6 \pm 0.8$ | $35.8 \pm 0.5$ | $18.8 \pm 2.2$ |
| 9 | $46.7 \pm 0.7$ | $46.7 \pm 0.8$ | $34.8 \pm 1.0$ | $17.6 \pm 1.6$ |
| 10 | $47.6 \pm 0.5$ | $47.6 \pm 0.8$ | $34.9 \pm 1.1$ | $16.8 \pm 2.7$ |
| 15 | $50.3 \pm 0.7$ | $47.9 \pm 0.8$ | $33.0 \pm 0.8$ | $17.8 \pm 1.0$ |
| 20 | $53.1 \pm 1.6$ | $47.0 \pm 0.2$ | $35.0 \pm 0.7$ | $17.7 \pm 1.2$ |

Table 19: Numeric values for results in Table 2 right plot (CIFAR-10, repeated participation, $(\epsilon, \delta) = (4, 10^{-5})$.

| number of updates | accuracy | | | |
|---|---|---|---|---|
| | BSR | AOF | Id | $A$ |
| 50 | $29.8 \pm 2.7$ | $31.5 \pm 1.4$ | $30.2 \pm 1.5$ | $20.8 \pm 0.8$ |
| 100 | $31.2 \pm 0.9$ | $34.0 \pm 1.5$ | $29.8 \pm 1.2$ | $16.5 \pm 2.9$ |
| 200 | $30.9 \pm 1.4$ | $31.5 \pm 1.9$ | $26.0 \pm 1.6$ | $15.9 \pm 2.3$ |
| 400 | $32.1 \pm 0.9$ | $33.3 \pm 0.8$ | $24.8 \pm 2.4$ | $14.7 \pm 2.0$ |
| 500 | $31.6 \pm 2.3$ | $32.8 \pm 1.8$ | $23.8 \pm 1.4$ | $14.9 \pm 2.0$ |
| 1000 | $30.2 \pm 1.6$ | $30.1 \pm 1.1$ | $17.5 \pm 0.7$ | $12.5 \pm 1.5$ |
| 2000 | $29.9 \pm 2.0$ | $30.5 \pm 1.2$ | $16.1 \pm 1.0$ | $11.3 \pm 1.1$ |

Table 20: Numeric values for results in Table 5 top left plot (CIFAR-10, single participation, $(\epsilon, \delta) = (2, 10^{-5})$.

| number of epochs | accuracy | | | |
|---|---|---|---|---|
| | BSR | AOF | Id | $A$ |
| 1 | $31.2 \pm 0.9$ | $34.0 \pm 1.5$ | $29.8 \pm 1.2$ | $16.5 \pm 2.9$ |
| 2 | $34.5 \pm 1.0$ | $36.5 \pm 0.6$ | $30.1 \pm 0.9$ | $18.6 \pm 2.1$ |
| 3 | $36.8 \pm 2.3$ | $38.0 \pm 1.2$ | $31.3 \pm 1.4$ | $17.6 \pm 1.6$ |
| 4 | $39.2 \pm 2.1$ | $40.1 \pm 1.6$ | $28.8 \pm 0.9$ | $17.4 \pm 2.1$ |
| 5 | $40.4 \pm 1.9$ | $41.0 \pm 0.9$ | $28.4 \pm 1.0$ | $16.1 \pm 2.1$ |
| 6 | $41.3 \pm 0.7$ | $41.8 \pm 1.3$ | $28.3 \pm 0.9$ | $15.2 \pm 1.3$ |
| 7 | $42.2 \pm 1.6$ | $44.1 \pm 0.9$ | $28.4 \pm 1.0$ | $15.4 \pm 1.9$ |
| 8 | $43.3 \pm 1.2$ | $44.4 \pm 1.1$ | $28.9 \pm 1.4$ | $16.1 \pm 1.7$ |
| 9 | $44.1 \pm 0.4$ | $45.2 \pm 0.5$ | $28.6 \pm 0.5$ | $15.6 \pm 1.6$ |
| 10 | $44.3 \pm 1.8$ | $43.8 \pm 0.3$ | $27.7 \pm 1.5$ | $15.8 \pm 0.9$ |
| 15 | $44.9 \pm 1.1$ | $43.9 \pm 0.8$ | $27.3 \pm 1.1$ | $14.9 \pm 1.0$ |
| 20 | $45.7 \pm 1.2$ | $43.0 \pm 1.3$ | $26.9 \pm 1.5$ | $14.2 \pm 0.9$ |

Table 21: Numeric values for results in Table 5 top right plot (CIFAR-10, repeated participation, $(\epsilon, \delta) = (2, 10^{-5})$.

| | | accuracy | | |
|---|---|---|---|---|
| number of updates | BSR | AOF | Id | $A$ |
| 50 | $31.4 \pm 1.8$ | $32.2 \pm 2.0$ | $32.6 \pm 3.4$ | $25.3 \pm 1.4$ |
| 100 | $37.7 \pm 2.6$ | $39.7 \pm 1.9$ | $38.3 \pm 1.3$ | $26.3 \pm 2.7$ |
| 200 | $40.3 \pm 1.8$ | $41.2 \pm 1.0$ | $35.6 \pm 0.5$ | $25.8 \pm 1.7$ |
| 400 | $38.8 \pm 1.1$ | $39.6 \pm 1.0$ | $34.4 \pm 1.4$ | $22.1 \pm 4.8$ |
| 500 | $40.4 \pm 1.1$ | $40.5 \pm 0.6$ | $33.1 \pm 1.5$ | $22.4 \pm 2.5$ |
| 1000 | $40.7 \pm 1.0$ | $40.0 \pm 0.6$ | $28.2 \pm 1.1$ | $16.1 \pm 4.2$ |
| 2000 | $39.2 \pm 1.5$ | $37.6 \pm 1.5$ | $23.5 \pm 1.7$ | $16.3 \pm 1.5$ |

Table 22: Numeric values for results in Table 5 bottom left plot (CIFAR-10, repeated participation, $(\epsilon, \delta) = (8, 10^{-5})$.

| | | accuracy | | |
|---|---|---|---|---|
| number of epochs | BSR | AOF | Id | $A$ |
| 1 | $37.7 \pm 2.6$ | $39.7 \pm 1.9$ | $38.3 \pm 1.3$ | $26.3 \pm 2.7$ |
| 2 | $44.7 \pm 1.4$ | $45.8 \pm 1.2$ | $36.8 \pm 1.0$ | $26.8 \pm 2.0$ |
| 3 | $49.1 \pm 1.0$ | $49.6 \pm 0.8$ | $40.2 \pm 1.1$ | $24.6 \pm 1.5$ |
| 4 | $51.2 \pm 1.0$ | $51.1 \pm 1.6$ | $40.8 \pm 1.7$ | $24.1 \pm 3.7$ |
| 5 | $53.3 \pm 1.4$ | $54.0 \pm 1.2$ | $41.4 \pm 0.3$ | $24.0 \pm 2.0$ |
| 6 | $54.5 \pm 0.6$ | $53.8 \pm 0.2$ | $40.9 \pm 0.4$ | $24.0 \pm 3.0$ |
| 7 | $55.5 \pm 0.8$ | $55.5 \pm 0.9$ | $39.9 \pm 1.3$ | $24.5 \pm 2.2$ |
| 8 | $55.7 \pm 0.9$ | $55.8 \pm 0.5$ | $41.0 \pm 1.4$ | $25.1 \pm 2.1$ |
| 9 | $55.9 \pm 0.5$ | $55.7 \pm 1.1$ | $41.7 \pm 0.6$ | $23.7 \pm 2.9$ |
| 10 | $56.2 \pm 1.2$ | $51.1 \pm 0.7$ | $40.6 \pm 1.1$ | $23.6 \pm 3.5$ |
| 15 | $56.0 \pm 0.6$ | $52.3 \pm 0.5$ | $39.3 \pm 1.0$ | $21.3 \pm 2.3$ |
| 20 | $55.2 \pm 1.2$ | $53.8 \pm 0.5$ | $38.1 \pm 1.4$ | $20.6 \pm 2.8$ |

Table 23: Numeric values for results in Table 5 bottom right plot (CIFAR-10, repeated participation, $(\epsilon, \delta) = (8, 10^{-5})$.

## E   Experimental Results for Different Optimizers

In this section we report on experimental results when different optimizers are used to (approximately) solve the AOF optimization problem (4). Besides cvxpy *(CVX)* these are standard *gradient descent (GD)* and the *Limited-Memory Broyden-Fletcher-Goldfarb-Shanno algorithm (LBFGS)*. The latter two we implement in *jax* using the *optax* toolbox. Similar to [Granqvist et al., 2024, ftrl_mechanism.py], we use an adaptive line-search for the step size of the gradient-based methods, which at the same time ensures the positive definiteness constraints of the optimization problem. Our implementation differs from theirs, however, in that our learning rate is not restricted to shrink monotonically, thereby avoiding premature termination.

### E.1   Runtime

We report the runtimes for the different methods in Tables 24 to 39. For comparison, we also include results for BSR and the CVX optimizers with three tolerance levels in the same settings, where practically feasible. Note that while the experiments for BSR and CVX used a single-core CPU-only environment, the experiments for GD and LBFGS were run on an NVIDIA H100 GPU with 16 available CPU cores. As a consequence, the absolute runtimes are not directly comparable between the methods, but they should rather be seen as illustrations of the scaling behavior of the method for different workload types and problem sizes.

Indeed, the results show a clear trend: BSR is the fastest, with almost no overhead. Even for the largest problem sizes of $n = 10\,000$, BSR never took more than 2.5s to despite running in the single-core CPU-only setup. GD and LBFGS benefit strongly from the GPU hardware. In the multiple participation setting ($p = 100, k = n/p$), they solve most workload sizes within a few

Table 24: $\alpha = 1.0$, $\beta = 0.9$, $p = 100$, $k = n/p$

| n | BSR | GD | LBFGS | CVX(tol=0.01) | CVX(tol=0.001) | CVX(tol=0.0001) |
|---|---|---|---|---|---|---|
| 100 | < 1s | 28.5s | 1m39s | 4.5s | 7m18s | 1h27m30s |
| 200 | < 1s | 1m10s | 2m31s | 37.1s | 21m00s | 10h35m00s |
| 300 | < 1s | 1m44s | 3m14s | 2m16s | 1h12m40s | 22h36m40s |
| 400 | < 1s | 2m35s | 3m46s | 6m06s | 2h14m50s | 53h03m20s |
| 500 | < 1s | 3m47s | 4m47s | 11m45s | 5h31m40s | 90h50m00s |
| 600 | < 1s | 4m27s | 5m11s | 26m50s | 17h36m40s | 40h16m40s |
| 700 | < 1s | 5m13s | 6m12s | 47m50s | 22h10m00s | 66h23m20s |
| 800 | < 1s | 5m52s | 7m30s | 1h29m40s | 38h03m20s | 164h26m40s |
| 900 | < 1s | 6m20s | 7m29s | 1h45m30s | 62h30m00s | 253h53m20s |
| 1000 | < 1s | 6m55s | 8m01s | 1h59m40s | 83h36m40s | 245h00m00s |
| 1500 | < 1s | 10m11s | 11m49s | 6h08m20s | 121h23m20s | timeout |
| 2000 | < 1s | 13m39s | 13m21s | 15h10m00s | 297h13m20s | timeout |
| 5000 | 1.1s | 1h09m55s | 33m00s | — | — | — |
| 10000 | 1.6s | 6h07m10s | 1h47m39s | — | — | — |

Table 25: $\alpha = 1.0$, $\beta = 0.0$, $p = 100$, $k = n/p$

| n | BSR | GD | LBFGS | CVX(tol=0.01) | CVX(tol=0.001) | CVX(tol=0.0001) |
|---|---|---|---|---|---|---|
| 100 | < 1s | 1.4s | 12.3s | 5.6s | 48.6s | 33.4s |
| 200 | < 1s | 2.4s | 20.0s | 1m09s | 2m38s | 3m39s |
| 300 | < 1s | 3.4s | 25.0s | 2m25s | 11m23s | 19m20s |
| 400 | < 1s | 6.4s | 30.2s | 12m15s | 26m10s | 1h14m40s |
| 500 | < 1s | 6.8s | 36.6s | 33m10s | 1h07m00s | 2h53m20s |
| 600 | < 1s | 9.8s | 44.1s | 52m00s | 7h45m00s | 4h53m20s |
| 700 | < 1s | 12.8s | 51.5s | 1h34m50s | 14h30m00s | 17h48m20s |
| 800 | < 1s | 13.2s | 54.3s | 3h33m20s | 17h26m40s | 26h55m00s |
| 900 | < 1s | 17.9s | 59.1s | 4h43m20s | 27h18m20s | 41h06m40s |
| 1000 | < 1s | 22.9s | 1m09s | 7h31m40s | 45h33m20s | 57h30m00s |
| 1500 | < 1s | 47.3s | 1m36s | 29h43m20s | 84h43m20s | 200h50m00s |
| 2000 | < 1s | 1m24s | 1m45s | 68h53m20s | 258h36m40s | timeout |
| 5000 | 2.4s | 16m35s | 4m18s | — | — | — |
| 10000 | 1.6s | 2h29m46s | 14m23s | — | — | — |

minutes, except the largest ones, which for GD can take a few hours. In the single participation setting ($k = 1$), LBFGS also occasionally need several hours to converge. In general, stronger weight decay (smaller $\alpha$) tends to lead to lower runtimes, while the use of momentum ($\beta = 0.9$) to higher times until convergence. CVX (on weak hardware) is orders of magnitude slower than the other methods. Furthermore, its runtime grow approximately cubic with the problem size, whereas for GD and LBFGS the relation is not too far from linear. Note that despite the stable patterns described above, all runtime results should be taken with caution, because internal parameters of the optimization, such as the convergence criterion and the specific implementation of the line search can substantially influence the overall runtime as well.

## E.2 Expected Approximation Error

Figures 6 and 7 report the expected approximation errors achieved by the different optimizers of AOF (4) and by BSR. For CVX, we report the smallest error value across all tolerance levels for which the optimization converged.

The curves show several clear trends. GD and LBFGS generally perform similarly, and achieve expected approximation errors slightly (at most a few percent) lower than BSR. An exception are the problems with momentum ($\beta = 0.9$) in the single participation setting, where it appears that SGD occasionally fails to find the optimum for large problem sizes ($n \geq 1500$). CVX performs comparably to the other methods for problems without momentum ($\beta = 0$). With momentum, however, the solutions it find are often worse than the other methods, especially in the single-participation setting and for medium to large problem sizes ($n \geq 500$). Presumably, even smaller *tolerance* values would be require here, which, however, would result in even longer runtimes.

Table 26: $\alpha = 0.9999, \beta = 0.9, p = 100, k = n/p$

| n | BSR | GD | LBFGS | CVX(tol=0.01) | CVX(tol=0.001) | CVX(tol=0.0001) |
|---|---|---|---|---|---|---|
| 100 | < 1s | 29.9s | 1m35s | 6.1s | 3m42s | 47m00s |
| 200 | < 1s | 1m10s | 2m33s | 36.3s | 18m00s | 6h56m40s |
| 300 | < 1s | 1m45s | 3m04s | 1m48s | 40m30s | 25h46m40s |
| 400 | < 1s | 2m32s | 3m46s | 6m21s | 2h35m20s | 57h13m20s |
| 500 | < 1s | 3m32s | 4m29s | 11m05s | 5h41m40s | 24h08m20s |
| 600 | < 1s | 4m21s | 4m56s | 10m41s | 13h36m40s | 41h56m40s |
| 700 | < 1s | 5m10s | 5m37s | 54m20s | 20h33m20s | 76h56m40s |
| 800 | < 1s | 5m39s | 6m20s | 1h21m40s | 38h20m00s | 158h53m20s |
| 900 | < 1s | 6m25s | 7m05s | 2h16m00s | 51h06m40s | 268h20m00s |
| 1000 | < 1s | 6m56s | 7m46s | 2h11m00s | 68h03m20s | 223h20m00s |
| 1500 | < 1s | 10m09s | 10m15s | 8h08m20s | 115h50m00s | timeout |
| 2000 | < 1s | 13m39s | 12m09s | 12h46m40s | 247h13m20s | timeout |
| 5000 | 2.4s | 1h10m18s | 26m09s | — | — | — |
| 10000 | 2.6s | 6h06m46s | 1h10m10s | — | — | — |

Table 27: $\alpha = 0.9999, \beta = 0.0, p = 100, k = n/p$

| n | BSR | GD | LBFGS | CVX(tol=0.01) | CVX(tol=0.001) | CVX(tol=0.0001) |
|---|---|---|---|---|---|---|
| 100 | < 1s | 1.2s | 13.7s | 6.0s | 14.1s | 20.3s |
| 200 | < 1s | 2.3s | 18.5s | 1m04s | 4m15s | 4m00s |
| 300 | < 1s | 3.5s | 23.8s | 4m03s | 15m10s | 10m51s |
| 400 | < 1s | 5.3s | 29.7s | 7m45s | 14m32s | 19m50s |
| 500 | < 1s | 6.1s | 38.5s | 22m10s | 1h23m50s | 2h44m20s |
| 600 | < 1s | 8.9s | 43.3s | 29m40s | 3h46m40s | 8h23m20s |
| 700 | < 1s | 12.0s | 44.5s | 1h56m50s | 12h03m20s | 12h11m40s |
| 800 | < 1s | 12.7s | 51.6s | 3h43m20s | 16h40m00s | 23h56m40s |
| 900 | < 1s | 16.1s | 1m02s | 5h11m40s | 26h30m00s | 30h33m20s |
| 1000 | < 1s | 20.3s | 1m05s | 8h23m20s | 40h33m20s | 41h23m20s |
| 1500 | < 1s | 39.8s | 1m19s | 35h50m00s | 78h53m20s | 170h00m00s |
| 2000 | < 1s | 1m10s | 1m27s | 72h30m00s | 239h26m40s | timeout |
| 5000 | 1.1s | 9m41s | 3m15s | — | — | — |
| 10000 | 2.5s | 1h00m37s | 10m51s | — | — | — |

Table 28: $\alpha = 0.999, \beta = 0.9, p = 100, k = n/p$

| n | BSR | GD | LBFGS | CVX(tol=0.01) | CVX(tol=0.001) | CVX(tol=0.0001) |
|---|---|---|---|---|---|---|
| 100 | < 1s | 32.6s | 1m52s | 3.7s | 3m36s | 58m10s |
| 200 | < 1s | 59.0s | 2m08s | 36.1s | 20m50s | 8h10m00s |
| 300 | < 1s | 1m25s | 2m36s | 3m38s | 37m30s | 25h48m20s |
| 400 | < 1s | 1m54s | 3m02s | 4m48s | 1h30m00s | 56h56m40s |
| 500 | < 1s | 2m13s | 3m33s | 13m45s | 2h58m20s | 84h43m20s |
| 600 | < 1s | 3m07s | 3m57s | 26m20s | 9h51m40s | 41h56m40s |
| 700 | < 1s | 3m04s | 4m09s | 52m30s | 13h10m00s | 85h33m20s |
| 800 | < 1s | 3m28s | 4m18s | 59m00s | 28h03m20s | 164h43m20s |
| 900 | < 1s | 3m59s | 4m33s | 57m00s | 39h10m00s | 280h33m20s |
| 1000 | < 1s | 4m34s | 5m21s | 1h27m00s | 59h43m20s | 258h03m20s |
| 1500 | < 1s | 5m48s | 5m21s | 4h41m40s | 81h40m00s | timeout |
| 2000 | < 1s | 7m26s | 5m28s | 14h16m40s | 219h10m00s | timeout |
| 5000 | < 1s | 34m57s | 9m16s | — | — | — |
| 10000 | 2.5s | 2h42m14s | 28m36s | — | — | — |

Table 29: $\alpha = 0.999$, $\beta = 0.0$, $p = 100$, $k = n/p$

| n | BSR | GD | LBFGS | CVX(tol=0.01) | CVX(tol=0.001) | CVX(tol=0.0001) |
|---|---|---|---|---|---|---|
| 100 | < 1s | 1.1s | 11.5s | 7.3s | 28.0s | 27.6s |
| 200 | < 1s | 2.0s | 15.6s | 1m10s | 46.8s | 3m06s |
| 300 | < 1s | 3.1s | 21.3s | 3m39s | 6m32s | 8m10s |
| 400 | < 1s | 3.8s | 26.0s | 8m19s | 30m30s | 29m50s |
| 500 | < 1s | 4.6s | 30.3s | 9m47s | 44m50s | 2h17m50s |
| 600 | < 1s | 5.3s | 29.3s | 48m30s | 2h01m40s | 1h59m50s |
| 700 | < 1s | 6.1s | 39.5s | 1h10m20s | 6h50m00s | 7h01m40s |
| 800 | < 1s | 6.7s | 38.6s | 3h16m40s | 13h10m00s | 13h50m00s |
| 900 | < 1s | 7.9s | 37.4s | 5h26m40s | 21h51m40s | 28h03m20s |
| 1000 | < 1s | 9.8s | 45.1s | 6h26m40s | 30h50m00s | 32h46m40s |
| 1500 | < 1s | 12.8s | 52.1s | 29h26m40s | 77h13m20s | 49h43m20s |
| 2000 | < 1s | 16.6s | 50.3s | 70h00m00s | 91h23m20s | 174h26m40s |
| 5000 | 1.3s | 1m19s | 1m29s | — | — | — |
| 10000 | 2.4s | 6m40s | 3m23s | — | — | — |

Table 30: $\alpha = 0.99$, $\beta = 0.9$, $p = 100$, $k = n/p$

| n | BSR | GD | LBFGS | CVX(tol=0.01) | CVX(tol=0.001) | CVX(tol=0.0001) |
|---|---|---|---|---|---|---|
| 100 | < 1s | 15.8s | 1m12s | 3.5s | 1m28s | 1h19m10s |
| 200 | < 1s | 19.0s | 1m09s | 14.4s | 22m00s | 11h23m20s |
| 300 | < 1s | 19.6s | 1m15s | 1m10s | 2h29m50s | 30h16m40s |
| 400 | < 1s | 24.5s | 1m23s | 3m07s | 4h55m00s | 59h43m20s |
| 500 | < 1s | 23.8s | 1m21s | 6m55s | 13h06m40s | 67h30m00s |
| 600 | < 1s | 28.7s | 1m20s | 24m10s | 31h40m00s | 44h43m20s |
| 700 | < 1s | 33.4s | 1m24s | 36m10s | 61h23m20s | 78h53m20s |
| 800 | < 1s | 33.7s | 1m29s | 48m20s | 73h36m40s | 146h06m40s |
| 900 | < 1s | 39.2s | 1m37s | 1h06m50s | 50h00m00s | 280h33m20s |
| 1000 | < 1s | 37.3s | 1m32s | 1h53m30s | 66h06m40s | 274h43m20s |
| 1500 | < 1s | 53.9s | 1m44s | 7h36m40s | 302h46m40s | timeout |
| 2000 | < 1s | 1m07s | 1m50s | 22h45m00s | timeout | timeout |
| 5000 | < 1s | 5m56s | 2m51s | — | — | — |
| 10000 | 2.4s | 29m34s | 7m45s | — | — | — |

Table 31: $\alpha = 0.99$, $\beta = 0.0$, $p = 100$, $k = n/p$

| n | BSR | GD | LBFGS | CVX(tol=0.01) | CVX(tol=0.001) | CVX(tol=0.0001) |
|---|---|---|---|---|---|---|
| 100 | < 1s | 1.0s | 9.9s | 3.9s | 9.0s | 15.9s |
| 200 | < 1s | < 1s | 9.8s | 30.3s | 1m40s | 1m38s |
| 300 | < 1s | < 1s | 10.2s | 2m11s | 2m11s | 1m50s |
| 400 | < 1s | 1.2s | 11.4s | 11m16s | 10m36s | 4m16s |
| 500 | < 1s | 1.3s | 14.6s | 24m10s | 17m40s | 22m20s |
| 600 | < 1s | 1.3s | 10.6s | 32m00s | 39m00s | 1h44m00s |
| 700 | < 1s | 1.4s | 10.7s | 1h06m10s | 1h33m20s | 1h46m30s |
| 800 | < 1s | 1.5s | 11.2s | 1h43m30s | 3h36m40s | 2h41m20s |
| 900 | < 1s | 1.6s | 11.3s | 3h15m00s | 7h23m20s | 8h21m40s |
| 1000 | < 1s | 1.8s | 13.0s | 5h11m40s | 8h31m40s | 9h26m40s |
| 1500 | < 1s | 2.7s | 12.7s | 20h10m00s | 29h10m00s | 30h00m00s |
| 2000 | < 1s | 3.2s | 12.5s | 46h40m00s | 31h56m40s | 33h03m20s |
| 5000 | < 1s | 17.0s | 18.3s | — | — | — |
| 10000 | 1.4s | 1m26s | 42.7s | — | — | — |

### Table 32: $\alpha = 1.0$, $\beta = 0.9$, $k = 1$

| n | BSR | GD | LBFGS | CVX(tol=0.01) | CVX(tol=0.001) | CVX(tol=0.0001) |
|---|---|---|---|---|---|---|
| 100 | < 1s | 28.5s | 1m39s | 2.3s | 1m46s | 50m20s |
| 200 | < 1s | 1m02s | 2m08s | 25.5s | 31m00s | 7h15m00s |
| 300 | < 1s | 1m37s | 3m07s | 1m45s | 1h53m40s | 21h03m20s |
| 400 | < 1s | 2m25s | 3m34s | 5m48s | 5h00m00s | 50h00m00s |
| 500 | < 1s | 2m39s | 4m05s | 7m38s | 12h28m20s | 71h40m00s |
| 600 | < 1s | 5m10s | 5m32s | 26m30s | 25h48m20s | 43h20m00s |
| 700 | < 1s | 4m22s | 5m30s | 37m10s | 49h26m40s | 94h26m40s |
| 800 | < 1s | 4m22s | 5m42s | 1h09m40s | 57h30m00s | 175h16m40s |
| 900 | < 1s | 5m37s | 6m25s | 2h04m50s | 36h23m20s | 305h33m20s |
| 1000 | < 1s | 6m53s | 6m32s | 2h19m20s | 53h03m20s | 260h00m00s |
| 1500 | < 1s | 10m24s | 10m08s | 13h13m20s | 242h30m00s | timeout |
| 2000 | < 1s | 13m39s | 12m45s | 33h03m20s | 64h10m00s | timeout |
| 5000 | 1.1s | 1h10m17s | 44m58s | — | — | — |
| 10000 | < 1s | 6h06m27s | 2h30m09s | — | — | — |

### Table 33: $\alpha = 1.0$, $\beta = 0.0$, $k = 1$

| n | BSR | GD | LBFGS | CVX(tol=0.01) | CVX(tol=0.001) | CVX(tol=0.0001) |
|---|---|---|---|---|---|---|
| 100 | < 1s | 1.4s | 12.3s | 6.8s | 6.4s | 19.6s |
| 200 | < 1s | 3.7s | 21.8s | 45.3s | 2m53s | 4m32s |
| 300 | < 1s | 3.3s | 25.1s | 3m00s | 11m09s | 7m37s |
| 400 | < 1s | 9.5s | 33.8s | 6m34s | 19m30s | 52m00s |
| 500 | < 1s | 19.3s | 50.4s | 16m10s | 41m40s | 22m50s |
| 600 | < 1s | 9.6s | 42.9s | 51m20s | 2h26m40s | 2h16m20s |
| 700 | < 1s | 20.3s | 54.3s | 1h47m10s | 4h40m00s | 5h45m00s |
| 800 | < 1s | 33.0s | 1m00s | 3h06m40s | 9h40m00s | 12h18m20s |
| 900 | < 1s | 57.2s | 1m22s | 5h06m40s | 14h05m00s | 16h28m20s |
| 1000 | < 1s | 1m12s | 1m39s | 7h03m20s | 21h13m20s | 28h20m00s |
| 1500 | < 1s | 1m54s | 1m39s | 24h51m40s | 80h16m40s | 53h20m00s |
| 2000 | < 1s | 5m39s | 2m56s | 61h56m40s | 110h33m20s | 146h23m20s |
| 5000 | 2.5s | 35m20s | 4m19s | — | — | — |
| 10000 | 1.6s | 5h11m15s | 13m51s | — | — | — |

### Table 34: $\alpha = 0.9999$, $\beta = 0.9$, $k = 1$

| n | BSR | GD | LBFGS | CVX(tol=0.01) | CVX(tol=0.001) | CVX(tol=0.0001) |
|---|---|---|---|---|---|---|
| 100 | < 1s | 29.9s | 1m35s | 6.5s | 5m12s | 1h14m00s |
| 200 | < 1s | 1m05s | 2m13s | 34.4s | 33m20s | 10h03m20s |
| 300 | < 1s | 1m38s | 2m49s | 2m52s | 1h41m10s | 29h43m20s |
| 400 | < 1s | 2m30s | 3m29s | 2m35s | 6h30m00s | 30h00m00s |
| 500 | < 1s | 2m29s | 4m03s | 7m17s | 11h55m00s | 75h50m00s |
| 600 | < 1s | 4m57s | 4m58s | 19m00s | 26h40m00s | 41h40m00s |
| 700 | < 1s | 5m13s | 7m24s | 42m20s | 43h36m40s | 99h10m00s |
| 800 | < 1s | 3m59s | 5m13s | 53m40s | 56h23m20s | 167h30m00s |
| 900 | < 1s | 5m04s | 5m47s | 1h27m30s | 34h10m00s | 302h46m40s |
| 1000 | < 1s | 6m53s | 6m12s | 2h15m10s | 54h10m00s | 274h10m00s |
| 1500 | < 1s | 10m05s | 13m04s | 12h23m20s | 308h20m00s | timeout |
| 2000 | < 1s | 13m37s | 9m39s | 35h00m00s | timeout | timeout |
| 5000 | 2.5s | 1h10m23s | 28m03s | — | — | — |
| 10000 | 2.5s | 6h06m15s | 1h54m24s | — | — | — |

Table 35: $\alpha = 0.9999$, $\beta = 0.0$, $k = 1$

| n | BSR | GD | LBFGS | CVX(tol=0.01) | CVX(tol=0.001) | CVX(tol=0.0001) |
|---|---|---|---|---|---|---|
| 100 | < 1s | 1.2s | 13.7s | 5.1s | 21.5s | 17.3s |
| 200 | < 1s | 3.1s | 20.5s | 55.7s | 3m45s | 7m08s |
| 300 | < 1s | 2.9s | 22.7s | 2m58s | 4m10s | 8m32s |
| 400 | < 1s | 9.3s | 34.4s | 6m59s | 31m30s | 26m20s |
| 500 | < 1s | 19.3s | 48.9s | 30m40s | 1h03m00s | 41m40s |
| 600 | < 1s | 8.2s | 41.8s | 1h24m40s | 2h06m40s | 1h22m10s |
| 700 | < 1s | 17.7s | 47.9s | 1h54m40s | 5h25m00s | 4h38m20s |
| 800 | < 1s | 30.3s | 57.8s | 2h26m50s | 8h36m40s | 7h43m20s |
| 900 | < 1s | 45.0s | 1m15s | 4h36m40s | 13h58m20s | 15h35m00s |
| 1000 | < 1s | 1m01s | 1m23s | 8h23m20s | 18h46m40s | 22h43m20s |
| 1500 | < 1s | 1m28s | 1m33s | 28h03m20s | 80h33m20s | 91h23m20s |
| 2000 | < 1s | 3m56s | 2m11s | 64h26m40s | 98h36m40s | 126h06m40s |
| 5000 | 2.5s | 58m03s | 4m57s | — | — | — |
| 10000 | 2.5s | 6h00m13s | 11m30s | — | — | — |

Table 36: $\alpha = 0.999$, $\beta = 0.9$, $k = 1$

| n | BSR | GD | LBFGS | CVX(tol=0.01) | CVX(tol=0.001) | CVX(tol=0.0001) |
|---|---|---|---|---|---|---|
| 100 | < 1s | 32.6s | 1m52s | 3.5s | 6m20s | 1h28m10s |
| 200 | < 1s | 1m00s | 2m13s | 32.2s | 26m30s | 8h36m40s |
| 300 | < 1s | 59.6s | 2m20s | 2m21s | 2h50m00s | 28h20m00s |
| 400 | < 1s | 2m08s | 2m56s | 5m53s | 3h36m40s | 58h03m20s |
| 500 | < 1s | 4m05s | 4m19s | 9m40s | 13h16m40s | 72h30m00s |
| 600 | < 1s | 2m03s | 3m35s | 12m33s | 22h50m00s | 44h10m00s |
| 700 | < 1s | 2m30s | 3m51s | 47m50s | 46h06m40s | 101h40m00s |
| 800 | < 1s | 3m33s | 4m01s | 1h06m20s | 78h03m20s | 145h00m00s |
| 900 | < 1s | 4m08s | 4m39s | 1h40m20s | 88h36m40s | 305h33m20s |
| 1000 | < 1s | 5m35s | 5m31s | 2h42m00s | 50h00m00s | timeout |
| 1500 | < 1s | 10m10s | 7m29s | 10h10m00s | 280h33m20s | timeout |
| 2000 | < 1s | 13m47s | 9m12s | 26h08m20s | timeout | timeout |
| 5000 | < 1s | 1h10m19s | 21m05s | — | — | — |
| 10000 | 2.5s | 6h08m30s | 45m33s | — | — | — |

Table 37: $\alpha = 0.999$, $\beta = 0.0$, $k = 1$

| n | BSR | GD | LBFGS | CVX(tol=0.01) | CVX(tol=0.001) | CVX(tol=0.0001) |
|---|---|---|---|---|---|---|
| 100 | < 1s | 1.1s | 11.5s | 5.9s | 6.3s | 32.4s |
| 200 | < 1s | 2.4s | 16.8s | 47.4s | 1m30s | 1m54s |
| 300 | < 1s | 6.3s | 31.3s | 2m48s | 7m17s | 17m40s |
| 400 | < 1s | 5.0s | 27.8s | 8m10s | 17m30s | 42m50s |
| 500 | < 1s | 9.4s | 33.0s | 13m49s | 45m50s | 1h06m00s |
| 600 | < 1s | 13.6s | 40.9s | 50m20s | 1h04m20s | 3h21m40s |
| 700 | < 1s | 19.3s | 48.0s | 1h53m00s | 3h06m40s | 3h21m40s |
| 800 | < 1s | 25.5s | 52.3s | 2h45m50s | 7h03m20s | 8h45m00s |
| 900 | < 1s | 34.5s | 1m07s | 3h55m00s | 9h23m20s | 13h15m00s |
| 1000 | < 1s | 42.4s | 1m10s | 5h46m40s | 17h08m20s | 18h21m40s |
| 1500 | < 1s | 1m35s | 1m40s | 26h55m00s | 54h10m00s | 56h40m00s |
| 2000 | < 1s | 2m45s | 2m14s | 60h33m20s | 52h13m20s | 82h30m00s |
| 5000 | < 1s | 21m16s | 4m32s | — | — | — |
| 10000 | 2.4s | 1h59m54s | 12m44s | — | — | — |

Table 38: $\alpha = 0.99$, $\beta = 0.9$, $k = 1$

| n | BSR | GD | LBFGS | CVX(tol=0.01) | CVX(tol=0.001) | CVX(tol=0.0001) |
|---|---|---|---|---|---|---|
| 100 | < 1s | 15.8s | 1m12s | 3.6s | 1m53s | 35m00s |
| 200 | < 1s | 15.6s | 1m14s | 31.8s | 14m57s | 9h00m00s |
| 300 | < 1s | 22.5s | 1m15s | 1m45s | 51m30s | 22h15m00s |
| 400 | < 1s | 33.6s | 1m33s | 3m56s | 2h19m00s | 61h40m00s |
| 500 | < 1s | 41.1s | 1m33s | 17m20s | 3h31m40s | 73h53m20s |
| 600 | < 1s | 47.8s | 1m40s | 31m50s | 8h40m00s | 41h40m00s |
| 700 | < 1s | 54.5s | 1m50s | 51m30s | 20h26m40s | 89h26m40s |
| 800 | < 1s | 1m03s | 1m53s | 1h12m40s | 39h43m20s | 140h00m00s |
| 900 | < 1s | 1m12s | 1m55s | 2h09m30s | 56h06m40s | 270h00m00s |
| 1000 | < 1s | 1m05s | 2m02s | 3h20m00s | 61h06m40s | 265h16m40s |
| 1500 | < 1s | 1m53s | 2m05s | 13h46m40s | 124h26m40s | timeout |
| 2000 | < 1s | 2m16s | 2m14s | 32h30m00s | 319h26m40s | timeout |
| 5000 | < 1s | 11m48s | 3m20s | — | — | — |
| 10000 | 2.5s | 1h02m59s | 7m50s | — | — | — |

Table 39: $\alpha = 0.99$, $\beta = 0.0$, $k = 1$

| n | BSR | GD | LBFGS | CVX(tol=0.01) | CVX(tol=0.001) | CVX(tol=0.0001) |
|---|---|---|---|---|---|---|
| 100 | < 1s | 1.0s | 9.9s | 3.8s | 8.1s | 22.1s |
| 200 | < 1s | 1.7s | 14.0s | 30.8s | 1m15s | 2m02s |
| 300 | < 1s | 2.5s | 18.8s | 1m18s | 5m23s | 8m00s |
| 400 | < 1s | 3.0s | 19.6s | 3m45s | 7m41s | 7m43s |
| 500 | < 1s | 3.5s | 23.9s | 9m51s | 14m17s | 38m30s |
| 600 | < 1s | 4.4s | 26.4s | 33m20s | 35m00s | 45m20s |
| 700 | < 1s | 4.3s | 24.7s | 1h02m20s | 1h22m40s | 1h42m10s |
| 800 | < 1s | 4.8s | 27.8s | 1h31m40s | 2h27m50s | 2h56m40s |
| 900 | < 1s | 5.4s | 28.7s | 2h11m10s | 4h01m40s | 4h26m40s |
| 1000 | < 1s | 5.6s | 30.3s | 2h51m40s | 5h45m00s | 6h45m00s |
| 1500 | < 1s | 8.5s | 39.3s | 11h21m40s | 17h55m00s | 17h26m40s |
| 2000 | < 1s | 11.5s | 42.9s | 26h46m40s | 45h00m00s | 42h46m40s |
| 5000 | < 1s | 57.1s | 1m07s | — | — | — |
| 10000 | 1.3s | 4m58s | 2m42s | — | — | — |

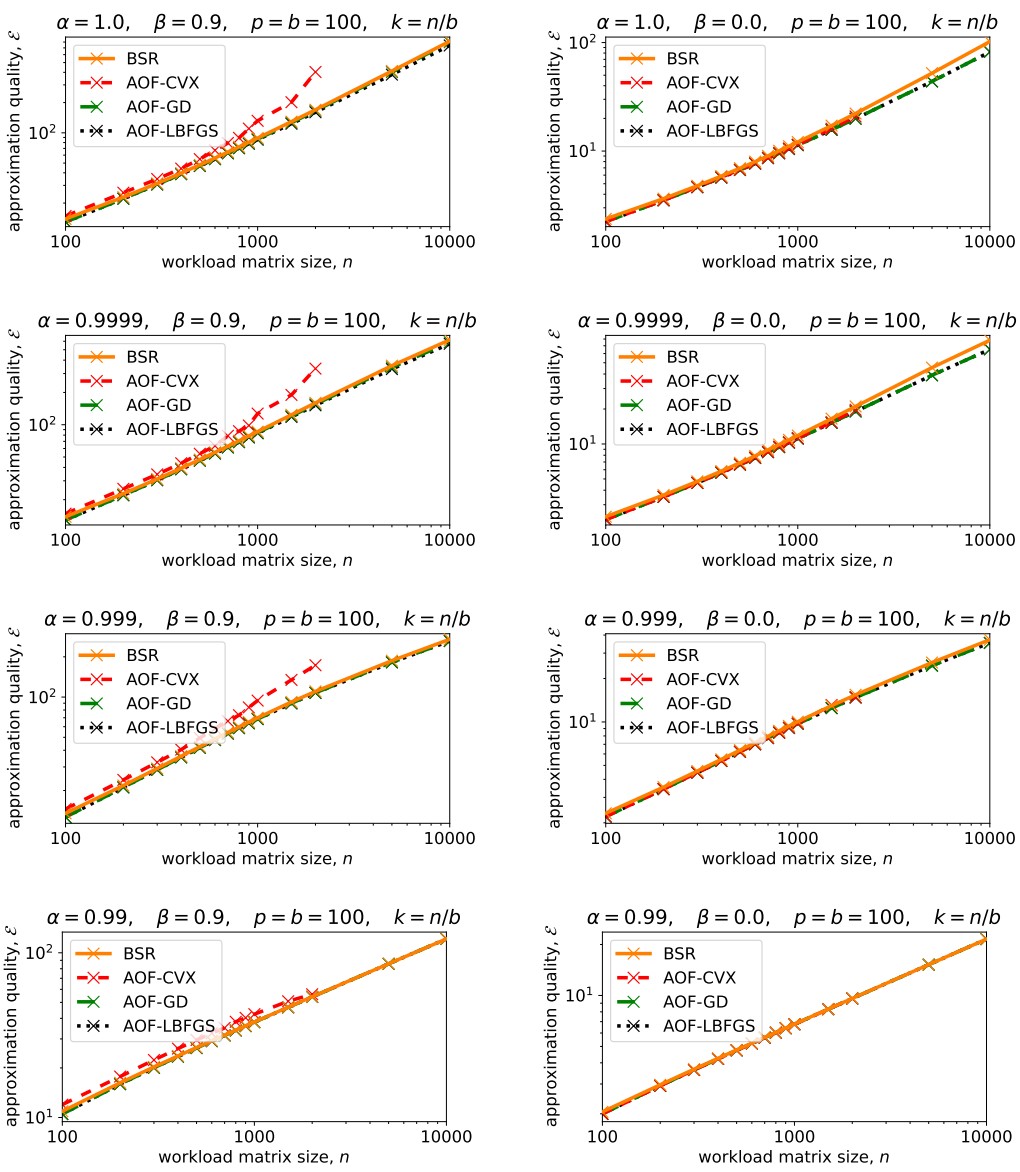

Figure 6: Expected approximation error for AOF with GD, LBFGS and CVX optimizers as well as for BSR in the multiple participations setting.

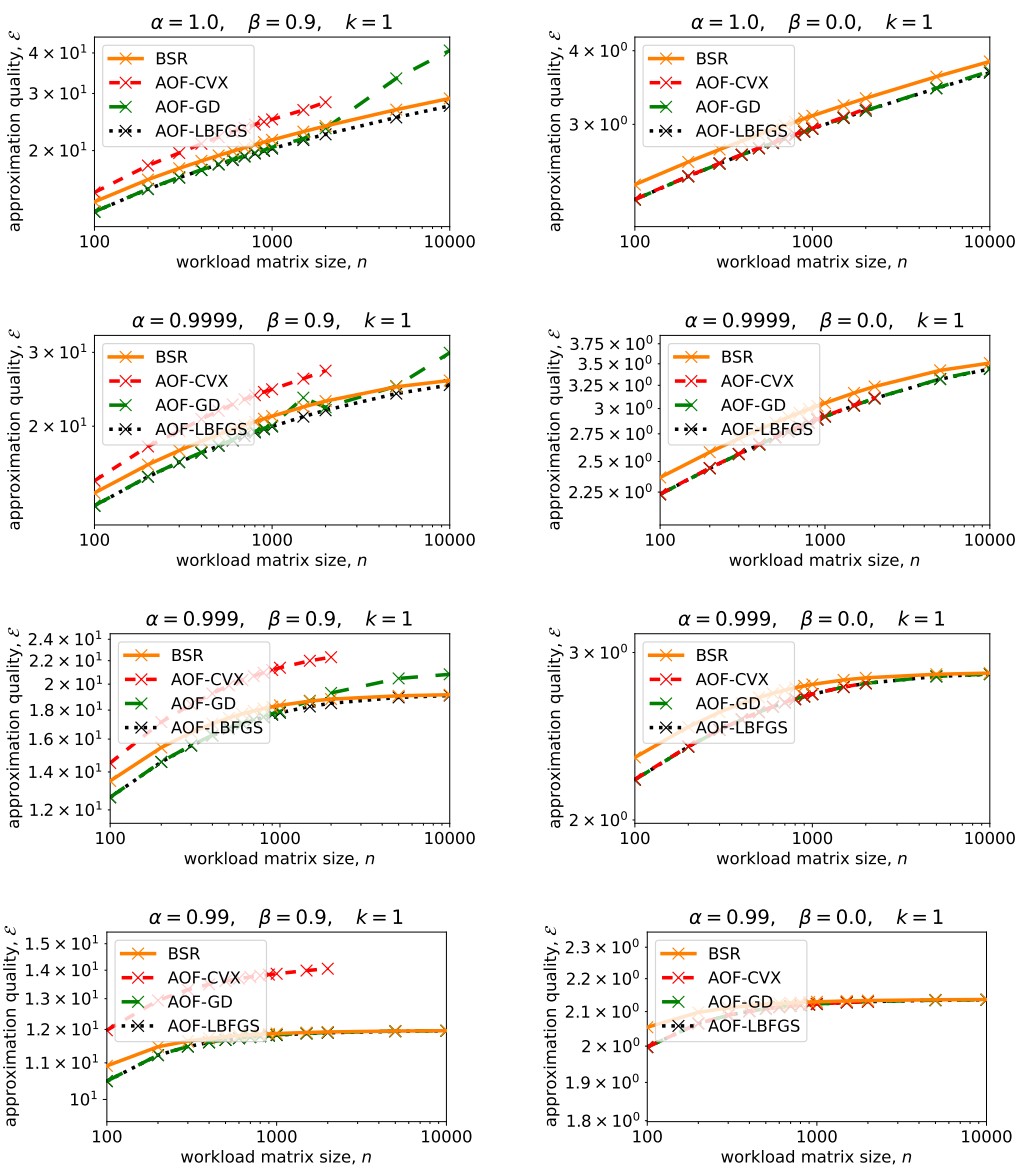

Figure 7: Expected approximation error for AOF with GD, LBFGS and CVX optimizers as well as for BSR in the single participation setting.

# F  Complete Proofs

In this section, we provide the complete proofs for our results from the main manuscript. For the convenience of the reader, we also restate the statements themselves.

## F.1  Proof of Theorem 1

**Theorem 1** (Square-Root of SGD Workload Matrix). *Let $A_{\alpha,\beta}$ be the workload matrix* (9). *Then $A_{\alpha,\beta} = C_{\alpha,\beta}^2$ for $C_{\alpha,\beta} = \mathrm{LDToep}(c_0, \ldots, c_{n-1})$, with $c_0 = 1$ and $c_j = \sum_{i=0}^{j} \alpha^{j-i} r_{j-i} r_i \beta^i$ for $j = 1, \ldots, n-1$ with coefficients $r_i = \left| \binom{-1/2}{i} \right|$. For any $p \in \{1, \ldots, n\}$, the $p$-banded BSR matrix $C_{\alpha,\beta}^{|p|}$ is obtained from this by setting all coefficients $c_j = 0$ for $j \geq p$.*

*Proof.* We observe that $A_{\alpha,\beta}$ can be written as

$$
A_{\alpha,\beta} = \begin{pmatrix} 1 & 0 & \ldots & 0 \\ \alpha & 1 & \ldots & 0 \\ \vdots & \vdots & \ddots & \vdots \\ \alpha^{n-1} & \alpha^{n-2} & \ldots & 1 \end{pmatrix} \times \begin{pmatrix} 1 & 0 & \ldots & 0 \\ \beta & 1 & \ldots & 0 \\ \vdots & \vdots & \ddots & \vdots \\ \beta^{n-1} & \beta^{n-2} & \ldots & 1 \end{pmatrix} =: E_\alpha \times E_\beta \qquad (20)
$$

Relying on the result from Henzinger et al. [2024], that $E_1^{1/2} = \begin{pmatrix} 1 & 0 & \ldots & 0 \\ r_1 & 1 & \ldots & 0 \\ \vdots & \vdots & \ddots & \vdots \\ r_{n-1} & r_{n-2} & \ldots & 1 \end{pmatrix}$ with

$r_k = \left| \binom{-1/2}{k} \right|$, one can check that the square roots of the matrices $E_\alpha, E_\beta$ are:

$$
E_\alpha^{1/2} = \begin{pmatrix} 1 & 0 & \ldots & 0 \\ \alpha r_1 & 1 & \ldots & 0 \\ \vdots & \vdots & \ddots & \vdots \\ \alpha^{n-1} r_{n-1} & \alpha^{n-2} r_{n-2} & \ldots & 1 \end{pmatrix} \qquad E_\beta^{1/2} = \begin{pmatrix} 1 & 0 & \ldots & 0 \\ \beta r_1 & 1 & \ldots & 0 \\ \vdots & \vdots & \ddots & \vdots \\ \beta^{n-1} r_{n-1} & \beta^{n-2} r_{n-2} & \ldots & 1 \end{pmatrix}.
$$
$$(21)$$

An explicit check yields that these matrices commute, i.e. $E_\alpha^{1/2} E_\beta^{1/2} = E_\beta^{1/2} E_\alpha^{1/2}$. Therefore

$$
C_{\alpha,\beta} = A_{\alpha,\beta}^{1/2} = E_\alpha^{1/2} \times E_\beta^{1/2} = \begin{pmatrix} 1 & 0 & \ldots & 0 \\ c_1 & 1 & \ldots & 0 \\ \vdots & \vdots & \ddots & \vdots \\ c_{n-1} & c_{n-2} & \ldots & 1 \end{pmatrix}, \quad \text{with } c_k = \sum_{i=0}^{k} \alpha^i r_i r_{k-i} \beta^{k-i}. \quad (22)
$$

$\square$

## F.2  Proof of Theorem 2

**Theorem 2** (Sensitivity for decreasing non-negative Toeplitz matrices). *Let $M = \mathrm{LDToep}(m_0, \ldots, m_{n-1})$ be a lower triangular Toeplitz matrix with decreasing non-negative entries, i.e.*

$m_0 \geq m_1 \geq m_2 \geq \ldots m_{n-1} \geq 0$. *Then its* sensitivity (2) *in the setting of $b$-min-separation is*

$$
\mathrm{sens}_{k,b}(M) = \left\| \sum_{j=0}^{k-1} M_{[\cdot, 1+jb]} \right\| = \left( \sum_{i=0}^{n-1} \left( \sum_{j=0}^{\min\{k-1, i/b\}} m_{i-jb} \right)^2 \right)^{1/2}, \qquad (10)
$$

*where $M_{[\cdot, 1+jb]}$ denotes the $(1 + jb)$-th column of $M$.*

*Proof.* Because all entries of $M$ are positive, so are the entries of $M^\top M$. Therefore, the condition is fulfilled such that (2) holds with equality, and

$$\operatorname{sens}^2_{k,b}(M) = \max_{\pi \in \Pi_{k,b}} \sum_{i,j \in \pi} (M^\top M)_{[i,j]} = \max_{\pi \in \Pi_{k,b}} f(\pi, \pi) \quad \text{for} \quad f(\pi, \pi') = \sum_{i \in \pi} \sum_{j \in \pi'} \langle M_{[\cdot,i]}, M_{[\cdot,j]} \rangle, \tag{23}$$

where $\Pi_{k,b} = \{ \pi \subset \{1, \dots, n\} : |\pi| \le k \wedge (\{i,j\} \subset \pi \Rightarrow i = j \vee |i - j| \ge b) \}$. We now establish that $\{1, 1+b, \dots, 1+(k-1)b\}$ is an optimal index set, which implies the statement of the theorem.

To see this, let $\pi^*$ be any optimal solution and let $i^* \in \pi^*$ be a column index that is not as far left as possible, i.e., $\pi^* \setminus \{i^*\} \cup \{i^* - 1\}$ would be a valid index set in $\Pi_{k,b}$. If such an index $i^*$ exists, we split $\pi^*$ into the *left* indices, which are smaller than $i^*$, and the remaining, *right*, ones: $\pi^* = \pi^*_L \dot\cup \pi^*_R$ with $\pi^*_L = \{i \mid i \in \pi \wedge i < i^*\}$, $\pi^*_R = \{i \mid i \in \pi \wedge i \ge i^*\}$. Then, we construct a new index set in which the left indices are kept but all right ones are shifted by one position to the left: $\pi' = \pi^*_L \cup \overleftarrow{\pi}^*_R$ with $\overleftarrow{\pi}^*_R = \{i - 1 \mid i \in \pi^*_R\}$. By the condition on $i^*$, we know $\pi' \in \Pi_{k,b}$.

We now prove that $f(\pi', \pi') \ge f(\pi, \pi)$, so $\pi'$ must also be optimal. First, we observe two inequalities: for any $i, j > 1$:

$$\langle M_{[\cdot,i-1]}, M_{[\cdot,j-1]} \rangle = \langle M_{[\cdot,i]}, M_{[\cdot,j]} \rangle + m_{n-i+1}m_{n-j+1} \ge \langle M_{[\cdot,i]}, M_{[\cdot,j]} \rangle, \tag{24}$$

and for $i \ge 1, j > 1$:

$$\langle M_{[\cdot,i]}, M_{[\cdot,j-1]} \rangle = \sum_{l=1}^{n} M_{[l,i]}, M_{[l,j-1]} = \sum_{l=j-1}^{n} m_{l-i}m_{l-j+1} \tag{25}$$

$$= \sum_{l=j-1}^{n-1} m_{l-i}m_{l-j+1} \quad + \quad m_{n-i}m_{n-j} \tag{26}$$

$$\ge \sum_{l=j}^{n} m_{l-i-1}m_{l-j} \tag{27}$$

$$\ge \sum_{l=j}^{n} m_{l-i}m_{l-j} = \sum_{l=1}^{n} M_{[l,i]}, M_{[l,j]} = \langle M_{[\cdot,i]}, M_{[\cdot,j]} \rangle, \tag{28}$$

where the last inequality holds because by assumption $m_{l-i-1} \ge m_{l-i}$ for $l \ge i + 1$.

Now, we split $f(\pi', \pi')$ and $f(\pi', \pi')$ into three terms: the inner products of indices below $i^*$, the ones of terms above $i^*$ and the ones between both,

$$f(\pi^*, \pi^*) = f(\pi^*_L, \pi^*_L) + f(\pi^*_R, \pi^*_R) + 2f(\pi^*_L, \pi^*_R). \tag{29}$$

$$f(\pi', \pi') = f(\pi^*_L, \pi^*_L) + f(\overleftarrow{\pi}^*_R, \overleftarrow{\pi}^*_R) + 2f(\pi^*_L, \overleftarrow{\pi}^*_R). \tag{30}$$

The first term appears identically in both expressions. The second term fulfills

$$f(\overleftarrow{\pi}^*_R, \overleftarrow{\pi}^*_R) = \sum_{i,j \in \overleftarrow{\pi}^*_R} \langle M_{[\cdot,i]}, M_{[\cdot,j]} \rangle = \sum_{i,j \in \pi^*_R} \langle M_{[\cdot,i-1]}, M_{[\cdot,j-1]} \rangle \tag{31}$$

$$\ge \sum_{i,j \in \pi^*_R} \langle M_{[\cdot,i]}, M_{[\cdot,j]} \rangle = f(\pi^*_R, \pi^*_R) \tag{32}$$

by Equation (24). The third term fulfills

$$f(\pi^*_L, \overleftarrow{\pi}^*_R) = \sum_{i \in \pi^*_L} \sum_{j \in \overleftarrow{\pi}^*_R} \langle M_{[\cdot,i]}, M_{[\cdot,j]} \rangle = \sum_{i \in \pi^*_L} \sum_{j \in \pi^*_R} \langle M_{[\cdot,i]}, M_{[\cdot,j-1]} \rangle \tag{33}$$

$$\ge \sum_{i \in \pi^*_L} \sum_{j \in \pi^*_R} \langle M_{[\cdot,i]}, M_{[\cdot,j]} \rangle = f(\pi^*_L, \pi^*_R) \tag{34}$$

by Equation (28). In combination, this establishes $f(\pi', \pi') \ge f(\pi^*, \pi^*)$, and since $\pi^*$ was already optimal, the same must hold for $\pi'$.

Using the above construction, we can create a new optimal index sets, $\pi^*$, until reaching one that does not contain any index $i^*$ as described anymore. Then $\pi^* = \{1, 1+b, \ldots, 1+(l-1)b\}$ for some $l \in \mathbb{N}$ must hold. If $l = k$, the statement of Theorem 2 is confirmed. Otherwise, $\pi' = \{1, \ldots, 1+(k-1)b\}$ is superset of $\pi^*$, so because of the positivity of entries, $f(\pi', \pi') \geq f(\pi^*, \pi^*)$ must hold. Once again, because $\pi^*$ was optimal, the same must hold for $\pi'$, which concludes the proof. $\square$

### F.3 Proof of Corollary 1

**Corollary 1.** *The sensitivity of the p-BSR for SGD can be computed using formula* (10).

*Proof.* From (1) we know that $C_{\alpha,\beta}$ is a Toeplitz matrix with coefficients $(1, c_1, \ldots, c_{n-1})$, where $c_j = \sum_{i=0}^{j} \alpha^i r_i r_{j-i} \beta^{j-i}$ for $0 \leq \beta < \alpha \leq 1$, with $r_i = |\binom{-1/2}{i}| = \frac{B_i}{4^i}$, where $B_i = \binom{2i}{i}$ is the *i-central binomial coefficient*. It suffices to show that $c_j \geq c_{j+1}$ for any $j \in \{1, \ldots, n-1\}$.

First, we show for the $r_i$ coefficients:

$$r_i - r_{i+1} = \frac{1}{4^i}\binom{2i}{i} - \frac{1}{4^{i+1}}\binom{2i+2}{i+1} = \frac{1}{4^i}\frac{(2i)!}{i!\,i!} - \frac{1}{4^{i+1}}\frac{(2i+2)!}{(i+1)!\,(i+1)!} \tag{35}$$

$$= \frac{1}{4^i}\frac{(2i)!}{i!\,i!}\left(1 - \frac{1}{4}\frac{(2i+2)(2i+1)}{(i+1)(i+1)}\right) = r_i\left(1 - \frac{2i+1}{2(i+1)}\right) \tag{36}$$

$$= \frac{r_i}{2(i+1)} = \frac{1}{4^i \cdot 2}C_{i+1} \tag{37}$$

where $C_j = \frac{1}{j+1}B_j = \frac{1}{j+1}\binom{2j}{j}$ is the $j$-th *Catalan number*.

Now, we study the case $\alpha = 1$. If $\beta = 0$, then $c_1 = c_2 = \cdots = c_n = 1$, so monotonicity is fulfilled. Otherwise, i.e. $0 < \beta < 1$, we write

$$c_k - c_{k+1} = \sum_{i=0}^{k} r_i(r_{k-i} - r_{k+1-i})\beta^i - r_{k+1}r_0\beta^{k+1} \tag{38}$$

$$\geq \frac{1}{4^k}\sum_{i=0}^{k}\frac{1}{2}B_i C_{k-i}\beta^{k-i} - \frac{1}{4^{k+1}}B_{k+1}\beta^{k+1} \tag{39}$$

$$= \frac{\beta^k}{4^{k+1}}\left[2\sum_{i=0}^{k}B_i C_{k-i}\beta^{-i} - B_{k+1}\beta\right] \tag{40}$$

Using the classic identity between Catalan numbers, $2\sum_{i=0}^{k}B_{k-i}C_i = B_{k+1}$, e.g. [Batir et al., 2021, Identity 4.2] we obtain

$$= \frac{\beta^k}{4^{k+1}}\left[2\sum_{i=0}^{k}B_i C_{k-i}(\beta^{-i} - \beta)\right] > 0, \tag{41}$$

where the last inequality follow from the fact that $\beta^{-i} - \beta > 0$ for each $i = 0, \ldots, k$ and any $\beta < 1$. This proves the monotonicity of $c_k$.

For $\alpha < 1$, we observe that $c_j = \alpha^j \sum_{i=1}^{j} r_i r_{k-i}\gamma^{j-i}$ for $\gamma = \frac{\alpha}{\beta} < 1$. Clearly, the sequence $\alpha^j$ is decreasing, and by the above argument, the sum is decreasing, too. Consequently, $c_j$ is the product of two decreasing sequences, so it is also decreasing, which concludes the proof. $\square$

### F.4 Useful Lemmas

Before the remaining proofs, we establish a number of useful lemmas.

**Lemma 2.** *For any $C \in \mathbb{R}^{n \times n}$ with $C^\top C \geq 0$ it holds for any $b \in \{1, \ldots, n\}$ that*

$$\text{sens}_{1,b}^2(C) = \|C\|_{2,\infty}, \tag{42}$$

*where $\|C\|_{2,\infty}^2 = \max_{i=1,\ldots,n}\|C_{[\cdot,i]}\|^2$.*

*Proof.* This follows directly from Theorem 2:

$$\text{sens}_{1,b}^2(C) = \max_{\pi \in \Pi_{1,b}} \sum_{i,j \in \pi} [C^\top C]_{i,j} = \max_{i=1,\dots,n} [C^\top C]_{i,i} = \max_{i=1,\dots,n} \|C_{[\cdot,i]}\|^2. \qquad (43)$$

$\square$

**Lemma 3.** *For any $C \in \mathbb{R}^{n \times n}$ with $C^\top C \geq 0$ it holds for any $b \in \{1,\dots,n\}$ and $k \in \{1,\dots,\frac{b}{n}\}$ that*

$$\frac{k}{n}\|C\|_F^2 \leq \text{sens}_{k,b}^2(C) \leq k\|C\|_F^2, \qquad (44)$$

*Proof.* We first show the upper bound. Observe that for any $\pi \subset [n]$:

$$\sum_{i,j \in \pi} [C^\top C]_{i,j} = \sum_{i,j \in \pi} \langle C_{[\cdot,i]}, C_{[\cdot,j]} \rangle \leq \sum_{i,j \in \pi} \|C_{[\cdot,i]}\| \|C_{[\cdot,j]}\| \qquad (45)$$

$$= (\sum_{i \in \pi} \|C_{[\cdot,i]}\|)^2 \leq |\pi| \sum_{i \in \pi} \|C_{[\cdot,i]}\|^2 \leq |\pi| \|C\|_F^2 \qquad (46)$$

Therefore, using Theorem 2:

$$\text{sens}^2(C) = \max_{\pi \in \Pi_{k,b}} \sum_{i,j \in \pi} [C^\top C]_{i,j} \leq k\|C\|_F^2 \qquad (47)$$

For the lower bound, we introduce some additional notation. Let $\tilde{\Pi}_k$ be the set of $b$-separated index sets with *exactly* $k$ elements. Then, from Theorem 2, we obtain

$$\text{sens}_{k,b}^2(C) = \max_{\pi \in \Pi_{k,b}} \sum_{i,j \in \pi} [C^\top C]_{ij} \geq \max_{\pi \in \Pi_{k,b}} \sum_{i \in \pi} [C^\top C]_{ii} = \max_{\pi \in \Pi_{k,b}} S(\pi) \geq \max_{\pi \in \tilde{\Pi}_k} S(\pi), \qquad (48)$$

with the notation $S(I) = \sum_{i \in I} \|C_{[\cdot,i]}\|^2$ for any index set $I \subset \{1,\dots,n\}$.

Now, we prove by backwards induction over $k = 1, \dots, \frac{n}{b}$:

$$\max_{\pi \in \tilde{\Pi}_k} S(\pi) \geq \frac{k}{n}\|C\|_F^2 \qquad (49)$$

As base case, let $k = \frac{n}{b}$. Denote by $\pi_i := \{i, i+b, i+2b, \dots, i+(n-b)\}$ for $i = 1, \dots, b$ the uniformly spaced index sets. By construction they all fulfill $\pi_i \in \tilde{\Pi}_{n/b}$ and $\bigcup_{i=1}^n \pi_i = [n]$, where the union is disjoint. Therefore

$$\max_{\pi \in \tilde{\Pi}_{n/b}} S(\pi) \geq \max_{i=1,\dots,b} S(\pi_i) \geq \frac{1}{b}\sum_{i=1}^b S(\pi_i) = \frac{1}{b}S([n]) = \frac{1}{b}\|C\|_F^2. \qquad (50)$$

This proves the statement (49), because $\frac{1}{b} = \frac{k}{n}$ in this case.

As an induction step, we prove that if (49) holds for some value $k \leq \frac{n}{b}$, then it also holds for $k - 1 \geq 1$.

Let $\pi^* \in \text{argmax}_{\pi \in \tilde{\Pi}_k} S(\pi)$ and $j^* = \text{argmin}_{j \in \pi^*} S(\{j\})$, such that we know that $S(\{j\}) \leq \frac{1}{k}S(\pi^*)$. Now, set $\pi' = \pi^* \setminus \{j^*\}$. Because $\pi' \in \tilde{\Pi}_{k-1}$, it follows that

$$\max_{\pi \in \tilde{\Pi}_{k-1}} S(\pi) \geq S(\pi') = S(\pi^*) - S(\{j\}) \geq \frac{k-1}{k}S(\pi^*) \geq \frac{k-1}{n}\|C\|_F^2, \qquad (51)$$

where in the last step we used the induction hypothesis. This concludes the proof. $\square$

**Lemma 4.** *For $C_{\alpha,\beta}$ as in (1), and $k = 1$, it holds that*

$$\frac{1}{n}\sum_{j=1}^n \sum_{i=0}^{n-j} c_i^2 \leq \mathcal{E}(C_{\alpha,\beta}, C_{\alpha,\beta}) \leq \sum_{i=0}^{n-1} c_i^2 \qquad (52)$$

*Proof.* From Lemmas 3 and 2 we obtain

$$\mathcal{E}(C_{\alpha,\beta}, C_{\alpha,\beta}) \leq \frac{1}{\sqrt{n}}\|C_{\alpha,\beta}\|_F \operatorname{sens}_{1,b}(C_{\alpha,\beta}) \leq \Big( \operatorname{sens}_{1,b}(C_{\alpha,\beta}) \Big)^2 \leq \|C\|_{2,\infty}^2 = \sum_{i=0}^{n-1} c_i^2, \quad (53)$$

where the last identify follows from the explicit form of $C_{\alpha,\beta}$. The lower bound follows from

$$\mathcal{E}(C_{\alpha,\beta}, C_{\alpha,\beta}) = \frac{1}{\sqrt{n}}\|C_{\alpha,\beta}\|_F \operatorname{sens}_{1,b}(C_{\alpha,\beta}) \geq \frac{1}{n}\|C\|_F^2 \tag{54}$$

and again the explicit form of $\|C\|_F^2$. $\qquad\square$

**Lemma 5.** *For* $r_j = |\binom{-1/2}{j}| = \frac{1}{4^j}\binom{2j}{j}$ *it holds that:*

$$r_0 = 1 \quad and \quad r_1 = \frac{1}{2} \quad and\ in\ general \quad \frac{1}{2\sqrt{j}} \leq r_j \leq \frac{1}{\sqrt{\pi j}} \quad for\ j \geq 1. \tag{55}$$

*Proof.* The double inequality is a particular case of a more general pair of binomial inequalities when $k = j$ and $m = 2j$:

$$\sqrt{\frac{m}{8k(m-k)}}2^{mH(k/m)} \leq \binom{m}{k} \leq \sqrt{\frac{m}{2\pi k(m-k)}}2^{mH(k/m)}, \tag{56}$$

where $H(k/m)$ is the binary entropy function, with $H(1/2) = 1$. The proof of the general result (56), can be found in MacWilliams and Sloane [1977, Chapter 10, Lemma 7, p309]. $\qquad\square$

**Lemma 6.** *Let* $c_k = \sum_{j=0}^k \alpha^j r_j r_{k-j} \beta^{k-j}$ *as in* (1). *Then* $c_0 = 1$, *and for* $j \geq 1$:

$$\frac{\alpha^j}{2\sqrt{j+1}} \leq c_j \leq \frac{\alpha^j}{(1-\frac{\beta}{\alpha})\sqrt{j+1}}. \tag{57}$$

*Proof.* We exploit the upper and lower bounds from Lemma 5. First, we write $c_k = \alpha^k \sum_{j=0}^k r_j r_{k-j} \gamma^j$ with $\gamma := \frac{\beta}{\alpha}$. Then we check immediately that $c_0 = 1$ and $c_1 = \frac{1}{2}(\alpha+\beta) = \frac{\alpha}{2}(1+\gamma) \leq \frac{\alpha}{2}\frac{1}{1-\gamma}$.

For $j \geq 2$ we derive the upper bound by

$$\frac{c_j}{\alpha^j} = r_j(1+\gamma^j) + \sum_{i=1}^{j-1} r_i r_{j-i}\gamma^i \leq \frac{1+\gamma^j}{\sqrt{\pi j}} + \sum_{i=1}^{j-1}\frac{\gamma^i}{\pi\sqrt{i(j-i)}} \tag{58}$$

$$\leq \frac{1+\gamma^j}{\sqrt{\pi j}} + \sum_{i=1}^{j-1}\frac{\gamma^i}{\pi\sqrt{j-1}} \leq \frac{\sqrt{\pi}-1}{\pi\sqrt{j}} + \frac{\sqrt{\pi}-1}{\pi\sqrt{j}}\gamma^j + \frac{1}{\pi\sqrt{j-1}}\sum_{i=0}^{j}\gamma^i \tag{59}$$

$$= \frac{\sqrt{\pi}-1}{\pi\sqrt{j}}(1+\gamma^j) + \frac{1}{\pi\sqrt{j-1}}\frac{1}{(1-\beta)} \tag{60}$$

$$= \underbrace{\frac{(\sqrt{\pi}-1)\sqrt{j+1}}{\sqrt{j}}}_{\leq 1}\frac{(1+\gamma^k j)}{\pi\sqrt{j+1}} + \underbrace{\frac{\sqrt{j+1}}{\sqrt{j-1}}}_{\leq\sqrt{3}\leq 2}\frac{1}{\pi\sqrt{j+1}}\frac{1}{(1-\gamma)} \tag{61}$$

$$\leq \frac{3}{\pi\sqrt{j+1}}\frac{1}{(1-\gamma)} \leq \frac{1}{\sqrt{j+1}}\frac{1}{(1-\gamma)}, \tag{62}$$

which proves the upper bound on $a_j$. The lower bound for $j \geq 1$ follows trivially from

$$c_j \geq \alpha^j r_j \geq \frac{\alpha^j}{2\sqrt{j}} \geq \frac{\alpha^j}{2\sqrt{j+1}} \tag{63}$$

$$\square$$

**Lemma 7.** *For $j \in \{1, \dots, n\}$ it holds*

$$\frac{\log(j+1)}{4} \leq \sum_{i=0}^{j-1} c_i^2 \leq \frac{1 + \log j}{(1-\beta)^2} \tag{64}$$

*for $\alpha = 1$, and otherwise*

$$1 \leq \sum_{i=0}^{j-1} c_i^2 \leq \frac{1}{(\alpha - \beta)^2} \log\left(\frac{1}{1 - \alpha^2}\right) \tag{65}$$

*Proof.* We first prove the result for $\alpha = 1$. Combining Lemmas 4 and 6 we obtain

$$\sum_{i=0}^{j-1} c_i^2 \leq \frac{1}{(1-\beta)^2} \sum_{i=0}^{j-1} \frac{1}{i+1} = \frac{1}{(1-\beta)^2} \sum_{i=1}^{j} \frac{1}{i} \leq \frac{1 + \log j}{(1-\beta)^2}. \tag{66}$$

$$\sum_{i=0}^{j-1} c_i^2 \geq \frac{1}{4} \sum_{i=0}^{j-1} \frac{1}{i+1} = \frac{1}{4} \sum_{i=1}^{j} \frac{1}{i} \geq \frac{\log(j+1)}{4}. \tag{67}$$

For $\alpha < 1$, if follows analogously:

$$\sum_{i=0}^{j-1} c_i^2 \leq \frac{1}{(1 - \frac{\alpha}{\beta})^2} \sum_{i=0}^{j-1} \frac{\alpha^{2i}}{i+1} \leq \frac{1}{(\alpha - \beta)^2} \sum_{i=1}^{\infty} \frac{\alpha^{2i}}{i} = \frac{1}{(\alpha - \beta)^2} \log\left(\frac{1}{1 - \alpha^2}\right). \tag{68}$$

$$\sum_{i=0}^{j-1} c_i^2 \geq \frac{1}{4} \sum_{i=0}^{j-1} \frac{\alpha^{2i}}{i+1} = \frac{1}{4\alpha^2} \sum_{i=1}^{j} \frac{\alpha^{2i}}{i} = \frac{1}{4\alpha^2} \left[ \sum_{i=1}^{\infty} \frac{\alpha^{2i}}{i} - \sum_{i=j+1}^{\infty} \frac{\alpha^{2i}}{i} \right] \tag{69}$$

$$\geq \frac{1}{4\alpha^2} \left[ \log\left(\frac{1}{1 - \alpha^2}\right) - \frac{\alpha^{2(j+1)}}{(j+1)(1 - \alpha^2)} \right], \tag{70}$$

where the last term emerges from $\sum_{i=j+1}^{\infty} \frac{\alpha^{2i}}{i} \geq \frac{\alpha^{2(j+1)}}{j+1} \sum_{i=0}^{\infty} \alpha^{2i} = \frac{\alpha^{2(j+1)}}{j+1} \frac{1}{1-\alpha^2}$.  $\square$

**Lemma 8.** *Let $0 \leq \beta < \alpha \leq 1$. Let $\sigma_1 \geq \cdots \geq \sigma_n$ be the sorted list of singular values of $A_{\alpha,\beta}$. If $\alpha < 1$, then for $j = 1, \dots, n$:*

$$\frac{1}{(1+\alpha)(1+\beta)} \leq \sigma_j \leq \frac{1}{(1-\alpha)(1-\beta)} \tag{71}$$

*and*

$$n \leq \|A_{\alpha,\beta}\|_* \leq \frac{n}{(1-\alpha)(1-\beta)}. \tag{72}$$

*If $\alpha = 1$, then for $j = 1, \dots, n$,*

$$\frac{2}{\pi} \frac{1}{1+\beta} \frac{n}{j} \leq \sigma_j \leq \frac{1}{1-\beta} \frac{n}{j} \tag{73}$$

*and consequently*

$$\frac{2}{\pi} \frac{(n+1)\log(n+1)}{1+\beta} \leq \|A_{1,\beta}\|_* \leq \frac{(n+1)(1+\log n)}{1+\beta} \tag{74}$$

*Proof.* The statements on the singular values follow from the following Lemma 9, because $A_{\alpha,\beta} = E_\alpha E_\beta$. Because $E_\alpha$ and $E_\beta$ are diagonalizable and they commute, we have $\sigma_n(E_\beta)\sigma_j(E_\alpha) \leq \sigma_j(E_\alpha E_\beta) \leq \sigma_1(E_\beta)\sigma_j(E_\alpha)$. For $\alpha < 1$ the lower bound follows from $\|A_{\alpha,\beta}\|_* \geq \operatorname{trace} A_{\alpha,\beta}$, and the upper bound follows from the identity $\|A_{\alpha,\beta}\|_* = \sum_{j=1}^{n} \sigma_j$.

For $\alpha = 1$, the bounds follow from the same identity together with the fact that

$$\log(n+1) \leq \sum_{j=1}^{n} \frac{1}{j} \leq \log(n) + 1. \tag{75}$$

$\square$

**Lemma 9** (Singular values of $E_t$)**.** *For $0 \leq t \leq 1$, let $E_t = \mathrm{LDToep}(1, t, \ldots, t^{n-1}) \in \mathbb{R}^{n \times n}$. Then the singular values $\sigma_1(E_t) \geq \cdots \geq \sigma_n(E_t)$ fulfill for $i = 1, \ldots, n$:*

$$\frac{1}{1+t} \leq \sigma_i(E_t) \leq \frac{1}{1-t} \quad \text{for } 0 \leq t < 1, \qquad \text{and} \qquad \sigma_i(E_1) = \frac{1}{\sin\left(\frac{i-\frac{1}{2}}{n+\frac{1}{2}} \frac{\pi}{2}\right)}. \tag{76}$$

*Proof.* We follow the steps of SebastienB [2017], and use that the singular values of $E_t$ are the reciprocals of the singular values of $E_t^{-1}$, which themselves are the eigenvalues of $(E_t)^{-1}((E_t)^{-1})^{\top} =: T$, i.e., for $i = 1, \ldots, n$:

$$\sigma_j(E_t) = \frac{1}{\sqrt{\lambda_{n+1-j}(T)}} \tag{77}$$

The $E_t^{-1}$ and $T$ can be computed explicitly as

$$E_t^{-1} = \begin{pmatrix} 1 & 0 & 0 & 0 & \ldots & 0 \\ -t & 1 & 0 & 0 & \ldots & 0 \\ 0 & -t & 1 & 0 & \ldots & 0 \\ \vdots & \ddots & \ddots & \ddots & \vdots & \vdots \\ 0 & \ldots & 0 & -t & 1 & 0 \\ 0 & \ldots & 0 & 0 & -t & 1 \end{pmatrix}, \qquad T = \begin{pmatrix} 1 & -t & 0 & \ldots & 0 \\ -t & 1+t^2 & -t & \ldots & 0 \\ 0 & \ddots & \ddots & \vdots & \vdots \\ 0 & \ldots & -t & 1+t^2 & -t \\ 0 & \ldots & 0 & -t & 1+t^2 \end{pmatrix} \tag{78}$$

**Lemma 10.** *All eigenvalues, $\mu$, of $T$ fulfill*

$$(1-t)^2 \leq \mu \leq (1+t)^2 \tag{79}$$

*Proof.* By Gershgorin's circle theorem [Gershgorin, 1931], we know that $\mu$ fulfills i) $|1 - \mu| \leq t$, i.e. $1 - t \leq t \leq \mu \leq 1 + t$ or ii) $|1 + t^2 - \mu| \leq 2t$, i.e. $1 - 2t + t^2 \leq \mu \leq 1 + 2t + t^2$. For $t \in [0, 1]$ the first condition implies the second, so (79) must hold. $\qquad \square$

**Case I:** For $t < 1$, the statement (76) follows from Lemma 10 in combination with (77).

**Case II:** For $t = 1$ the matrix simplifies to $T = \begin{pmatrix} 1 & -1 & 0 & \ldots & 0 \\ -1 & 2 & -1 & \ldots & 0 \\ 0 & \vdots & \ddots & \vdots & \vdots \\ 0 & \ldots & -1 & 2 & -1 \\ 0 & \ldots & 0 & -1 & 2 \end{pmatrix}$. Note that $T$ is not exactly Toeplitz, because of the top left entry, so closed-form expressions for the eigenvalues of tridiagonal Toeplitz matrices do not apply to it. Instead, we can compute its eigenvalues explicitly. Matrices of this form have been studied by Elliott [1953]; for completeness, we provide a full proof here.

Let $\mu$ be an eigenvalue of $T$ with eigenvector $\Psi = (\Psi_0, \ldots, \Psi_{n-1})$. From the eigenvector equation $T\Psi = \mu\Psi$ we obtain

$$\mu\Psi_0 = \Psi_0 - \Psi_1 \tag{80}$$
$$\mu\Psi_k = -\Psi_{k-1} + 2\Psi_k - \Psi_{k+1} \qquad \text{for } k = 1, \ldots, n-2 \tag{81}$$
$$\mu\Psi_{n-1} = -\Psi_{n-2} + 2\Psi_{n-1} \tag{82}$$

which yields a linear recurrence relation

$$\Psi_{k+1} = (2 - \mu)\Psi_k - \Psi_{k-1} \quad \text{for } k = 1, \ldots, n-2 \tag{83}$$

with two boundary conditions

$$\Psi_1 = (1 - \mu)\Psi_0 \tag{84}$$
$$\Psi_{n-2} = (2 - \mu)\Psi_{n-1}. \tag{85}$$

We solve the recurrence relation using the polynomial method [Greene and Knuth, 1990]. The characteristic polynomial of (83) is $P(z) = z^2 + (\mu - 2)z + 1$. Its roots are

$$r_{\pm} = \frac{2 - \mu}{2} \pm \sqrt{\left(\frac{2 - \mu}{2}\right)^2 - 1} = \frac{(2 - \mu) \pm i\sqrt{4 - (2 - \mu)^2}}{2} = e^{\pm i\theta} \tag{86}$$

for some value $\theta \in [0, 2\pi)$. Note that the expression under the second square root is positive, because of Lemma 10. The last equation is a consequence of, $|r_\pm|^2 = \frac{1}{4}\left((2-\mu)^2 + (4-(2-\mu)^2)\right) = 1$. Consequently,

$$\mu = 2 - 2\Re(e^{i\theta}) = 2 - 2\cos\theta. \tag{87}$$

From standard results on linear recurrence, it follows that any solution to (83) has the form $\Psi_j = c_1(r_+)^j + c_2(r_-)^j$ for some constants $c_1, c_2 \in \mathbb{C}$. The fact that $\Psi_j$ must be real-valued implies that $c_1 = c_2 =: \alpha e^{i\phi}$ for some values $\alpha \in \mathbb{R}, \phi \in [0, 2\pi)$. Dropping the normalization constant (which we could recover later if needed), we obtain

$$\Psi_j = e^{i(\phi+j\theta)} + e^{-i(\phi+j\theta)} = 2\cos(\phi + j\theta). \tag{88}$$

Next, we use the boundary conditions to establish values for $\phi$ and $\theta$.

Equation (85) can be rewritten as

$$\cos(\phi + (n-2)\theta) = 2\cos(\theta)\cos((n-1)\theta) \tag{89}$$

which, using $2\cos(\alpha+\beta) = \cos(a+b) + \cos(a-b)$, simplifies to

$$0 = \cos(\phi + n\theta) \tag{90}$$

Consequently, $\phi + n\theta = \frac{1}{2}\pi + k\pi$ must hold for some $k \in \mathbb{N}$.

Equation (84) can be rewritten as

$$\cos(\phi + \theta) = (2\cos(\theta) - 1)\cos(\phi) \tag{91}$$

which simplifies to

$$\cos(\phi) = \cos(\theta - \phi) \tag{92}$$

One solution to this would be $\theta = 0$, but that would imply $\mu = 0$, which is inconsistent with $T$ being an invertible matrix. So instead, it must hold that $\phi = \frac{\theta}{2} + k\pi$ for some $k \in \mathbb{N}$.

Combining both conditions and solving for $\theta$ we obtain

$$\theta = \frac{\frac{1}{2} + k}{n + \frac{1}{2}}\pi = \frac{1}{2}\pi + k\pi \qquad \text{for some } k \in \mathbb{N}. \tag{93}$$

Each such value $\theta_k$ for $k \in \{0, \ldots, n-1\}$ yields an eigenvector with associated eigenvalue $\mu = 2 - 2\cos\theta_k = 4\sin^2(\theta_k/2)$. Now, (76) follows from this in combination with (77). $\qquad\square$

### F.5  Proof of Theorem 3

**Theorem 3** (Expected approximation error with single participation). *Let $A_{\alpha,\beta} \in \mathbb{R}^{n\times n}$ be the workload matrix (9) of SGD with momentum $0 \le \beta < 1$ and weight decay parameter $0 < \alpha \le 1$, where $\alpha > \beta$. Assume that each data item can contribute at most once to an update vector (e.g. single participation, $k = 1$). Then, the expected approximation error of the* square root factorization, $A_{\alpha,\beta} = C_{\alpha,\beta}^2$, *fulfills*

$$1 \le \mathcal{E}(C_{\alpha,\beta}, C_{\alpha,\beta}) \le \frac{1}{(\alpha-\beta)^2}\log\frac{1}{1-\alpha^2} \tag{11}$$

*for $\alpha < 1$, and*

$$\max\left\{1, \frac{\log(n+1) - 1}{4}\right\} \le \mathcal{E}(C_{1,\beta}, C_{1,\beta}) \le \frac{1 + \log(n)}{(1-\beta)^2}. \tag{12}$$

*Proof.* The proof consists of a combination of Lemmas 4 and 7. Because in the single participation $k = 1$, so we need just the first column of matrix $C_{\alpha,\beta}$:

$$\mathcal{E}(C_{\alpha,\beta}, C_{\alpha,\beta}) \le \sum_{i=0}^{n-1} c_i^2 \le \begin{cases} \dfrac{1 + \log n}{(1-\beta)^2} & \text{for } \alpha = 1, \\[2mm] \dfrac{1}{(\alpha-\beta)^2}\log\dfrac{1}{1-\alpha^2} & \text{otherwise.} \end{cases} \tag{94}$$

which proves the upper bounds. For the lower bounds, for any $\alpha \leq 1$:

$$\mathcal{E}(C_{\alpha,\beta}, C_{\alpha,\beta}) \geq \frac{1}{n}\sum_{j=0}^{n-1}\sum_{i=0}^{j} c_i^2 \geq \frac{1}{n}\sum_{j=0}^{n-1} c_0 = 1. \tag{95}$$

Also, for $\alpha = 1$:

$$\mathcal{E}(C_{1,\beta}, C_{1,\beta}) \geq \frac{1}{n}\sum_{j=1}^{n}\sum_{i=0}^{j-1} c_i^2 \geq \frac{1}{4n}\sum_{j=1}^{n} \log(j+1) = \frac{\log((n+1)!)}{4n} \geq \frac{\log(n+1)-1}{4}, \tag{96}$$

because $\log((n+1)!) \geq (n+1)\log(n+1) - n$. $\qquad\square$

### F.6 Proof of Theorem 5

**Theorem 5.** *Assume the setting of Theorem 3. Then, the baseline factorizations $A_{\alpha,\beta} = A_{\alpha,\beta} \cdot \mathrm{Id}$ and $A_{\alpha,\beta} = \mathrm{Id} \cdot A_{\alpha,\beta}$ fulfill, for $\alpha < 1$,*

$$\mathcal{E}(A_{\alpha,\beta}, \mathrm{Id}) = \frac{\sqrt{1+\alpha\beta}}{\sqrt{(1-\alpha\beta)(1-\alpha^2)(1-\beta^2)}} + o(1) \quad and \quad \mathcal{E}(A_{1,\beta}, \mathrm{Id}) \leq \frac{\sqrt{n}}{\sqrt{2}(1-\beta)} + o(\sqrt{n}) \tag{14}$$

$$\mathcal{E}(\mathrm{Id}, A_{\alpha,\beta}) = \frac{\sqrt{1+\alpha\beta}}{\sqrt{(1-\alpha\beta)(1-\alpha^2)(1-\beta^2)}} + o(1) \quad and \quad \mathcal{E}(\mathrm{Id}, A_{1,\beta}) \leq \frac{\sqrt{n}}{1-\beta} + o(\sqrt{n}). \tag{15}$$

*Proof.* For $\alpha = 1$, by Lemma 2 we have:

$$\mathrm{sens}_{1,b}^2(A_{1,\beta}) = \sum_{i=0}^{n-1} a_i^2 = \sum_{i=0}^{n-1}\left(\frac{1-\beta^{i+1}}{1-\beta}\right)^2 = \frac{n}{(1-\beta)^2} - \frac{2\beta}{(1-\beta)^2}\sum_{i=0}^{n-1}\beta^i + \frac{\beta^2}{(1-\beta)^2}\sum_{i=0}^{n-1}\beta^{2i} \tag{97}$$

$$= \frac{n}{(1-\beta)^2} - \frac{2\beta^{n+1}}{(1-\beta)^3} + \frac{\beta^{2n+2}}{(1-\beta)^2(1-\beta^2)} = \frac{n}{(1-\beta)^2}(1+o(1)). \tag{98}$$

For $\alpha < 1$:

$$\mathrm{sens}_{1,b}^2(A_{\alpha,\beta}) = \sum_{i=0}^{n-1} a_i^2 = \sum_{i=0}^{n-1} \frac{(\alpha^{i+1}-\beta^{i+1})^2}{(\alpha-\beta)^2} \tag{99}$$

$$= \frac{1}{\alpha-\beta}\left[\alpha^2\sum_{i=0}^{n-1}(\alpha^2)^i - 2\alpha\beta\sum_{i=0}^{n-1}(\alpha\beta)^i + \beta^2\sum_{i=0}^{n-1}(\beta^2)^i\right] \tag{100}$$

$$= \frac{1}{(\alpha-\beta)^2}\left[\frac{\alpha^2}{1-\alpha^2} - \frac{2\alpha\beta}{1-\alpha\beta} + \frac{\beta^2}{1-\beta^2}\right](1+o(1)) \tag{101}$$

$$= \frac{1+\alpha\beta}{(1-\alpha\beta)(1-\alpha^2)(1-\beta^2)}(1+o(1)). \tag{102}$$

Together with

$$\|A_\beta\|_F^2/n = \frac{1}{n}\sum_{j=0}^{n-1}(n-j)\left[\sum_{i=0}^{j}\beta^i\right]^2 = \frac{1}{n}\sum_{j=0}^{n-1}(n-j)\left[\frac{1-\beta^{j+1}}{1-\beta}\right]^2 \tag{103}$$

$$= \frac{1}{n(1-\beta)^2}\sum_{j=0}^{n-1}(n-j)(1-2\beta^{j+1}+\beta^{2j+2}) \tag{104}$$

$$= \frac{(n+1)}{2(1-\beta)^2} + O(1) = \frac{n}{2(1-\beta)^2}(1+o(1)) \tag{105}$$

as $\beta < 1$ and the sum $\sum_{j=0}^{n-1} j\beta^{j+1}$ is uniformly bounded by $\beta \sum_{j=0}^{\infty} j\beta^j = \frac{\beta^2}{(1-\beta)^2}$

For $\alpha < 1$:

$$\|A_{\alpha,\beta}\|_F^2/n = \frac{1}{n} \sum_{j=0}^{n-1} (n-j) \frac{(\alpha^{j+1} - \beta^{j+1})^2}{(\alpha - \beta)^2} = \sum_{j=0}^{n-1} \frac{(\alpha^{j+1} - \beta^{j+1})^2}{(\alpha - \beta)^2} + o(1) \quad (106)$$

$$= \frac{1 + \alpha\beta}{(1 - \alpha\beta)(1 - \alpha^2)(1 - \beta^2)}(1 + o(1)) \quad (107)$$

where the second equality due to the fact that we average over the sequence $jx^{j+1}$ which converges to 0 for $|x| < 1$. $\qquad\square$

### F.7 Proof of Theorem 6

**Theorem 6** (Approximation error of BSR). *Let $A_{\alpha,\beta} \in \mathbb{R}^{n \times n}$ be the workload matrix (9) of SGD with momentum $0 \le \beta < 1$ and weight decay $0 < \alpha \le 1$, with $\alpha > \beta$. Let $A_{\alpha,\beta} = B_{\alpha,\beta}^{|p|} C_{\alpha,\beta}^{|p|}$, be its banded square root factorization as in Definition 3. Then, for any $b \in \{1, \ldots, n\}$, $p \le b$, and $k \in \{1, \ldots, \frac{n}{b}\}$ it holds:*

$$\mathcal{E}(B_{\alpha,\beta}^{|p|}, C_{\alpha,\beta}^{|p|}) = \begin{cases} O_\beta\left(\sqrt{\dfrac{nk\log p}{p}}\right) + O_{\beta,p}(\sqrt{k}) & \text{for } \alpha = 1, \\[4mm] O_{\beta,p,\alpha}\left(\sqrt{k}\right) & \text{for } \alpha < 1. \end{cases} \quad (16)$$

*Proof.* Consider a Lower Triangular Toeplitz (LTT) matrix multiplication:

$$\begin{pmatrix} a_1 & 0 & \ldots & 0 \\ a_2 & a_1 & \ldots & 0 \\ \ldots & \ldots & \ldots & \ldots \\ a_n & a_{n-1} & \ldots & a_1 \end{pmatrix} \times \begin{pmatrix} b_1 & 0 & \ldots & 0 \\ b_2 & b_1 & \ldots & 0 \\ \ldots & \ldots & \ldots & \ldots \\ b_n & b_{n-1} & \ldots & b_1 \end{pmatrix} = \begin{pmatrix} c_1 & 0 & \ldots & 0 \\ c_2 & c_1 & \ldots & 0 \\ \ldots & \ldots & \ldots & \ldots \\ c_n & c_{n-1} & \ldots & c_1 \end{pmatrix}, \quad (108)$$

where $c_j = \sum_{i=1}^{j} a_i b_{n+1-i}$ so the LTT structure is preserved with multiplication that allows us to work with sequences and their convolutions rather than matrix multiplication. For instance, we would write the previous product in the form:

$$(a_1, \ldots, a_n) * (b_1, \ldots, b_n) = (c_1, \ldots, c_n). \quad (109)$$

The inverse of the Lower Triangular Toeplitz matrix remains a Lower Triangular Toeplitz (LTT) matrix because we can find a unique sequence $(c_1, \ldots, c_n)$ such that:

$$(c_1, \ldots, c_n) * (a_1, \ldots, a_n) = (1, 0, \ldots, 0) \quad (110)$$

$$c_j = -\frac{1}{a_1} \sum_{i=1}^{j-1} c_j a_{j+1-i}, \text{ and } c_1 = \frac{1}{a_1}, \quad (111)$$

with the restriction that $a_1 \ne 0$; otherwise, the original matrix was not invertible. We consider the banded square root factorization $A_{\alpha,\beta} = B_{\alpha,\beta}^{|p|} C_{\alpha,\beta}^{|p|}$ which is characterized by the following identity:

$$(b_0, \ldots, b_{n-1}) * (1, c_1, \ldots, c_{p-1}, 0, \ldots, 0) = (1, c_1, \ldots, c_{n-1}) * (1, c_1, \ldots, c_{n-1}). \quad (112)$$

We will bound the Frobenius norm of the LTT matrix $(b_0, \ldots, b_{n-1})$. By the uniqueness of the solution, we obtain that for the first $p$ values we have $b_i = c_i$. For the next $p$ values we have the following formula:

$$b_{p+j} + \cdots + b_p c_j + \cdots + b_{j+1} c_{p-1} = c_{p+j} + \cdots + c_{p+1} c_{p-1} + \cdots + c_{p+j} \tag{113}$$

$$b_{p+j} + b_{p+j-1} c_1 + \cdots + b_p c_j = 2 \left( c_{p+j} + \cdots + c_p c_j \right). \tag{114}$$

By induction argument, we can see that $b_{p+j} = 2c_{p+j}$ for $0 \le j \le p - 1$. For the remaining $n - 2p$ values we will prove convergence to a constant.

$$\sum_{j=0}^{p-1} b_{j-i} c_i = a_j = \frac{\alpha^{j+1} - \beta^{j+1}}{\alpha - \beta}. \tag{115}$$

We make an ansatz for the solution of the linear recurrence in the form:

$$b_j = \frac{\alpha^{j+1}}{(\alpha - \beta) \sum_{i=0}^{p-1} c_i \alpha^{-i}} - \frac{\beta^{j+1}}{(\alpha - \beta) \sum_{i=0}^{p-1} c_i \beta^{-i}} + \alpha^j y_j, \tag{116}$$

where $y_j$ represents the error terms, which will be proven to converge to $0$. The sequence $y_j$ satisfies the following recurrence formula:

$$y_j = - \sum_{i=1}^{p-1} y_{j-i} c_i \alpha^{-i}. \tag{117}$$

We denote $w_j = c_j \alpha^{-j}$ which is a decreasing sequence because the values correspond to the $C_{1,\beta/\alpha}$ matrix. We rewrite the recurrence in matrix notation:

$$\begin{pmatrix} -w_1 & -w_2 & -w_3 & \cdots & -w_{p-2} & -w_{p-1} \\ 1 & 0 & 0 & \cdots & 0 & 0 \\ 0 & 1 & 0 & \cdots & 0 & 0 \\ 0 & 0 & 1 & \cdots & 0 & 0 \\ \cdots & \cdots & \cdots & \cdots & \cdots & \cdots \\ 0 & 0 & 0 & \cdots & 1 & 0 \end{pmatrix} \begin{pmatrix} y_{k-1} \\ y_{k-2} \\ y_{k-3} \\ y_{k-4} \\ \cdots \\ y_{k-b} \end{pmatrix} = \begin{pmatrix} y_k \\ y_{k-1} \\ y_{k-2} \\ y_{k-3} \\ \cdots \\ y_{k-b+1} \end{pmatrix}. \tag{118}$$

To show that the error terms $y_j$ goes to $0$ as $j$ goes to infinity, we first study the characteristic polynomial of the associate homogeneous relations:

$$g(\lambda) = \lambda^{p-1} + w_1 \lambda^{p-2} + \cdots + w_{p-2} \lambda + w_{p-1}. \tag{119}$$

Because $1 > w_1 > w_2 > \cdots > w_{b-1} > 0$, it follows from *Schur's (relaxed) stability condition* [Nguyen et al., 2007, Theorem 1] that all its (complex) roots lie inside of the open unit circle. Therefore, all solutions to the homogeneous relation converge to zero at a rate exponential in $j$ and $y_j = o(1)$ and $\sum_{j=0}^{\infty} y_j^2 = O_{\alpha,\beta,p}(1)$. Then we can bound the Frobenious norm of the matrix $B_{\alpha,\beta}^{|p|}$ as:

$$\frac{1}{n} \|B_{\alpha,\beta}^{|p|}\|_F^2 \le \sum_{j=0}^{n-1} b_j^2 \le \sum_{j=0}^{p-1} c_j^2 + \sum_{j=p}^{n-1} \frac{\alpha^{2j+2}}{(\alpha - \beta)^2 \left[ \sum_{i=0}^{p-1} w_i \right]^2} + \alpha^{2j} y_j^2. \tag{120}$$

We use the following lower bound for the sum of $w_j$:

$$\sum_{j=0}^{p-1} w_j \ge \frac{1}{2} \sum_{j=0}^{p-1} \frac{1}{\sqrt{j+1}} \ge \sqrt{p+1} - 1. \tag{121}$$

Combining these bounds we can upper bound the Frobenious norm of the matrix $B_{\alpha,\beta}^{|p|}$ the following way:

$$\|B_{\alpha,\beta}^{|p|}\|_F^2/n \leq \begin{cases} \dfrac{1}{(\alpha-\beta)^2} \log\left(\dfrac{1}{1-\alpha^2}\right) + \dfrac{\alpha^2}{(\sqrt{p+1}-1)^2(\alpha-\beta)^2} + O_{\alpha,\beta,p}(1) & \text{for } \alpha < 1 \\ \dfrac{1+\log(p)}{(1-\beta)^2} + \dfrac{n-p}{(1-\beta)^2(\sqrt{p+1}-1)^2} + O_{p,\beta}(1) & \text{for } \alpha = 1. \end{cases}$$

(122)

Simplifying for the leading terms in asymptotics, we have:

$$\|B_{\alpha,\beta}^{|p|}\|_F^2/n = \begin{cases} O_{\alpha,\beta,p}(1) & \text{for } \alpha < 1 \\ O_\beta\left(\dfrac{n}{p}\right) + O_{p,\beta}(1) & \text{for } \alpha = 1. \end{cases}$$

(123)

**Sensitivity of $C_\beta^{|p|}$.** For the $b$-min-separation participation sensitivity we have the following bound for any $p \leq b$:

$$\text{sens}_{k,b}^2(C_{\alpha,\beta}^{|p|}) \leq k \sum_{j=0}^{p-1} c_j^2 \leq \begin{cases} \dfrac{k}{(\alpha-\beta)^2} \log\left(\dfrac{1}{1-\alpha^2}\right) & \text{for } \alpha < 1 \\ k\dfrac{1+\log(p)}{(1-\beta)^2} & \text{for } \alpha = 1. \end{cases}$$

(124)

Combining sensitivity with the upper bound for the Frobenious norm we obtain:

$$\mathcal{E}(B_{\alpha,\beta}^{|p|}, C_{\alpha,\beta}^{|p|}) = \begin{cases} O_{p,\alpha,\beta}(\sqrt{k}) & \text{for } \alpha < 1 \\ O_\beta\left(\sqrt{\dfrac{nk\log p}{p}}\right) + O_{\beta,p}(\sqrt{k}) & \text{for } \alpha = 1. \end{cases}$$

(125)

$\square$

### F.8 Proof of Theorem 7 for Square Root Factorization

**Theorem 7** (Approximation error of Square Root Factorization). *Let $A_{\alpha,\beta} \in \mathbb{R}^{n \times n}$ be the workload matrix (9) of SGD with momentum $0 \leq \beta < 1$ and weight decay $0 < \alpha \leq 1$, with $\alpha > \beta$. Let $A_{\alpha,\beta} = C_{\alpha,\beta}^2$ be its square root factorization. Then, for any $b \in \{1, \ldots, n\}$ and $k = \frac{n}{b}$ it holds:*

$$\mathcal{E}(C_{\alpha,\beta}, C_{\alpha,\beta}) = \begin{cases} \Theta_\beta\left(k\sqrt{\log n} + \sqrt{k}\log n\right) & \text{for } \alpha = 1, \\ \Theta_{\alpha,\beta}(\sqrt{k}) & \text{for } \alpha < 1. \end{cases}$$

(17)

*Proof.* We prove the case without weight decay ($\alpha = 1$) and with weight decay ($\alpha < 1$) separately.

**Case 1) no weight decay ($\alpha = 1$).**

We start by bounding the $b$-min-separation sensitivity:

$$\text{sens}_{k,b}^2(C_{1,\beta}) = \sum_{i=0}^{k-1}\sum_{j=0}^{k-1} \langle (C_{1,\beta})_{[\cdot,ib]}, (C_{1,\beta})_{[\cdot,jb]} \rangle.$$

(126)

Consider a scalar product for a general pair of indices, $j > i$:

$$\langle (C_{1,\beta})_{[\cdot,i]}, (C_{1,\beta})_{[\cdot,j]} \rangle = \sum_{t=0}^{n-1-j} c_t c_{j-i+t}.$$

(127)

Using the bounds on $c_k$ (6) for $\alpha = 1$ we can lower and upper bound this sum by:

$$\sum_{t=0}^{n-1-j} c_t c_{j-i+t} \leq \frac{1}{(1-\beta)^2} \sum_{t=1}^{n-j} \frac{1}{\sqrt{t(j-i+t)}} \leq \frac{1}{(1-\beta)^2} \int_0^{n-j} \frac{dx}{\sqrt{x(j-i+x)}} \tag{128}$$

$$\sum_{t=0}^{n-1-j} c_t c_{j-i+t} \geq \frac{1}{4} \sum_{t=1}^{n-j} \frac{1}{\sqrt{t(j-i+t)}} \geq \frac{1}{4} \int_1^{n-j} \frac{dx}{\sqrt{x(j-i+x)}}. \tag{129}$$

We can compute the indefinite integral explicitly:

$$\int \frac{dx}{\sqrt{x(j-i+x)}} = F\left(\frac{j-i}{x}\right) + C \tag{130}$$

for $F(a) = 2\log\left(\sqrt{\frac{1}{a}+1} + \sqrt{\frac{1}{a}}\right)$. In combination, we obtain the upper and lower bound for (126):

$$\frac{1}{4} f\left(\frac{j-i}{n-j}\right) - \frac{1}{4} f(j-i) \leq \langle (C_\beta)_i, (C_\beta)_j \rangle \leq \frac{1}{(1-\beta)^2} f\left(\frac{j-i}{n-j}\right). \tag{131}$$

Now we are ready to bound the sensitivity of the matrix $C_{1,\beta}$:

$$\mathrm{sens}_{k,b}^2(C_{1,\beta}) = \sum_{i=0}^{k-1} \langle (C_{1,\beta})_{ib}, (C_{1,\beta})_{ib} \rangle + 2 \sum_{i=0}^{k-1} \sum_{j=i+1}^{k-1} \langle (C_{1,\beta})_{ib}, (C_{1,\beta})_{jb} \rangle \tag{132}$$

$$\leq \frac{1}{(1-\beta)^2} \sum_{i=0}^{k-1} (\log(n-ib)+1) + \frac{2}{(1-\beta)^2} \sum_{i=0}^{k-1} \sum_{j=i+1}^{k-1} f\left(\frac{j-i}{k-j}\right) \tag{133}$$

and, analogously

$$\mathrm{sens}_{k,b}^2(C_{1,\beta}) \geq \frac{1}{4} \sum_{i=0}^{k-1} \log(n-ib) + \frac{1}{2} \sum_{0 \leq i < j \leq k-1} \left[ f\left(\frac{j-i}{k-j}\right) - f(b(j-i)) \right] \tag{134}$$

Firstly, using $(\frac{k}{e})^k \leq k! \leq k^k$, we bound the sum of the logarithms:

$$\sum_{j=0}^{k-1} \log(n-jb) = \sum_{j=1}^{k} [\log b + \log j] = k\log b + \log k! \leq k\log b + k\log k = k\log n, \tag{135}$$

$$\sum_{j=0}^{k-1} \log(n-jb) = k\log b + \log k! \geq k\log b + k\log k - k = k\log n - k. \tag{136}$$

To upper bound the last term in sensitivity lower bound (134), we use the auxiliary inequality $f(a) = 2\log\left(\sqrt{\frac{1}{a}+1} + \sqrt{\frac{1}{a}}\right) \leq \frac{4}{\sqrt{a}}$ to derive:

$$\frac{1}{2} \sum_{0 \leq i < j \leq k-1} f(b(j-i)) \leq \frac{2}{\sqrt{b}} \sum_{0 \leq i < j \leq k-1} \frac{1}{\sqrt{j-i}} = \frac{2}{\sqrt{b}} \sum_{j=1}^{k-1} \sum_{t=1}^{j} \frac{1}{\sqrt{t}} \leq \frac{4}{\sqrt{b}} \sum_{j=1}^{k-1} \sqrt{j} \leq \frac{8}{3\sqrt{b}} k^{3/2} \tag{137}$$

To bound the final term we establish the following inequalities for $f(a)$:

$$f(a) = 2\log\left(\sqrt{\frac{1}{a}+1} + \sqrt{\frac{1}{a}}\right) = \log\left(\frac{1}{a}+1\right) + 2\log\left(1 + \frac{1}{\sqrt{a+1}}\right) \tag{138}$$

$$\log\left(\frac{1}{a}+1\right) < f(a) < \log\left(\frac{1}{a}+1\right) + 2\log 2 \tag{139}$$

Then we can bound the first double sum in (134) as

$$\sum_{0\le i<j\le k-1} \log\left(\frac{k-i}{j-i}\right) \le \sum_{0\le i<j\le k-1} f\left(\frac{j-i}{k-j}\right) \le \sum_{0\le i<j\le k-1} \log\left(\frac{k-i}{j-i}\right) + 2k^2 \log 2. \tag{140}$$

To bound the term $\sum_{0\le i<j\le k-1} \log\left(\frac{k-i}{j-i}\right)$ we use the following identities:

$$\sum_{0\le i<j\le k-1} \log\left(\frac{k-i}{j-i}\right) = \log \prod_{i=0}^{k-2} \frac{(k-i)^{k-i-1}}{(k-i-1)!} \tag{141}$$

$$= \log \frac{k^{k-1}(k-1)^{k-2}\dots 2^1}{1!\cdot 2!\cdot 3!\dots(k-1)!} = \log \frac{2^1\cdot 3^2\cdot 4^3\dots k^{k-1}}{1^{k-1}\cdot 2^{k-2}\dots(k-1)^1} \tag{142}$$

$$= \log \prod_{j=1}^{k-1}\left(\frac{j+1}{k-j}\right)^j = \sum_{j=1}^{k-1} j\log(j+1) - \sum_{j=1}^{k-1} j\log(k-j) \tag{143}$$

$$= \sum_{j=1}^{k}(j-1)\log(j) - \sum_{j=1}^{k-1}(k-j)\log(j) \tag{144}$$

$$= 2\sum_{j=1}^{k-1} j\log(j) - \log k! + 2k\log k - k\log k! \tag{145}$$

Now, using that $x\log x$ is a monotonically increasing function,

$$\sum \sum_{0\le i<j\le k-1} \log\left(\frac{k-i}{j-i}\right) \le 2\int_1^k x\log x\, dx + k\log k + k - k^2\log k + k^2 \tag{146}$$

$$= k^2\log k - \frac{k^2}{2} + k\log k - k - k^2\log k + k^2 \tag{147}$$

$$\le \frac{3}{2}k^2 \tag{148}$$

As a lower bound, we obtain

$$\sum_{0\le i<j\le k-1} \log\left(\frac{k-i}{j-i}\right) \ge 2\int_1^{k-1} x\log x\, dx - k\log k + k\log k - k(k-1)\log(k-1) \tag{149}$$

$$= (k-1)^2\log(k-1) - \frac{(k-1)^2}{2} + k\log k - k(k-1)\log(k-1) \tag{150}$$

$$= -(k-1)\log(k-1) - \frac{(k-1)^2}{2} + k\log k \tag{151}$$

$$\ge -\frac{k^2}{2} \tag{152}$$

Therefore, combining the upper bound (146) and the lower bound (149) yields

$$\sum_{0\le i<j\le k-1} f\left(\frac{j-i}{k-j}\right) \le (2\log 2 + 3/2)k^2 \le 3k^2, \tag{153}$$

$$\sum_{0\le i<j\le k-1} f\left(\frac{j-i}{k-j}\right) \ge (2\log 2 - 1/2)k^2 \ge \frac{4k^2}{5}. \tag{154}$$

Combining all three terms together we obtain the following bounds for the squared sensitivity (134):

$$\mathrm{sens}_{k,b}^2(C_{1,\beta}) \leq \frac{k}{(1-\beta)^2}(\log n + 1) + \frac{6}{(1-\beta)^2}k^2 \tag{155}$$

$$\mathrm{sens}_{k,b}^2(C_{1,\beta}) \geq \frac{k}{4}(\log n - 1) - \frac{8}{3\sqrt{b}}k^{3/2} + \frac{2}{5}k^2 \tag{156}$$

Now, we recall the bounds for the Frobenius norm of the matrix $C_{1,\beta}$ 7 and (96):

$$\frac{\log(n+1) - 1}{4} \leq \|C_{1,\beta}\|_F^2/n \leq \frac{\log n + 1}{(1-\beta)^2}. \tag{157}$$

With the auxiliary inequality $\sqrt{\frac{a}{2}} + \sqrt{\frac{b}{2}} \leq \sqrt{a+b} \leq \sqrt{a} + \sqrt{b}$ and combining (155) and the bounds on Frobenius norm (7) and (96) we get that:

$$\mathcal{E}(C_{1,\beta}, C_{1,\beta}) \leq \frac{\sqrt{k}}{(1-\beta)^2}(\log n + 1) + \frac{\sqrt{5}k}{(1-\beta)^2}\sqrt{\log n + 1} \tag{158}$$

$$\mathcal{E}(C_{1,\beta}, C_{1,\beta}) \geq \frac{1}{4\sqrt{2}}\sqrt{k}(\log n - 1) + \frac{k}{2\sqrt{5}}\sqrt{\log(n) - 1}\sqrt{1 - \frac{20}{3\sqrt{n}}} \tag{159}$$

Making the lower bound well-defined requires $n \geq 45$, otherwise one can simply take $\mathcal{E}(C_{1,\beta}, C_{1,\beta}) \geq 1$. As a final step, we combine both inequalities in the following asymptotic statement:

$$\mathcal{E}(C_{1,\beta}, C_{1,\beta}) = \Theta_\beta\left(\sqrt{k}\log n + k\sqrt{\log n}\right), \tag{160}$$

which concludes the proof of the case without weight decay.

**Case 2) with weight decay ($\alpha < 1$).**

As above, we first express the $b$-min-separation sensitivity of the matrix $C_{\alpha,\beta}$ in terms of inner products,

$$\mathrm{sens}_{k,b}^2(C_{\alpha,\beta}) = \sum_{i=0}^{k-1}\sum_{j=0}^{k-1}\langle(C_{\alpha,\beta})_{ib}, (C_{\alpha,\beta})_{jb}\rangle. \tag{161}$$

and then consider a scalar product for a general pair of indexes $j > i$:

$$\langle(C_{\alpha,\beta})_i, (C_{\alpha,\beta})_j\rangle = \sum_{t=0}^{n-1-j} c_t c_{j-i+t}. \tag{162}$$

Now, we use the bounds on $c_t$ from Lemma 6 for $\alpha < 1$, to upper and lower bound this sum with the following expression, where $\gamma = \frac{\beta}{\alpha}$:

$$\langle(C_{\alpha,\beta})_i, (C_{\alpha,\beta})_j\rangle \leq \frac{\alpha^{j-i}}{(1-\gamma)^2}\sum_{t=0}^{n-1-j}\frac{\alpha^{2t}}{\sqrt{(t+1)(j-i+t+1)}} \leq \frac{\alpha^{j-i}}{(1-\gamma)^2(1-\alpha^2)\sqrt{j-i}} \tag{163}$$

$$\langle(C_{\alpha,\beta})_i, (C_{\alpha,\beta})_j\rangle \geq \frac{\alpha^{j-i}}{4}\sum_{t=0}^{n-1-j}\frac{\alpha^{2t}}{\sqrt{(t+1)(j-i+t+1)}} \geq \frac{\alpha^{j-i}}{4\sqrt{j-i+1}}. \tag{164}$$

We substitute these bounds into Equation (161) to obtain the following upper bound for sensitivity of matrix $C_{\alpha,\beta}$:

$$\text{sens}_{k,b}^2(C_{\alpha,\beta}) \leq \sum_{i=0}^{k-1} \langle (C_{\alpha,\beta})_{ib}, (C_{\alpha,\beta})_{ib} \rangle + \frac{2}{(1-\gamma)^2(1-\alpha^2)\sqrt{b}} \sum_{j>i} \frac{\alpha^{b(j-i)}}{\sqrt{j-i}} \quad (165)$$

$$\leq \frac{k}{(\alpha-\beta)^2} \log \frac{1}{1-\alpha^2} + \frac{2}{(1-\gamma)^2(1-\alpha^2)\sqrt{b}} \sum_{j>i} \alpha^{b(j-i)} \quad (166)$$

$$\leq \frac{k}{(\alpha-\beta)^2} \log \frac{1}{1-\alpha^2} + \frac{2k\alpha^b}{(1-\gamma)^2(1-\alpha^2)(1-\alpha^b)\sqrt{b}}, \quad (167)$$

where the second inequality is due to Equation (95).

A lower bound for the sensitivity follows directly from Lemma 3:

$$\text{sens}_{k,b}^2(C_{\alpha,\beta}) \geq k\|C_{\alpha,\beta}\|_F \geq k. \quad (168)$$

The Frobenius norm of the matrix $C_{\alpha,\beta}$ is the same as that for one round participation; thus, we could reuse Inequalities (95):

$$1 \leq \|C_{\alpha,\beta}\|_F^2/n \leq \frac{1}{(\alpha-\beta)^2} \log \frac{1}{1-\alpha^2}. \quad (169)$$

By merging the bounds for sensitivity and Frobenius norm, we derive the following bounds for error:

$$\mathcal{E}(C_{\alpha,\beta}, C_{\alpha,\beta}) \leq \sqrt{k} \left[ \frac{1}{(\alpha-\beta)^2} \log \frac{1}{1-\alpha^2} + \frac{2\alpha^b}{(1-\gamma)^2(1-\alpha^2)(1-\alpha^b)\sqrt{b}} \right] \quad (170)$$

$$\mathcal{E}(C_{\alpha,\beta}, C_{\alpha,\beta}) \geq \sqrt{k}. \quad (171)$$

The combination of these results yields the following asymptotic statement:

$$\mathcal{E}(C_{\alpha,\beta}, C_{\alpha,\beta}) = \Theta_{\alpha,\beta}(\sqrt{k}), \quad (172)$$

which concludes the proof. $\qquad\square$

### F.9 Proof of Theorem 8

**Theorem 8.** *Assume the setting of Theorem 6. Then, for any factorization $A_{\alpha,\beta} = BC$ with $C^\top C \geq 0$, the approximation error fulfills*

$$\mathcal{E}(B,C) \geq \begin{cases} \sqrt{k} \log n & \text{for } \alpha = 1, \\ \sqrt{k} & \text{for } \alpha < 1, \end{cases} \quad (18)$$

*Proof.* Let $A_{\alpha,\beta} = BC$ be any factorization with $CC^\top \geq 0$. From Lemma 3 it follows that

$$\mathcal{E}(B,C) = \frac{1}{\sqrt{n}}\|B\|_F \text{sens}_{k,b}(C) \geq \frac{\sqrt{k}}{n}\|B\|_F\|C\|_F \geq \frac{\sqrt{k}}{n}\|A_{\alpha,\beta}\|_*, \quad (173)$$

where $\|\cdot\|_*$ denotes the *nuclear norm*, and the last inequality follows from its variational form, $\|M\|_* = \min_{\{X,Y:XY^T=M\}} \|X\|_F\|Y\|_F$. The statement of the Theorem follows by inserting the corresponding bounds on $\|A_{\alpha,\beta}\|_*$ from Lemma 8. $\qquad\square$

### F.10 Proof of Theorem 9

**Theorem 9.** *Assume the setting of Theorem 6. Then, the baseline factorizations $A_{\alpha,\beta} = A_{\alpha,\beta} \cdot \text{Id}$ and $A_{\alpha,\beta} = \text{Id} \cdot A_{\alpha,\beta}$ fulfill*

$$\mathcal{E}(A_{\alpha,\beta}, \text{Id}) \geq \begin{cases} \sqrt{\dfrac{nk}{2}} & \text{for } \alpha = 1, \\ \sqrt{k} & \text{for } \alpha < 1. \end{cases} \qquad \mathcal{E}(\text{Id}, A_{\alpha,\beta}) \geq \begin{cases} \dfrac{k\sqrt{n}}{\sqrt{3}} & \text{for } \alpha = 1, \\ \sqrt{k} & \text{for } \alpha < 1. \end{cases} \quad (19)$$

*Proof.* **Case 1)** $A_{\alpha,\beta} = BC$ **with** $B = A_{\alpha,\beta}$ **and** $C = \mathrm{Id}$. It is easy to check that $\mathrm{sens}_{k,b}(C) = \sqrt{k}$, so $\mathcal{E}(B,C) = \sqrt{\frac{k}{n}}\|A_{\alpha,\beta}\|_F$. Because $A_{\alpha,0} \le A_{\alpha,\beta} \le \frac{1}{\alpha-\beta}A_{\alpha,0}$ componentwise, we have for $\alpha = 1$,

$$\frac{n(n+1)}{2} = \|A_{1,0}\|_F^2 \le \|A_{1,\beta}\|_F^2 \le \frac{1}{(1-\beta)^2}\|A_{1,0}\|_F^2 = \frac{n(n+1)}{2(1-\beta)^2}, \tag{174}$$

which implies the corresponding statement of the theorem. For $0 < \alpha < 1$, we use that $A_{1,0} > \mathrm{Id}$ componentwise, so $\|A_{\alpha,0}\|_F^2 > \|\mathrm{Id}\|_F^2 = n$, which conclude the proof of this case.

**Case 2)** $A_{\alpha,\beta} = BC$ **with** $B = \mathrm{Id}$ **and** $C = A_{\alpha,\beta}$. We observe that $\|\mathrm{Id}\|_F = \sqrt{n}$, so

$$\mathcal{E}(B,C) = \mathrm{sens}_{k,b}(A_{\alpha,\beta}). \tag{175}$$

Again, we use the fact that $A_{\alpha,0} \le A_{\alpha,\beta} \le \frac{1}{\alpha-\beta}A_{\alpha,0}$. Now $A_{\alpha,0}$ fulfills the conditions of Theorem 2, so from Equation (10) we know

$$\mathrm{sens}_{k,b}(A_{\alpha,0}) = \Big\|\sum_{j=0}^{k-1}(A_{\alpha,0})_{[\cdot,1+jb]}\Big\|, \tag{176}$$

We first study (176) for $\alpha = 1$. Then, from the explicit structure of $A_{1,0} = \mathrm{LDToep}(1,1,\ldots,1)$ one sees that the vectors inside the norm have a block structure

$$\sum_{j=0}^{k-1}(A_{1,0})_{[\cdot,1+jb]} = \begin{pmatrix} v_1 \\ \vdots \\ v_k \\ v' \end{pmatrix} \qquad \text{with} \qquad v_i = \begin{pmatrix} i \\ \vdots \\ i \end{pmatrix} \in \mathbb{R}^b \tag{177}$$

for $i = 1,\ldots,k$, and $v' = \begin{pmatrix} k \\ \vdots \\ k \end{pmatrix} \in \mathbb{R}^{n-bk}$, appears only if $k < \frac{n}{b}$. Now we check

$$\|v'\|^2 + \sum_{i=0}^{k}\|v_i\|_2^2 = (n-bk)k^2 + b\Big(\sum_{i=1}^{k}i^2\Big) \tag{178}$$

$$= nk^2 - bk^3 + b\frac{k(k+1)(2k+1)}{6} \tag{179}$$

$$\ge nk^2 - \frac{2}{3}bk^3 \ge \frac{1}{3}nk^2, \tag{180}$$

because $bk \ge n$. Consequently

$$\mathrm{sens}_{k,b}(A_{1,\beta}) \ge \frac{k\sqrt{n}}{\sqrt{3}}, \tag{181}$$

which concludes the proof of this case. For $\alpha < 1$, $A_{\alpha,\beta} \ge \mathrm{Id}$ componentwise readily implies

$$\mathrm{sens}_{k,b}(A_{\alpha,\beta}) \ge \sqrt{k}, \tag{182}$$

which implies the statement of the theorem. $\qquad\square$

