# OpenReview forum: "Banded Square Root Matrix Factorization for Differentially Private Model Training"
_NeurIPS.cc/2024/Conference — NeurIPS 2024 poster_

### Official Review · Reviewer_VxsQ · 2024-07-07

**Soundness:** 3
**Presentation:** 3
**Contribution:** 3
**Rating:** 6
**Confidence:** 4

**Summary:**

The paper proposes the Banded Square Root (BSR) matrix factorization to speed up banded matrix factorization. The authors demonstrate that the workload matrix for SGD with momentum and weight decay can be expressed as a lower triangular Toeplitz matrix. They utilize explicit recursive expressions to compute the square root of this workload matrix, and Theorem 1 provides a closed-form expression for the square root matrix. Theoretical analysis shows that the calculation of the square root matrix is efficient, and the sensitivity of the mechanism is discussed. Detailed analysis of the approximation error is also provided, demonstrating that the square root factorization has asymptotically optimal approximation quality. Experiments show that BSR can achieve performance comparable to previous state-of-the-art methods.

**Strengths:**

1. The discussion regarding SGD with momentum and weight decay is interesting. The workload matrix in this context is a lower triangular Toeplitz matrix, which allows for efficient factorization algorithms. Prior work has not explored this setting; they typically treat the workload matrix with momentum as a general lower triangular matrix. This finding is significant for future research, suggesting potential improvements over current solvers for banded matrix factorization.

2. The theoretical analysis demonstrates that Banded Square Root (BSR) matrix factorization can be computed efficiently even for large problem sizes. The expected approximation error for BSR indicates that it achieves asymptotically optimal approximation quality. These analyses ensure that BSR is both fast and accurate.

**Weaknesses:**

1. Using CVXPY with SCS to compute AOF is not efficient; L-BFGS can solve this problem more efficiently. For instance, as shown in [1], an implementation using GPUs can solve a problem size of 10,000 in less than an hour. Even with CPUs, the implementation in [2] can solve a problem size of 2,000 in less than an hour, with slight modifications to add the b-min-step constraint.

2. The current experiment only demonstrates results for matrix sizes up to 2000, which is insufficient. A comparison involving matrix sizes of 10,000 or more would be more convincing. How long would it take to obtain a solution for problem sizes of 10,000 and 100,000 using BSR?

[1] https://github.com/apple/pfl-research/blob/2a9d65bd66dc89ef80a24e1be0d28446a0422133/pfl/privacy/ftrl_mechanism.py#L140

[2] https://github.com/dpcomp-org/hdmm/blob/7a5079a7d4f1a06b0be78019adadf83c538d0514/src/hdmm/templates.py#L391

**Questions:**

1. Enforcing positive semi-definiteness (PSD) for the matrix is crucial for convergence in AOF. In line 274, you mentioned a post-processing step for S to ensure that all its eigenvalues are at least $\sqrt{1/n}$. Could you provide a more detailed description of this step? I couldn't find it in the source code provided in Appendix A.

2. In equation (6), b should be p, right?

**Limitations:**

1. If the learning rate is not constant, then the workload matrix will not be a Toeplitz matrix, right? This would significantly impact the efficiency of BSR. During the training process, a proper learning rate schedule can enhance convergence. Therefore, a discussion about varying learning rates should be included.

2. The paper does not provide a general introduction to differential privacy, and Figure 2 presents (epsilon, delta) without sufficient background information. This lack of context makes the paper not self-contained.

---

> ### Author Rebuttal · Authors · 2024-08-07
>
> Thank you for the encouraging feedback. In the following we hope to address all your concerns.
>
> **\> Using CVXPY with SCS to compute AOF is not efficient.**
>
> There are, of course, many ways to solve AOF (4). We used SCS, because it is a dedicated SPD-solver which allows for a simple and reproducible implementation without additional hyperparameters (Algorithm 2). General-purpose gradient-based solvers, such as L-BFGS, were meant to be included in our comment about approximate solvers (lines 122-125). These can be faster for solving (4) up to a certain precision, but they require additional design choices, such as the number of update steps, learning rates, a barrier function to enforce the constraints, and potentially projections to ensure positive definiteness.
> Importantly, however, note that our claim in the work is not that “BSR is x-times faster than AOF”. Actual runtimes always depend on many factors, including hardware, programming language, software packages, etc. We mention runtimes rather as a qualitative guidance to the reader, which is why we do not report a formal table or plot of runtimes. Our core argument is that (4) is a challenging optimization problem to solve. *Any* numeric solver will be slower and less convenient to use than our $O(n \log n)$ closed-form expressions. In real-world problems solving (4) creates a bottleneck, especially because it has to be solved anew for any change of hyperparameters. BSR overcomes this bottleneck.
>
> **\> References to larger-scale matrix factorizations.**
>
> Thank you for the references; we will include them into our manuscript. Ref [1] indeed aims to solve the optimization problem (4). Could you please provide a reference for the timing results you mentioned? We did not spot them in the repository or associated report arXiv:2404.06430, but would be happy to adapt our presentation accordingly. [2] solves a simpler problem (from [McKenna et al., VLDB 2018]). Although adapting the formulation might be possible, the additional constraints could complicate the numerical solution. Nevertheless, we will be happy to include this in our discussion.
>
> **\> The current experiment only demonstrates results for matrix sizes up to 2000, which is insufficient**
>
> We report experiments to a size of 2000 due to the computational limits of calculating AOF. For BSR, much larger workload sizes are no problem, and we will be happy to add them to the manuscript. The speed advantage of BSR over AOF will increase with $n$, regardless of the solver. We illustrate this in the response PDF: a plain python implementation of Theorem 1 (Alg.3 in PDF) computes the BSR-coefficients for $n=10,000$ in 24ms , for $n=100,000$ in 2.4s and for $n=500,000$ in ~67s (Table 1 in PDF, single-core CPU, double precision). In practice, one would likely not use vanilla MF-SGD for $n \ge 500,000$, because the resulting matrices (e.g. $C^{-1}$) would be inconveniently large (1 TB of memory in dense fp32 form).
>
> **\> Enforcing positive semi-definiteness (PSD).**
>
> We did not include this step in Algorithm 2, because we considered it a post-processing operation only needed when trying to extract $C$ from the solution matrix $S$ (which despite the use of SCS, is not always perfectly PSD). We will be happy to add it, though. The actual operation is a standard procedure (Alg.4 in the response PDF). Our experiments use $tol=1/\sqrt{n}$ as a heuristic, because we observed it to give good results across a wide range of problem sizes, in particular $C^{-1}$ remaining stable. Constant values for tol or values proportional to $1/n$ did not work as well. We did not find this issue discussed in the literature.
>
> **\> Equation (6).**
>
> Thanks for spotting this typo. We will fix it.
>
> **\> Different learning rates.**
>
> Indeed, different learning rates would result in a non-Toeplitz workload matrix, so our analytical expressions no longer apply. However, the banded-square-root method remains applicable, only that the matrix square root has to be computed numerically. This step remains quite efficient because the workload matrix is still lower triangular, for which specialized routines exist, e.g. [Björck& Hammarling. “A Schur method for the square root of a matrix”, 1983]. We will include a discussion of the impact of varying learning rates in the revised version.
>
> **\> General introduction to differential privacy.**
>
> Indeed, due to space constraints, we were not able to give an introduction in the manuscript, which primarily targets an audience already familiar with DP. We will add a section in the appendix.

---

> > ### Comment · Reviewer_VxsQ · 2024-08-11
> >
> > Thank you for addressing my concerns. I have raised my rating.
> >
> > The runtime was observed when I did several experiments. I really encourage to have a comparison experiment on the latest algorithm. Although I agree that the main purpose of the paper is not for runtime comparison.

---

### Official Review · Reviewer_1wZc · 2024-07-09

**Soundness:** 4
**Presentation:** 4
**Contribution:** 3
**Rating:** 7
**Confidence:** 5

**Summary:**

This paper proposes the Banded Square Root Matrix Factorization, an efficient approximation of the optimal banded matrix factorization for differentially private machine learning applications.  The authors give a closed form expression for the BSR C matrix for any SGD + Momentum + Weight Decay workload by exploiting the Toepltiz structure of the workload and factors.  These factorizations can be used in place of the prior work and enjoy some efficiency advantages.

**Strengths:**

* The scope, contributions, and key results are clearly written.
* Near-optimal and efficiency computable banded matrix factorizations could make it feasible to use this mechanism in some new settings where it was previously not possible.
* Authors demonstrated deep understanding of the area, and made a compelling story with a strong mix of theoretical derivations and empirical observations.

**Weaknesses:**

* The implementation of AOF seems suboptimal and hence the comparison appears to be biased.  Multiple-day runtimes for n ~ 700 seems high when prior work conducts experiments for n >= 2000 (https://arxiv.org/pdf/2306.08153).  Doing a little digging, I found https://www.kdd.org/kdd2016/papers/files/rpp0224-yuanA.pdf, which ran experiments up to n ~ 8192.  The limits of the BSR approach are not really demonstrated besides the statement "Even sizes of n = 10,000 or more take at most a few seconds."  How much beyond this can you scale?  Would be good to add a scalability experiment.

**Questions:**

* Assuming your method scales well beyond the prior work of n ~ 8192, can you make a convincing case that in modern DP + ML applications we need to scale beyond that point?  I'm not super familiar with how many iterations DP-SGD is usually run for.

* Thm 2 is an interesting result that allows you to compute sensitivity efficiently under this special Toepltiz structure without requiring bandedness, but it's not clear where you are using this Thm.  Is it better to find a banded factorization with bands = minsep or use a non-banded factorization with Thm 2?
>* Also minor point: might be better to write Eq 10 in terms of m_0, ... m_{n-1} instead of M (or maybe include both expressions).
>* There is an incorrect reference to Eq 10 in the experiments
>* Typo: n_{n-1}

* Regarding the statement: "Apart from the factorization itself, the computational cost of BSR and AOF are nearly identical."  Is the factorization time usually the dominating term in the cost for AOF?  Is that still the case for BSR, even when scaling up well beyond n ~ 8192?  What factor(s) other than n (if any) impact the complexity of these mechanisms?

* It is remarkable that the approximation quality of BSR and AOF is so close.  It looks like you evaluated a few settings, one of them being b=100, k=n/100 in Fig 1.  Does this hold more generally across other settings, and can you demonstrate that with an experiment?  There are two extremes where it I might expect *some* gap: (a) 2 bands (b) n bands.
>* In Fig 1, the most interesting comparison to me is BSR vs. AOF, the baselines have already been established to be far from optimal in prior work and their presence makes it difficult to perceive difference between the two most interesting lines, would be good to update the plot accordingly.

**Limitations:**

The authors have discussed some limitations, but perhaps adding some of the things I mentioned in this review (or addressing them directly) would be good.

---

> ### Author Rebuttal · Authors · 2024-08-07
>
> Thank you for the valuable feedback. Below we clarify the raised points:
>
> **\> Implementation of AOF seems suboptimal.**
>
> (note that Reviewer VxsQ had a similar question, and for easier reading we provide our answer in both our replies)
>
> There are, of course, many ways to solve AOF (4). We used SCS, because it is a dedicated SPD-solver that allows a simple and reproducible implementation without additional hyperparameters (Algorithm 2). General-purpose gradient-based solvers, such as L-BFGS, were meant by our comment about approximate solvers (lines 122-125). These can be faster for solving (4) up to a certain precision, but they require additional design choices, such as the learning rates, number of update steps, a barrier function to enforce the constraints, and potentially projections to ensure positive definiteness.
>
> **\> References on larger-scale matrix factorizations.**
>
> Ref arXiv:2306.08153 uses an in-house implementation by Google that is not freely available. The manuscript does not state runtimes or hardware. However, the fact that they also only report experiments up to $n=2,052$, we take as confirmation that solving (4) for large $n$ is indeed challenging. Regarding the KDD2016 reference, we do not believe it is comparable. It solves a different optimization problem (in fact, AOF’s formulation (4) was just proposed in 2023), which is also an SDP in the context of DP but with a simpler constraint set. Nevertheless, we’ll be happy to include it in our discussion.
>
> Importantly, however, note that our claim in the work is not that “BSR is x-times faster than AOF”. Actual runtimes always depend on many factors, including hardware, programming language, software packages, etc. We mention runtimes rather as a qualitative guidance to the reader, which is why we do not report a formal table or plot of runtimes. Our core argument is that solving (4) is a challenging optimization problem. *Any* numeric solver will be slower and less convenient to use than our $O(n \log n)$ closed-form expressions. In real-world problems solving (4) creates a bottleneck, especially because it has to be solved anew for any change of hyperparameters. BSR overcomes this bottleneck.
>
> **\> Scalability of BSR.**
>
> We reported experiments to a size of 2000 only due to the computational limits of AOF. For BSR, much larger workload sizes are no problem. We will be happy to add them to the manuscript. The speed advantage of BSR over AOF will increase with $n$, regardless of the solver. We illustrate this in the response PDF: a plain python implementation of Theorem 1 (Alg.3 in PDF) computes the BSR-coefficients for $n=10,000$ in 24ms , for $n=100,000$ in 2.4s and for $n=500,000$ in ~66s (CPU, single-threaded). If a dense matrix is wanted for C, times are approximately 3x higher due to the memory access overhead (Table 1 in PDF).
>
> **\> Scale of $n$ for modern DP + ML applications?**
>
> In this context, $n$ is simply the total number of model updates, so large values of $n$ are quite common for standard network training. E.g., training MNIST (60,000 examples) with batchsize 128 for 10 epochs requires $n=(60000/128)*10= 4680$. A standard ResNet-Training on ImageNet1K (1.2M examples, batchsize 512, 90 epochs) yields $n=225,180$. Llama2-like LLM training (2T tokens, batchsize 4M tokens, 1 epoch) would result in $n=500,000$. However, in practice one would not use vanilla MF-SGD for such sizes, because the resulting $C^{-1}$ matrix would require 1 TB of memory to store (in dense form). It remains an open research question how to scale MF-SGD to such sizes. Our contribution only covers a part of that, namely avoiding the need to solve the preparatory (4).
>
> **\> Theorem 2**
>
> We use this theorem to compute b-min-sep sensitivity for all cases, including Theorems 6 and 7. The idea of using Theorem 2 to find a better (potentially non-banded) factorization is indeed interesting. However, implementing this is not straightforward, because Theorem 2 requires the matrix $C$ to have decreasing coefficients. It is unclear how one could impose this property during the optimization, where one only has access to $S = C^TC$.
>
> **\> Is the factorization time the dominating cost?**
>
> For AOF: yes. The factorization (4) is by far the most costly step, regardless of the solver. The Cholesky-type decomposition (line 118) is also costly but faster (seconds for $n=10000$, minutes for $n=50000$). BSR avoids both steps. Instead, one computes $C$ explicitly, which is efficient due to Theorem 1 (see above), and computing C’s sensitivity is trivial due to Theorem 2. The speed of AOF and BSR are both functions of $n$ and the bandwidth. The runtime of numeric solvers for AOF will also depend on the desired precision and the entries of the workload matrix (which determine the hardness of the optimization problem).
>
> **\> Notation and typos.**
>
> Thank you for the suggestions, we will include them.
>
> **\> It is remarkable that the approximation quality of BSR and AOF is so close**
>
> We found BSR to be consistently close to AOF across all settings we tested, not just the one in the manuscript. We’ll be happy to extend the manuscript with more results.
> Regarding bandwidth: Fig.3 in the response PDF shows that for the recommended p=b, BSR is typically close to AOF even for extreme values of $b$ ($b=2$ or $b=n$). For $p\ll b$ or $p\gg b$, indeed the distance to (b-banded) AOF grows. Some numeric results for $p=n$ can already be found in Appendix C in column “sqrt” (Tables 10-17 for $b=n$, Tables 2-9 for $b=100$). Unfortunately, their captions were mislabeled, which we will fix.
>
> **\> Figures**:  the baselines were meant as a reference to help judge the significance of the differences between the methods. However, we could put the plots without baselines in the appendix, or if preferred, we could put the non-banded “sqrt” instead. Please let us know what you would prefer. We hope that also the numeric tables in the appendix allow judging the differences.

---

> > ### Comment · Reviewer_1wZc · 2024-08-07
> >
> > Thank you for the response, I went ahead and bumped up my rating -- I think this is a solid paper with nice impact that should be of interest to people working in the space DP ML.   Three things I'd like to see addressed in the revision:
> >
> > 1. I would like to see in the final version the results from Appendix C expanded and included in the main text.  From what I can tell, there is a 10%+ degredation in expected error for BSR in Table 2, which could be meaningful in practice.  Including plots that show how this degredation changes with n and b would be informative and strengthen the experiments.
> >
> > 2. The results on AOF should also be cleaned up -- make sure you are using an implementation that converges to the optimal solution so you don't end up with strange artifacts where BSR outperforms AOF.  If your custom implementation of AOF is not converging, try modifying an existing implementation, such as one of the links from Reviewer VxsQ below, or finding the source from the kdd paper.  Alternatively asking the authors of arXiv:2306.08153 for their source code might be worth a shot (if you haven't done so already).
> >
> > 3. Add limitations section, discussing challenges to scaling to large n beyond solving Eq (4).
> >
> > Also, please clarify two follow-up questions:
> >
> > I'm not sure how you were able to run Algorithm 3 BSR(..., full='true') for n >= 100000 -- wouldn't that require 40 GB of RAM?  Do you need to set full='true' to use this mechanism in an ML training pipeline?
> >
> > One final comment: It is clear that C has a closed form expression but from Eq (6) it seems like you compute B by materializing A and C, and computing A C^{-1} using something like numpy, which I wouldn't expect to scale well due to the size of the matrices.  Does B also have a Toeplitz structure that you exploit to represent it efficiently?

---

> > > ### Author Response · Authors · 2024-08-09
> > >
> > > Thank you for your feedback and for increasing your rating. We appreciate your insights and will address your recommendations in the revision. Below are responses to your follow-up questions:
> > >
> > >  **\> Memory Usage**:
> > > Yes, running Algorithm 3 with $n=100{,}000$ does require around 40 GB of RAM and $n=500{,}000$ approximately 1 TB. We used a machine with 1.5 TB of RAM for these experiments. However, setting `full='true'` is not necessary for using this mechanism in an ML training pipeline.
> > >
> > > **\> Matrix Structures and Efficient Representation**:
> > > The matrices $A$, $B$, and $C^{-1}$ are indeed all Toeplitz. However, $A$ and $B$ are not explicitly needed for MF-SGD, only implicitly (see Alg.1). In our demo code, we instantiate $C$ and $C^{-1}$ explicitly for simplicity, but for large-scale applications, this would not be necessary. The Toeplitz structure allows for efficient representation, avoiding the need to materialize these matrices fully.

---

### Official Review · Reviewer_sFEq · 2024-07-12

**Soundness:** 4
**Presentation:** 3
**Contribution:** 3
**Rating:** 7
**Confidence:** 3

**Summary:**

This paper addresses optimal matrix factorization mechanism for differential privacy, focussing on stochastic gradient descent (SGD) optimization.
It introduces the banded squared root (BSR) factorization and provide formal lower and upper bounds to the approximation error.
The BSR achieves competitive formal approximation error compared to state-of-the-art methods and demonstrates practical utility in real-world training tasks.
The authors provide theoretical bounds for single and repeated participation training, showing that BSR maintains high training accuracy with provable privacy guarantees while being computationally efficient.

**Strengths:**

- The authors make a significant contribution to the field of differentially private machine learning under matrix factorization mechanisms by leveraging efficient BSR for the SGD workload matrix.
- The incorporation of the SDG workload matrix in this line of research is elegant.
- The banded squared root (BSR) factorization is computationally efficient and scalable to large workload matrices, making it a valuable tool.
- The proof sketches are intuitive and convincing, although I haven't thoroughly reviewed the technical proofs in the appendix.
- A notable advantage of BSR is its independence from specific training objectives.
- BSR achieves competitive approximation error, on par with state-of-the-art methods in differentially private training.
- The approach demonstrates practical utility by maintaining high training accuracy in real-world training tasks.
- The authors provide theoretical guarantees for both single and repeated participation scenarios, adding to the method's credibility.
- The theoretical explanations are (for the most part) well-written and easy to follow.
- The paper includes extensive technical supplementary material.
- The inclusion of useful, copy-pasteable Python code listings is interesting.
- The authors show respect for existing work by referencing relevant StackOverflow answers and software packages.

**Weaknesses:**

- The experimental evaluation is limited to a small set of real-world and synthetic experiments.
- The range of synthetic datasets used in the experiments is restricted, which may not be representative of diverse scenarios.
- The data generation process is unclear, and only homogeneous partitions are considered, with no variation in data distributions.
- The real-world experiments are limited to a single dataset, CIFAR-10, which is insufficient to draw general conclusions.
- The post-processing step (line 275) lacks clear explanation, leaving it unclear whether it addresses a principled problem or a numerical issue.
- The experiments do not provide a comparison with non-private approaches, which makes it difficult to understand the practical limitations of the method.
- The evaluation is limited to a single real-world dataset, CIFAR-10, and a simple ConvNet model.
- The experiments do not discuss potential practical limitations of BSR, which may impact its applicability.
- The comparative analysis is limited and does not include a wide range of existing differential privacy methods.
- The formal proof of Theorem 2 (Appendix D.2) employs unclear and undefined notation (Π), which makes it difficult to follow.
- Neither the proof sketch nor the formal proof are straightforwardly understandable, which may hinder comprehension.

Minor:

- The tone of "supposedly optimal" (line 254) could be perceived as inappropriate and may benefit from rephrasing.
- The proof sketch of Theorem 2 refers to the wrong appendix (Appendix D.3 instead of Appendix D.2, presumably).

**Questions:**

./.

**Limitations:**

./.

---

> ### Author Rebuttal · Authors · 2024-08-07
>
> Thank you for the encouraging feedback and detailed comments.
>
> **\> Experimental Evaluation.**
>
> We believe there may be a slight misunderstanding. Our main contribution lies in the algorithm and properties of BSR. Specifically, we demonstrate that in the context of MF-SGD, using BSR requires adding less noise compared to the baselines (for identical levels of privacy) and it is essentially on par with the computationally much more costly AOF (e.g., Figure 1). This fact is independent of any subsequent model training steps and requires no data, not even synthetic. We do not introduce a new technique for differential private model training, which would indeed require more extensive experimental validation and comparison to other private training techniques.
>
> The MF-SGD experiments on CIFAR that we report (Figure 2) are only meant as a proof-of-concept to illustrate that, by keeping all other components fixed, the lower added noise tends to translate into higher accuracy. The exact experimental setup we will be happy to clarify in the manuscript: like prior work, we split the dataset uniformly at random into batches and train in a centralized way.
>
> **\> Limitations of BSR.**
>
> We will be happy to extend our discussion in Section 5 to provide a broader view. In particular, we observe that BSR achieves results comparable to AOF, although we currently cannot prove this, as the theoretical properties of AOF are not well understood enough. Regarding real-world usability, the analytic expression for BSR does not apply to variable learning rates, which would be desirable.
>
> **\> Post-processing step (line 275)**
>
> Note that this step is unrelated to BSR; it was only needed for AOF. We believe the problem is both numerical and principled. By this, we mean that, on the one hand, the problem is indeed numerical: if we could solve (4) with infinite mathematical precision, post-processing would not be needed. On the other hand, the numerical issues arise because of a principled drawback, namely that the AOF procedure (solving (4), computing a Cholesky decomposition, inverting the resulting matrix) is quite sensitive to small numeric deviations. Consequently, errors from solving (4) only to a certain precision, as all numerical solvers do, can lead to disproportionately large undesirable effects.
>
> **\> Comparison to non-private approaches.**
>
> We will be happy to include such information in the manuscript. For example, non-private training in the setting of Figure 2 achieves an accuracy of approximately 65%. However, we would like to emphasize that our contribution is orthogonal to the contrast between private and non-private training. Our submission shows that by using a different processing step (BSR), one benefits in terms of speed (over AOF) and accuracy (over the baselines). Outside the context of MF-SGD, such a comparison cannot be made.
>
> **\> Notation ($\Pi$), clarity of proof and sketches.**
>
> Thank you for bringing this to our attention. We will clarify the proof and sketches and we will remind the reader of the definition of $\Pi$.
>
> **\> Minor issues.**
>
> Thank you, we will fix these items.

---

### Official Review · Reviewer_Cfm2 · 2024-07-12

**Soundness:** 4
**Presentation:** 4
**Contribution:** 3
**Rating:** 6
**Confidence:** 4

**Summary:**

The paper considers the problem of adding correlated noise $C^{-1}z$ instead of independent noise $z$ across iterations in continual counting (equivalently, DP-SGD). Past work gave an algorithm to choose $C^{-1}$ that optimizes some objective on the noise under b-min-separated participation, but this algorithm requires solving an SDP which is infeasible when using a large number of iterations, i.e. optimizing over $C$ that has a large number of rows / columns $n$. The authors propose a choice of $C$, which they call banded square root, which has a succinct representation that is efficiently computable in time independent of $n$. Furthermore, their choice of $C$ is banded, i.e. is non-zero below the $b$-th diagonal for $b \geq 1$, which gives it some nice properties such as being able to compute $C^{-1}z$ efficiently in a streaming manner. To define the banded square root $C$, they authors take $A$ which represents the workload, effectively a representation of the updates in SGD which can include momentum and weight decay, and take its matrix square root. They then truncate the matrix square root to its first $b$ diagonals. The authors give an explicitly and efficiently calculable formula for the computing the first column of this matrix, which specifies the whole matrix as it is Toeplitz. The authors analyze the asymptotic error guarantees of different choices of $C$, including the banded square root, in the single- and multiple-participation settings. Specifically, their error guarantees show that banded square root improves on $C = I$ or $C = A$, and nearly matches a lower bound on any factorization if using enough bands. The authors conduct experiments training models for classification on CIFAR10 and show that banded square root is competitive with the choice of $C$ given by solving a more expensive SDP.

**Strengths:**

* The work makes theoretical progress on a practical problem. Correlated noise/DP-MF is now being used in practice to train models with DP, and especially as model sizes and training runs get larger, more efficient ways to implement DP-MF will lead to better models in practice. In particular, itmakes the results of the DP-MF literature more accessible to those who do not have large amounts of compute to use the techniques they introduced.
* The theoretical analysis of the error of DP-MF as a function of the number of bands is novel and adds theoretical understanding of banded $C$ that was not present in the previous paper of Choquette-Choo et al. Furthermore, the set of theoretical results is rather extensive and gives a pretty complete theoretical understanding of the authors' methods.
* While DP-MF is a relatively niche topic, the paper does a good job slowly introducing the problem, their approach, and their theoretical results.

**Weaknesses:**

* While the previous work did require solving an SDP, to my understanding this is for the case where $C$ can be is only required to be a banded lower-triangular matrix. If $C$ is required to be Toeplitz as well, as the banded square root factorization is, then one only needs to optimize over $b$ variables instead of $\approx nb$ and it is not clear that the computations in the previous work are expensive, which mitigates the improvements in this paper.
* It is worth pointing out there is a work of https://arxiv.org/abs/2404.16706, that also gives an efficient-to-compute way to choose $C$, and leads to faster streaming noise generation for than banded matrices. This work I would consider concurrent, so I did not account for any possible overlap between the two papers' results and impact when assigning my score.

**Questions:**

* wrt the first weakness, did the authors consider this alternate approach / if so, do you believe banded square root still offers speedups in this setting?
* The previous work of Choquette-Choo et al. shows that banded $C$ are compatible with privacy amplification with sampling. Have you considered combining your error analysis with their privacy analysis (e.g., maybe for RDP to simplify) in the setting where batches are sampled in DP-SGD? It might be an interesting question to see how the number of bands affects error in the presence of amplification, since they observed more bands reduce the benefits of amplification.

**Limitations:**

Yes

---

> ### Author Rebuttal · Authors · 2024-08-07
>
> Thank you for the insightful review. Here, we address the individual concerns:
>
> **\>  Relation to arXiv:2404.16706.**
>
> Indeed, this interesting preprint (which we cite as [Dvijotham et al., 2024]) is concurrent work. It studies only the case of MF-SGD without momentum or weight decay (“prefix sum”) and is limited to streaming data ($k=1$, “single epoch”).  This setting enables a deeper analytic treatment (in particular the sensitivity computation, which e.g. does not benefit from bandedness), but it comes at the cost of covering fewer real-world scenarios.
>
> **\> While the previous work did require solving an SDP, to my understanding this is for the case where can be is only required to be a banded lower-triangular matrix. If is required to be Toeplitz as well, as the banded square root factorization is, then one only needs to optimize over variables $b$ instead $nb$ [...]**
>
> This is an interesting observation that could lead to a hybrid between [Choquette-Choo et al., 2023a] and our work. As a compromise between speed and optimality, one could keep the objective of the problem (4) but restrict it to Toeplitz matrices, thereby parametrizing the problem with $b$ variables. Unfortunately, the objective function would be quite more involved. It would no longer have a standard structure, such as the “matrix fractional function” form of the SDP (4), and it would not be convex in its variables (AOF’s (4) is only convex in the product matrix $S=C^T C$, which enters the objective via $S^{-1}$). Gradient-based optimizers could converge quickly but one would lose the guarantee of being close to a global optimum within a specifiable tolerance.
>
> **\> Did the authors consider this alternate approach / if so, do you believe banded square root still offers speedups in this setting?**
>
> We had not considered this approach, but it sounds like promising future work that we would be happy to follow up on (with the permission of the reviewer). That said, we believe it would be unfair to consider it a shortcoming of our work that an alternative path exists which, to our knowledge, has not appeared in any prior work and will come with its own challenges.
>
> Regarding the speed: even if the above idea might accelerate the numerical optimization, it will not surpass the simplicity and efficiency of BSR’s closed-form expressions. Furthermore, even if solved optimally, the quality of such a solution should lie between BSR and AOF, which are close together already.
>
> **\> Privacy amplification.**
>
> Another interesting point, thank you for bringing it up. Since the BSR matrix has the same band structure as the AOF matrix, the privacy properties amplification arguments from Choquette-Choo et al. will still be applicable to our results. We will add a note to this in the manuscript. We did not include this aspect in the original submission because the material was already quite dense and we considered it somewhat orthogonal to our main analysis. But indeed, it would be interesting future work.

---

> > ### Comment · Reviewer_Cfm2 · 2024-08-11
> >
> > Thanks for your response. Of course, feel free to pursue any of the directions discussed here, if they interest you. I remain in support of accepting the paper.

---

### Author Rebuttal · Authors · 2024-08-07

Attached is our response PDF; please see the responses to the individual reviews for details.

---

### Decision · Program_Chairs · 2024-09-25

**Decision:**

Accept (poster)

**Comment:**

This paper studies the problem of optimal matrix factorization mechanism for differentially private training. Specifically, the authors develop a banded square root factorization approach and provide formal lower and upper bounds on the approximation error.

All the reviewers agree that the paper presents solid results and makes significant contributions to the field of differentially private training. In light of this, I recommend acceptance and encourage the authors to incorporate the reviewers' suggestions into the revision.